# Mastering Board Games by External and Internal Planning with Language Models

**John Schultz** [* 1] **Jakub Adamek** [* 1] **Matej Jusup** [* † 2] **Marc Lanctot** [1] **Michael Kaisers** [1] **Sarah Perrin** [1]
**Daniel Hennes** [1] **Jeremy Shar** [1] **Cannada Lewis** [3] **Anian Ruoss** [1] **Tom Zahavy** [1] **Petar Veličković** [1]
**Laurel Prince** [1] **Satinder Singh** [1] **Eric Malmi** [** 1] **Nenad Tomašev** [** 1]

## Abstract

Advancing planning and reasoning capabilities of Large Language Models (LLMs) is one of the key prerequisites towards unlocking their potential for performing reliably in complex and impactful domains. In this paper, we aim to demonstrate this across board games (Chess, Fischer Random / Chess960, Connect Four, and Hex), and we show that search-based planning can yield significant improvements in LLM game-playing strength. We introduce, compare and contrast two major approaches: In *external search*, the model guides Monte Carlo Tree Search (MCTS) rollouts and evaluations without calls to an external game engine, and in *internal search*, the model is trained to generate in-context a linearized tree of search and a resulting final choice. Both build on a language model pre-trained on relevant domain knowledge, reliably capturing the transition and value functions in the respective environments, with minimal hallucinations. We evaluate our LLM search implementations against game-specific state-of-the-art engines, showcasing substantial improvements in strength over the base model, and reaching Grandmaster-level performance in chess while operating closer to the human search budget. Our proposed approach, combining search with domain knowledge, is not specific to board games, hinting at more general future applications.

## 1. Introduction

While large language models (LLMs) perform fluently on text generation, language understanding and translation, they are prone to hallucinations and reasoning errors, especially in complex contexts (Chang et al., 2024; Hadi et al., 2023). Hence, special attention has been given to the development of planning and reasoning capabilities in LLMs (Minaee et al., 2024). In terms of Kahneman's cognitive theory of two systems (Kahneman, 2011), prior work primarily improved associative System 1 inference in language models, whereas planning and reasoning now focuses on improving the more deliberate System 2 thinking (Plaat et al., 2024).

Planning and reasoning approaches typically fall into one of two distinct categories: In **internal planning** the LLM develops a plan *in context*, like Chain-of-Thought prompting (Wei et al., 2022), by autoregressively considering possible steps towards the goal and their consequences. By contrast, **external planning** uses the LLM to generate steps in a neurosymbolic system, such as in Tree of Thought (Yao et al., 2024), where an outer loop performs explicit search over possible sequences of steps. This paper presents how language models can be trained for internal and external planning to improve reasoning in sequential decision-making, using board games as an experimental domain.

Board games have historically played an important role in the development of automated decision-making, with Torres' automaton *El Ajedrecista* playing three-piece chess endgames before the advent of digital computation (tor, November 1915). Games provide diverse reasoning challenges about both environment dynamics and opponent strategies, and thus have gained significant attention in pushing the boundaries of LLM reasoning capabilities (Hu et al., 2024b; Costarelli et al., 2024; Duan et al., 2024). LLMs have incidentally been struggling with reliably playing common board games, like chess or even tic-tac-toe (Topsakal & Harper, 2024). This may seem somewhat surprising, considering how much headway has been made in other areas. However, as *games astutely expose the inability of LLMs to consistently reason over possible futures with world models*, they make a great testbed for planning and reasoning.

---

[*]Equal contribution [**]Equal senior authorship [†]Research conducted during an internship at Google [1]Google DeepMind [2]ETH Zürich [3]Google. Correspondence to: Eric Malmi <emalmi@google.com>, Nenad Tomasev <nenadt@google.com>.

Making progress in game-playing could inform how to best instill this ability in LLMs going forward. We present several contributions towards this aspirational goal.

> **Contribution 1: *MAV model*.** We pre-train a Transformer model, the multi action-value model (MAV), capable of playing several board games (Chess, Chess960, Connect Four, Hex) at a strong level. This model is capable of reliably tracking the board state throughout games, and makes (good) legal moves.

> **Contribution 2: *External search*.** We use MAV within an external MCTS controller, as a value/implicit policy and transition function. Our Async MCTS reaches Grandmaster level with the number of moves considered per decision ($\sim$100 to $\sim$1k) comparable to human players (for reference, AlphaZero (Silver et al., 2017) used $\sim$10k simulations and traditional engines use up to millions).

> **Contribution 3: *Internal search*.** We distill the search procedure directly into the LLM, generalizing the Stream of Search (SoS) (Gandhi et al., 2024b) approach to a significantly more complex domain. The performance of the resulting agent scales smoothly with the given search budget.

## 2. Multi-Action-Value Model

The multi-action-value (MAV) model is a Transformer model pre-trained exclusively on textual game data that functions simultaneously as a: ($i$) world model, ($ii$) value function, and ($iii$) policy function for multiple perfect-information board games. Acting as world model further requires the following capabilities: ($i$) state-tracking (determining the new state of the game after a move has been played in the previous state), ($ii$) legal move prediction, ($iii$) terminal state detection. To achieve this, MAV is trained on examples following a flexible format illustrated in Figure 1.

**Command specification.** MAV input starts with a header containing a series of commands specifying the game being played, the inputs provided and the expected outputs.

**State representation.** Each game uses a different textual format to represent its state, used with the `%state` and `%prev_state` commands. All of them are designed to be easy for the model to navigate and manipulate: each field on the board is a separate token, and fields maintain constant relative positions in token space. For chess we additionally support using the standard FEN representation (with the `%fen` command).

**Starting state.** The starting state for the model can be given in two ways: either directly with the `%state` command, or as in Figure 1 with the commands `%prev_state` `%prev_action %state`, which instruct the model to take a previous state (`%prev_state`) and a move played in that state (`%prev_action`) and use these to infer and output the starting state (`%state`). The latter functionality makes the model act as a transition function.

**Value function.** The `%top_k` command instructs the model to consider the starting state and output the best $k$ legal moves and their action values, in order of preference from highest action value to lowest. $k$ is configurable to allow for varying the amount of inference-time computation MAV performs. If there are fewer legal moves than $k$, the model simply outputs all of them (this can also be achieved by setting $k$ to "*all*"). In the case where the starting state is terminal, the `%top_k` command outputs the game outcome instead (e.g., [`%top_1 invalid : "1-0"`] when the first player has won).

The action value for a move corresponds to the predicted win probability for the player in the current state if this move is taken. For chess, state-action values are represented similarly to Ruoss et al. (2024a): Stockfish centipawn evaluations are mapped to win probabilities using the formula

$$\text{Win \%} = 50 + 50 \left( \frac{2}{1 + e^{-0.00368208 \cdot \text{centipawns}}} - 1 \right)$$

obtained from https://lichess.org/page/accuracy. Win probabilities are then mapped to discrete non-overlapping buckets, making the value prediction problem a classification task rather than a regression task. This has proven to be beneficial in other works (Farebrother et al., 2024). We use 64 buckets, each represented by a different special token (e.g., `<ctrl28>` for bucket 28). Figure 2 shows two win probability distributions predicted by MAV. Connect Four and Hex use game engines Fhourstones (Tromp) and neurobenzene (Gao & Pawlewicz), respectively, to obtain state-action values, which are similarly mapped to the same 64 bucket tokens.

When producing training data we vary $k$ randomly, making sure to include examples with $k$ greater than the number of legal moves. We randomize the order of the moves to: ($i$) encourage the model to treat the moves independently, and ($ii$) help to prevent hallucinations. With a fixed order, e.g., lexicographical from $a1a2$ to $h8h7$, then ($i$) if the model skipped a legal move $a1a4$ and emitted $a1a5$, it couldn't go back and fix its mistake, and ($ii$) the model might learn to heavily lean on evaluations of previous moves when considering a move like $h8h7$, which is late in the order. Using randomised ordering of moves in the training data, the model is steered towards an approximate *permutation symmetry* (Bronstein et al., 2021).

**Value Definition: Scoring Methods.** During inference,

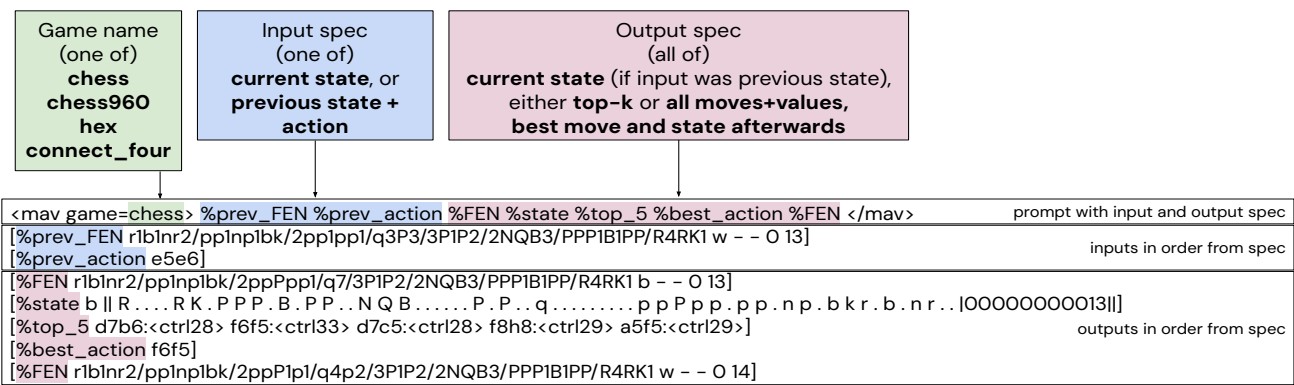

Figure 1. A complete example used for training the multi-action-value models. Robert J. Fischer (2690) – Mijo Udovčić (2460), Bled 1961, 1/2–1/2.

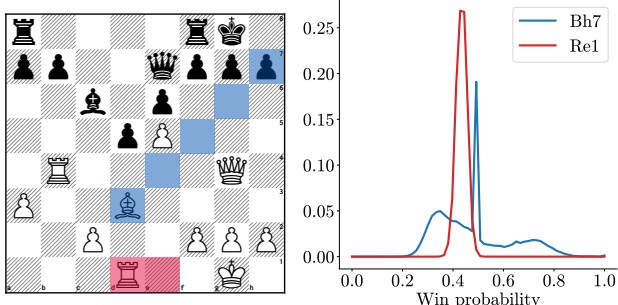

Figure 2. For a safe positional move, such as ♖e1, MAV assigns little uncertainty and gives a win probability of 40-45% since white is a pawn down. In contrast, the bishop sacrifice ♗×h7+ (best move according to Stockfish) is a riskier attacking move and thus has substantial probability mass around low, equal, and high win probabilities.

for each move the model produces a distribution over the 64 tokens that correspond to action value buckets. We experiment with two methods to turn this distribution into a final score for the move: *max scoring*, which corresponds to greedy decoding, uses the mode of the model's distribution, while *mean scoring* uses the expectation over buckets.

Mean scoring (also used in Ruoss et al. (2024a)) utilizes more of the model's information, and is better able to distinguish actions with equal modes. In an almost tied position, the most likely outcome for many moves may be a draw, but still some moves may have a higher winning chance than others. Mean scoring will be able to pick the best move in this case, while max scoring can not. This is further illustrated in Figure 2.

**Best action prediction.** In positions where the model has a large advantage, it may have many moves available with ca. 100% win probability that don't directly lead to finishing the game by checkmate. This may cause the model to play aimlessly in such positions. This problem was encountered

in Ruoss et al. (2024a), which they resolved by using Stockfish to break ties when all top 5 moves lie above a win probability of 99%. To eliminate reliance on a game engine at inference time, we introduce the %best_action command, which instructs the model to emit the game engine's chosen action in the position. In this way we teach the model the tie-breaking procedure described above.

**Novelty.** To summarize, MAV makes four key improvements over the existing state-of-the-art Transformer-based chess engines (Ruoss et al., 2024a; Monroe & Leela Chess Zero Team, 2024; Czech et al., 2024; Farebrother et al., 2024) First, it performs world modeling, policy and action-value computation together in one model. Second, it is trained to output the best action at the end of action-value modeling, enabling the model to reliably finish games where it has a decisive advantage. These two improvements enable MAV to play complete games without relying on an external game engine for legal moves or for finishing the game. Third, all of the above steps can be done in a single model call without having to evaluate every action separately—an important feature that guided the design of the MAV format to reduce the cost and infrastructural complexity of inference. Fourth, the amount of compute performed by the model at inference time can be varied dynamically, enabling us to achieve higher quality at the cost of higher latency when performing both internal and external search, where the performance increases as we scale the planning.

**Datasets.** We curate a dataset of diverse, relevant positions in four games: Chess, Chess960, Connect Four, and Hex. The statistics and sources for each of these datasets are shown in Table 4 in Appendix B. Each position is used to produce a single training example, randomly varying (*i*) the $k$ action values in %top_k, (*ii*) the presence of the initial or final state tracking commands, (*iii*) the use and order of %state or %FEN representations in chess. Further details on the datasets are provided in Appendix B.

**Models.** We train two randomly initialized decoder-only Transformer models using the Gemini architecture (Gemini Team et al., 2024), called MAV and MAV–small,[1] with 2.7 billion and 1 billion parameters respectively[2]. They were trained on 1.9 and 1.2 epochs of the dataset. Except for the max scoring result for MAV–small in Table 1, all other experiments and results use the 2.7 billion MAV. In order to efficiently use our models' parameters, the input part of each training example is masked out during loss computation. This means our models do not waste capacity on learning to generate game positions.

## 3. External Search

External search employs MAV to generate planning steps, and applies a search algorithm to direct and optimise over sequences of planning steps. In this paper, we evaluate planning on top of the previously discussed MAV, using MCTS as the search procedure.

External search is based on an AlphaZero-style MCTS (Silver et al., 2017). There are two key ingredients: a **prior** function which returns a probability distribution over actions at each state, and a **value** function returning a numerical value indicating value of a state or state-action pair, both extracted from the MAV output. External search is an adaptation of MCTS; in its most basic form, as in AlphaZero, MCTS relies on an explicit world model, querying a game engine (e.g., a chess implementation in OpenSpiel (Lanctot et al., 2019)) to provide legal actions, state transitions and terminal states. Inspired by MuZero (Schrittwieser et al., 2020), in a subsection below we describe how we remove this dependency on the game engine and instead use only MAV to track states, transitions, and provide legal actions and game outcome during planning.

The external search algorithm guided by a learned world model is summarized in Algorithm 1, with subroutines contained in Appendix A. A search is started at an initial game state $s_0$. First, the legal actions $\boldsymbol{a}_0^L$ and associated state-action values $\boldsymbol{Q}^{(i)}(s_0, \boldsymbol{a}_0^L)$ from active player's $i$ perspective are obtained from MAV. The prior function is an $\varepsilon$-greedy policy, derived entirely from state-action values comprised of a greedy softmax over the top $k$ values, mixed with a uniform distribution over all actions to encourage exploration. Specifically, let $\boldsymbol{a}^{g,k} \subseteq \boldsymbol{a}^L$ be the subset of legal actions $\boldsymbol{a}^L$ whose values are among the top $k$-ranked values, or equal to $\boldsymbol{a}^L$ if $|\boldsymbol{a}^L| \leq k$. Define the probability

---

[1]As of May 21, 2025, it is possible to play chess against MAV–small at https://goo.gle/ChessChamp.

[2]Note that in practice MAV and MAV–small don't utilize many natural-language specific parameters.

---

**Algorithm 1** EXTERNAL-MCTS($s_0$)

1: **Input:** Initial state $s_0$, active player $i$, top $k$ values, num. simulations $M$
2: **Output:** Recommended action $a^*$
3: $\boldsymbol{a}_0^L, \boldsymbol{Q}^{(i)}(s_0, \boldsymbol{a}_0^L) = \text{MAV}(s_0)$
4: Compute prior $\boldsymbol{P}(s_0, \boldsymbol{a}_0^L; k)$ according to Eq. 1.
5: Initialize root node:
$\quad \mathcal{N}_0(i) \leftarrow i$
$\quad \mathcal{N}_0(s) \leftarrow s_0$
$\quad \mathcal{N}_0(\boldsymbol{a}^L) \leftarrow \boldsymbol{a}_0^L$
6: EXPAND($\mathcal{N}_0, \boldsymbol{P}$)
7: **for** $m = 1, \ldots, M$ {this can be async} **do**
8: $\quad$ SIMULATION($\mathcal{N}_0, \text{MAV}, k$)
9: **end for**
10: $a^* \leftarrow$ FINALMOVESELECTION($\mathcal{N}_0$)
11: **return** $a^*$

---

of action $a$ in state $s$ under the greedy policy to be

$$\pi^{g,k}(s, a; \boldsymbol{Q}^{(i)}) = \begin{cases} \frac{\exp \frac{1}{\tau} Q^{(i)}(s,a)}{\sum_{a' \in \boldsymbol{a}^{g,k}} \exp \frac{1}{\tau} Q^{(i)}(s,a')} & \text{if } a \in \boldsymbol{a}^{g,k}; \\ 0 & \text{otherwise,} \end{cases}$$

where $\tau$ is a temperature parameter. In practice, we dynamically adapt the temperature (Veličković et al., 2024) as a function of the number of moves played and transform state-action values to win probabilities before computing softmax.

The probability of the uniform policy is $\pi^u(s, a; \boldsymbol{a}^L) = \frac{1}{|\boldsymbol{a}^L|}$. The prior probability of taking action $a$ in state $s$ is

$$\begin{aligned} P(s, a; \boldsymbol{Q}^{(i)}, \boldsymbol{a}^L, k) = &(1 - \varepsilon)\pi^{g,k}(s, a; \boldsymbol{Q}^{(i)}) \\ &+ \varepsilon\pi^u(s, a; \boldsymbol{a}^L). \end{aligned} \quad (1)$$

The root node $\mathcal{N}_0$ is then initialized with the active player, string description of a state corresponding to the node, and legal actions (respectively: $\mathcal{N}_0(i)$, $\mathcal{N}_0(s)$, $\mathcal{N}_0(\boldsymbol{a}^L)$). The node is then expanded so that a child node is added for each action and a prior probability attached to the action leading to the child. Then simulations are run: each simulation starts at the root node, actions and children nodes are selected according to PUCT until a leaf node is expanding and evaluated, and values are backpropagated through the nodes that the simulation visited. After $M$ such simulations, a final move $a^*$ is selected.

### 3.1. External Search with State-Tracking MAV

Classical MCTS relies on several components of a game engine to simulate the game. MAV replaces these components by combining state tracking, legal action prediction, action value prediction, and terminal state detection in a single model. This allows us to remove the dependency on a game engine, which results in a few benefits. First, we use the board games to showcase that MCTS achieves astonishing

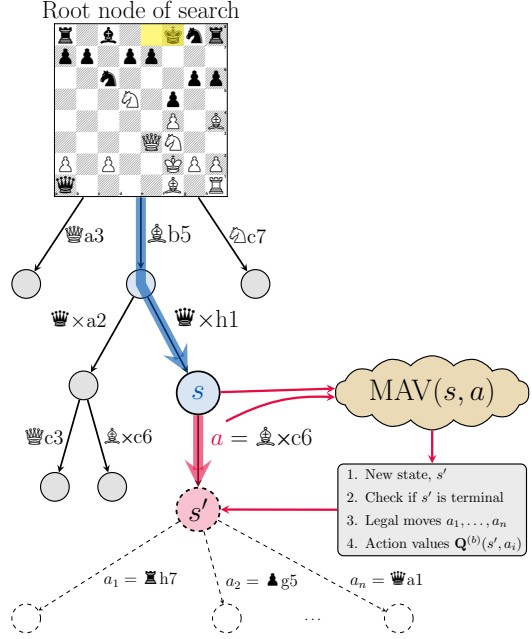

*Figure 3.* State-tracking MCTS with a learned world model.

performance even without access to an explicit world model, which is the case for many interesting real-world problems. Second, it is a step closer towards the internal search with LLMs.

We adapt MCTS (see Algorithm 1) to use the learned MAV to predict the state transition from the string description of a parent node $\mathcal{N}_{t-1}(s)$ and action $a_{t-1}$ to the child node $\mathcal{N}_t$, as shown in Figure 3. We use the obtained information to store a (predicted) string description $s_t$ of the child node, which is then expanded with legal actions $\boldsymbol{a}_t^L$ and their associated state-action values $\boldsymbol{Q}^{(i)}(s_t, \boldsymbol{a}_t^L)$ used to compute a prior according to Equation (1) and the child's value as the maximum over all state-action values, i.e., $\max_{a_t \in \boldsymbol{a}_t^L} Q^{(i)}(s_t, a_t)$.

MAV may hallucinate and return responses that are improperly-formatted. To address this, for each game, a parser function is responsible for translating the output into the next state, legal actions, and their values. If the parsing fails – which can *only* happen if a response violates a pre-defined format – a special value of $-\infty$ is assigned to the node to avoid future consideration. This procedure makes state-tracking MCTS robust to hallucinations, by explicitly avoiding states where the hallucination occurred.

### 3.2. Async MCTS and Dynamic Virtual Counts

Due to the heavy cost of inference in LLMs, we use an Asynchronous Monte Carlo tree search (Async MCTS) that queues multiple leaf evaluations simultaneously (i.e., in a "batch" of simulations). To avoid redundant concurrent

computation, we employ a commonly-used optimization in parallel MCTS: *virtual counts*, which temporarily increase the visit counts by some constant $n_c$ during the simulation, which are then removed during the backpropagation phase. For more details, please see Appendix A.

With low simulation counts, using the virtual losses accompanied by the fixed virtual counts $n_c$ did not strike a satisfactory balance between exploration and exploitation. To address this, we introduce the *dynamic virtual counts* that dynamically assign more weight to the virtual count values closer to the leaf nodes. Suppose simulation $m$ encounters leaf node $\mathcal{N}_l$. We define a virtual count for states-action pairs $(s_t, a_t)$ visited in *simulation $m$ and leaf $\mathcal{N}_l$*

$$n_c(m, \mathcal{N}_l) = \min\left(n_{\min}, \left\lfloor n_{\max} \cdot 2^{d_{\mathcal{N}_t} - d_{\mathcal{N}_l}} \right\rfloor\right),$$

where $d_{\mathcal{N}_t}$ represents the depth of a child node $\mathcal{N}_t$ relative to the leaf node $\mathcal{N}_l$ reached during simulation $m$ and $\lfloor \cdot \rfloor$ is a floor function. As depicted in Figure 7 in Appendix A, it proved beneficial to exponentially decrease the virtual counts starting with the maximum virtual count $n_{\max}$ at the leaf node which is then halved at each parent up to the root node while maintaining a minimum virtual count $n_{\min}$.

## 4. Internal Search

In contrast to *external* search, in *internal* search the model does not require an external controller. Instead, the search procedure is distilled into the model so that it is capable of (*i*) evaluating search nodes (states), (*ii*) expanding nodes while updating the current state, and (*iii*) backpropagating the results from leaf nodes back to the root node—all within a single model call. The distillation is done by linearizing search trees into a text format and training the model on those linearized trees.

**Data.** The prompt for internal search resembles that of MAV, but includes a preamble with search parameters (tree depth and breadth, see Figure 4). The format of the target data is inspired by depth-first order traversal of minimax trees, as an iterative and linearized sequence of minimax tree traversal. Hence, following this format corresponds to an *algorithmic execution* task, akin to CLRS-Text (Markeeva et al., 2024).

The training data is based on target search trees, which were constructed using depth 3 (N.B., depth-zero is MAV), by annotating states (e.g., chess states being annotated with Stockfish), and expanding the top 5 moves into trees. This results in high quality target search trees, similar to those internally generated by game engines. To diversify training data and enable search budget control for trained models, prompts were composed with diverse search parameters, ranging depth 1–3 and breadth 2–5, and continuations yield corresponding trees that were subsampled to match the parameters while fitting into context size. This necessitated

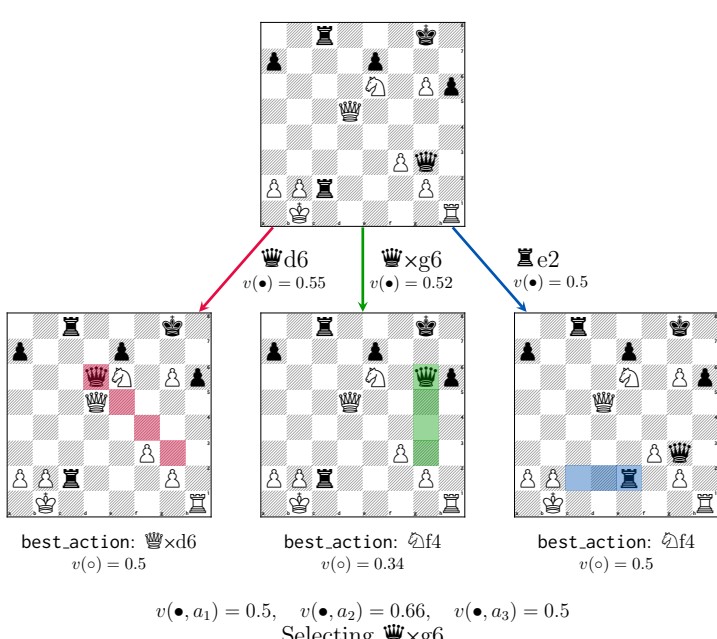

```
Minimax breadth=3 depth=1
Evaluation for node: <root>
<mav game=chess> %state %top_all %best_action </mav>
[%state 2r3k1/p3p3/4N1Pp/3Q4/8/5Pq1/PPr3P1/1K5R b]
[%top_all ... g3g6:<ctrl33> ... c2e2:<ctrl32> ... g3d6:<ctrl35> ... ]
[%best_action g3d6]

Expand from node <root>:
<root g3d6>
<root g3g6>
<root c2e2>

Evaluation for node: <root g3d6>
<mav game=chess> %prev_state %prev_action %state %top_all %best_action </mav>
[%prev_state 2r3k1/p3p3/4N1Pp/3Q4/8/5Pq1/PPr3P1/1K5R b]
[%prev_action g3d6]
[%state 2r3k1/p3p3/3qN1Pp/3Q4/8/5P2/PPr3P1/1K5R w]
[%top_all d5d4:<ctrl18> ... d5b3:<ctrl23> d5d6:<ctrl32> a2a3:<ctrl11> ... ]
[%best_action d5d6]
Decision for node: <root g3d6>
Selecting: d5d6 : "<ctrl32>"

Evaluation for node: <root g3g6>
<mav game=chess> %prev_state %prev_action %state %top_all %best_action </mav>
[%prev_state 2r3k1/p3p3/4N1Pp/3Q4/8/5Pq1/PPr3P1/1K5R b]
[%prev_action g3g6]
[%state 2r3k1/p3p3/4N1qp/3Q4/8/5P2/PPr3P1/1K5R w]
[%top_all ... d5c5:<ctrl11> e6f4:<ctrl22> b2b3:<ctrl1> ... ]
[%best_action e6f4]
Decision for node: <root g3g6>
Selecting: e6f4 : "<ctrl22>"

Evaluation for node: <root c2e2>
<mav game=chess> %prev_state %prev_action %state %top_all %best_action </mav>
[%prev_state 2r3k1/p3p3/4N1Pp/3Q4/8/5Pq1/PPr3P1/1K5R b]
[%prev_action c2e2]
[%state 2r3k1/p3p3/4N1Pp/3Q4/8/5Pq1/PP2r1P1/1K5R w]
[%top_all ... h1h6:<ctrl1> e6f4:<ctrl32> h1c1:<ctrl1> ... ]
[%best_action e6f4]
Decision for node: <root c2e2>
Selecting: e6f4 : "<ctrl32>"

Decision for node: <root> g3d6 : "<ctrl33>" g3g5 : "<ctrl43>" c2e2: "<ctrl33>"
Selecting g3g6 : "<ctrl43>"
Playing g3g6!
```

*Figure 4.* Internal search input prompt and response (breadth=3, depth=1). The response is an actual model response, but for brevity, we only show some of the %top_all action-value pairs. The corresponding minimax search tree is visualized on the right. MAV assigns ♛d6 the highest value, while internal search identifies the only winning move, ♛xg6.

excluding the biggest parameter combination (i.e., omitting depth 3, breadth 5 examples).

An example internal search trace predicted by the trained internal search MAV (MAV–IS), along with the corresponding search tree, is shown in Figure 4.

**Training details.** We leveraged the pre-trained MAV and fine-tuned it using a mixture of 60% MAV data and 40% search data. Fine-tuning was run for 20,000 steps, using a batch size of 512. Thus the model saw an order of magnitude more tokens during the MAV pre-training compared to the fine-tuning on internal search traces.

# 5. Experiments

## 5.1. Evaluation

Evaluating LLMs in general and reasoning specifically is a broad field (Chang et al., 2024), within which games have been established as an evaluation benchmark (Costarelli et al., 2024; Duan et al., 2024). We evaluate MAV language models in a *games league* which head-to-head match-ups, sampling combinations uniformly at random from a pool of MAV and baseline agents. We report *internal Elo* ratings (relative Elo only between members of the population) as

well as *external Elo* ratings where possible.

For chess evaluation, we rely on the state-of-the-art engine Stockfish at different playing strengths to estimate the widely used external Elo rating. This is done by calibrating the internal Elo with externally-reported ratings using a linear fit. As an additional, we include the Ext-BoN model (Ruoss et al., 2024a), which showed Grandmaster-level performance on chess with Transformers (see Appendix B for details). We use a set of Top Chess Engine Championship (TCEC) opening positions (tce) (see Table 5 in Appendix G). In each match-up between two agents, a specific opening is used, and agents swap seats to ensure each agent plays each opening both as black and as white. Every instance of Stockfish is run with 2 seconds of search time. We delineate further our evaluation setting together with the closest related works in Appendix B and Appendix C.

An overall comparison of the playing strength of different methods is shown in Table 1. Next, we perform a deeper dive into the performance of the different approaches.

## 5.2. Multi-Action-Value Results

In terms of chess playing strength, MAV reaches an external Elo of 2923 when using mean scoring and 2875 when using

*Table 1.* Performance comparison of different agents in Chess, Chess960, Connect Four, and Hex. In MAV–MCTS agents, $M$ refers to the number of simulations. The external Elos are estimated from the internal Elos achieved among agents.
**Important:** The external Elos of the Stockfish agents are anchored to the CCLR Blitz Elos. Therefore, the Elo estimates for the MAV–MCTS agents are not accurate as those agents would not be able to play at blitz time controls.

| Agent | Chess | | Chess960 | Connect Four | Hex |
| --- | --- | --- | --- | --- | --- |
| | Int. Elo | Ext. Elo | Int. Elo | Int. Elo | Int. Elo |
| Stockfish–L20 | 1990 | 3474 | 2053 | | |
| MAV–MCTS($M = 2000$) | 1707 | 3209 | 1602 | | |
| Stockfish–L19 | 1689 | 3191 | 1645 | | |
| Stockfish–L18 | 1655 | 3170 | 1604 | | |
| MAV–MCTS($M = 1000$) | 1633 | 3157 | 1561 | 499 | |
| Stockfish–L17 | 1604 | 3141 | 1569 | | |
| MAV–MCTS($M = 500$) | 1592 | 3131 | 1506 | 445 | 1048 |
| MAV–MCTS($M = 250$) | 1542 | 3088 | 1432 | 428 | 1002 |
| Stockfish–L15 | 1520 | 3069 | 1436 | | |
| MAV–MCTS($M = 100$) | 1435 | 2988 | 1377 | 342 | 884 |
| Ext-BoN (Ruoss et al., 2024a) | 1370 | 2926 | 1208 | | |
| MAV (mean scoring) | 1367 | 2923 | 1278 | | |
| MAV (max scoring) | 1317 | 2875 | 1213 | 98 | 713 |
| MAV–small (max scoring) | 1259 | 2820 | | | |
| Stockfish–L10 | 1225 | 2788 | 1141 | | |
| Stockfish–L5 | 808 | 2203 | 769 | | |
| Stockfish–L0 | 515 | 1320 | 533 | | |
| MCTS | 0 | 805 | 0 | 0 | 0 |

*Table 2.* Error rate analysis for MAV in chess positions. We consider positions drawn from Lichess puzzles, and from a dataset of randomly generated boards that never occur in the course of normal gameplay, and are highly out-of-distribution compared to the training data.

| | Lichess puzzles | OOD positions |
| --- | --- | --- |
| `%best_action` legal | 0.999 | 0.996 |
| `%top_all` precision | 0.999 | 0.997 |
| `%top_all` recall | 1.000 | 0.997 |
| `%top_all` F1 score | 0.999 | 0.995 |
| Updated FEN correct | 1.000 | 0.996 |

max scoring. It thus outperforms several strong baselines, including Stockfish–L10 and performs comparably with the Ext-BoN model by Ruoss et al. (2024a). For Chess960 MAV outperforms Ruoss et al. (2024a), perhaps due to being trained explicitly on Chess960 data.

In Table 2, we analyze the legal move rate, the precision and recall of the predicted `%top_all` moves, and the accuracy of predicting the next FEN state. The results demonstrate that MAV is able to reliably perform all of these actions.

**Generalization.** During the opening and endgame, human players often rely on memorized opening lines and endgame theory, while middlegame requires more calculation and intuition. In Appendix D, we report the same pattern in the games of MAV and a concrete position where the model displays a creative play in unseen position. Overall, 10%

of the positions played by MAV in evaluation appear in its training data, while between moves 20 and 50, virtually no position has been seen by MAV during training. These results show that in order to avoid losing games during middlegame, MAV is required to generalize.

### 5.3. External Search Results

For our external-MCTS agents, we tune hyper-parameters using a combination of manual and head-to-head comparisons and report the details in Appendix A. We also include a basic MCTS baseline, which refers to MCTS with a uniform prior and random rollouts run with 100 simulations.

We first analyze the performance of the scoring method for the value function when used within MAV–MCTS. We do this by running a large tournament between various models with the details, including results shown in Figure 9 in Appendix A. MAV–MCTS with mean scoring performs noticeably better in all cases. Hence, for the remainder of the external search experiments, we report results only for the mean scoring method.

We then run an even bigger tournament including Ext-BoN with the results shown in Table 1. The total number of games played in each respective tournament were 14689 for Chess, 4480 for Chess960, 2189 for Connect Four, and 6334 for Hex. In chess, MAV–MCTS with just 100 simulations achieves an internal Elo 68 higher than the searchless MAV and generally Elo performance improves logarithmically as a function of the number of simulations as illustrated in Figure 5 (right).

In the case of Chess960, where every game begins with a random position (among a preset 960 initial positions), we obtain comparable results to Chess. In the case of Connect Four, we notice that MAV–MCTS is particularly helpful in improving upon MAV, with all agents improving by at least 244 Elo. Similarly to Chess, the improvements consistently rise with added simulations, but here we notice a relatively smaller performance gain of MAV over a basic MCTS, and larger improvements of MAV–MCTS over MAV.

Since our final model was not adequately trained with state-tracking capabilities for Hex, in the Hex results only a game engine is used to determine legal actions, state transitions, and terminal states. In contrast to Connect Four, we notice a large improvement between basic MCTS and MAV in Hex, and improvements similar to chess from search. We suspect that this is due to the relative complexity of the games, but more research is needed to clarify the reason for these differences. It should be possible to support the engine-free logic in Hex similarly to Chess going forward. Across all games, external MCTS performs consistently better as the number of simulations is increased.

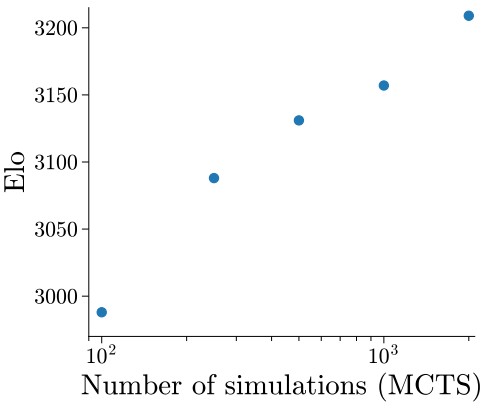 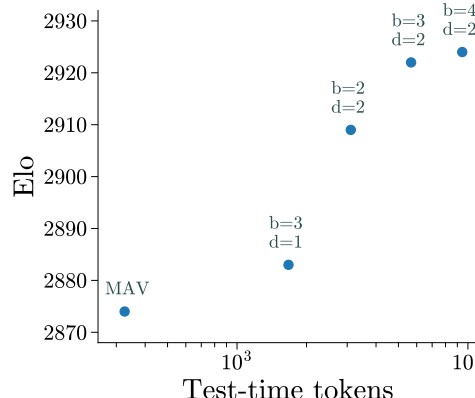

*Figure 5.* Chess Elo improves with more test-time compute. **Left:** External search results, varying the number of simulations / MAV model calls. **Right:** Internal search results, varying the breadth ($b$) and depth ($d$) of the minimax search performed in a single model call. The $x$-axis shows the number of tokens in the prompt+response.

## 5.4. Internal Search Results

For our internal search experiments, we vary the depth and breadth parameters of the minimax search, up to breadth 4 and depth 2. We map these different configurations into token counts by computing the average length of the prompt + response per configuration in our training data. Figure 5 (right) shows the performance of MAV–IS as the function of token counts. We can see that playing strength, measured by external Elo[3], increases with the search budget.

We also analyzed if the MAV–IS search produces the tree of the requested shape whenever possible, considering the available number of legal moves at each node. MAV–IS constructed an accurately shaped tree 99.6% of the time on a sample 5980 positions taken from existing games.

For an example of how internal search can improve playing strength, see Figure 4. In this example, the initial, MAV section, of the response predicts ♛**d6** as the best move. However, as the internal search continues, the model is able to find a better move ♛×**g6** after exploring the top-3 lines one step further. This points to the model's ability to self-correct, which is an important capability of LLMs that can reason (Kumar et al., 2024).

## 6. Discussions and Limitations

Despite promising results in the domain of perfect information board games indicating the potential of external and internal planning with LLMs, our initial study makes a number of assumptions that future work may need to

---

[3]For the internal search experiments, we stopped games once either side had reached a decisive, 1200+ centipawn advantage. This was done since the internal search training data did not include quickest wins, which would sometimes result in the model aimlessly moving pieces around in winning positions.

address – namely, the ability to acquire or generate large quantities of game play data, as well as the availability of reliable solvers or game engines that can be used to annotate this data in order to create an appropriate training curriculum for the model.

Another important limitation of our MAV models is that they have been trained exclusively on game data, and therefore do not possess the ability to communicate verbally using natural language. However, there should be no fundamental obstacles in achieving the same ability in potentially larger models that may also incorporate natural language data, as we train on the exact same architecture and tokenizer used in classical text-based LLMs.

It remains an open question how to design good value functions for general conversational task, and how to incorporate these value functions or other highly specialized knowledge in training such that the model can draw upon them flexibly at inference time, in a wide variety of conversational contexts.

## 7. Conclusions

This paper demonstrates the capacity of LLMs to learn strong value functions and act as a world model across multiple perfect information games. This enables their use in MCTS, where we observe significant performance gains of approximately +300 Elo points even with a fairly limited search budget. Going further, we find that training on search traces enables the model to learn an effective search procedure that can be executed via a single model call. This adds to the rapidly growing body of literature highlighting the promise of planning and reasoning with LLMs.

## Acknowledgements

We would like to thank Alexis Jacq, Yoram Bachrach, and Roma Patel for their contributions early on in the project, and GM Bogdan-Daniel Deac, GM Dharshan Kumaran, and Luka Rimanić for playing test games and providing feedback on the MAV model. We are grateful to Aliaksei Severyn, David Silver, Jessica Hamrick, Gabriel Dulac-Arnold, Been Kim, Simon Osindero, Matthew Lai, and Demis Hassabis for their support and advising on the project.

## Impact Statement

This paper presents work whose goal is to advance the field of Machine Learning. There are many potential societal consequences of our work, none which we feel must be specifically highlighted here.

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

# A. External MCTS

## A.1. Additional Plots and Results

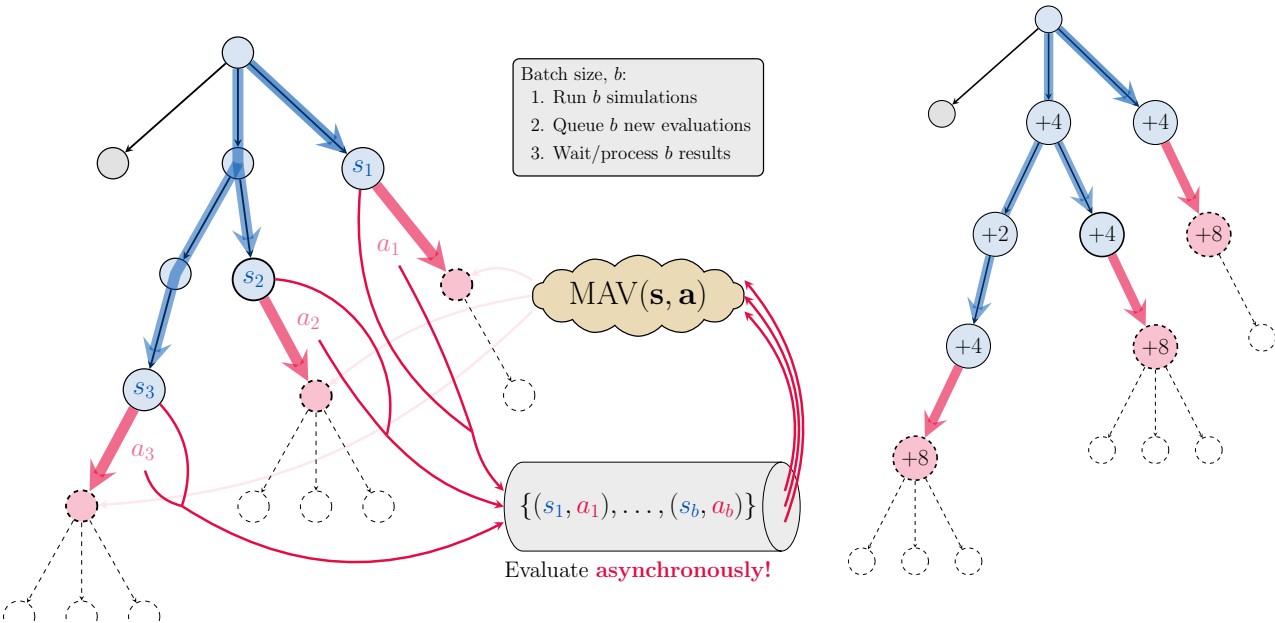

Figure 6. An overview of Async MCTS.

Figure 7. Dynamic virtual counts for $n_{\max} = 8$ and $n_{\min} = 2$. Virtual counts from separate simulations are added, as is the case in the $+4$ node with two children $(+4 = (+2) + (+2))$.

Figure 8. Async MCTS and Dynamic Virtual Counts building blocks.

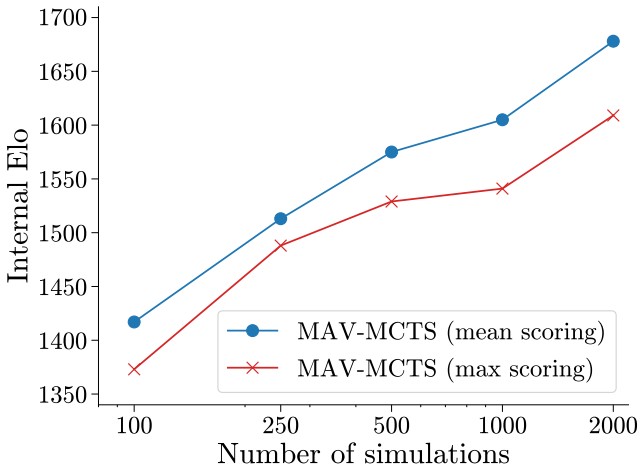

Figure 9. Performance of scoring methods in MCTS among a tournaments between all MCTS agents, various levels of Stockfish, and basic MCTS. The y-axis shows the internal Elo of each agent with basic MCTS set to 0 internal Elo.

## A.2. Performance Considerations and Async MCTS

Since modern large transformer models can take orders of magnitude longer for a single inference than an optimized deep convolutional network (such as the ones used in MuZero (Schrittwieser et al., 2020)), keeping the elapsed time for an MCTS search to a reasonable level can be a challenge. The inference time of a single model call dominates the time take for a

| Number of Simulations, $M$ | Average Elapsed Time per Search (seconds) |
|---|---|
| 100 | 39.82 |
| 250 | 114.12 |
| 500 | 187.10 |
| 1000 | 378.32 |
| 2000 | 729.23 |

*Table 3.* Average elapsed time taken per search by Async MCTS.

search, so we aim to keep the total calls low. Since the prior and value for a new node are both obtained for the MAV, there is at most one model call per simulation of MCTS. In the low simulation count regime, reusing the search trees is a simple optimization that reduces the number of server calls by 5-10% in our experiments.

Despite this lower number of inference calls, it can still be costly to make many MAV calls. Hence, we run an *asynchronous* version of MCTS that is conceptually very similar to Algorithm 1, but with the main loop changed. An overview is shown in Figure 6.

Parallel MCTS implementations usually rely on *virtual losses* (Mirsoleimani et al., 2017; Chaslot et al., 2008) to avoid multiple threads choosing the same simulation paths down due to deterministic action choices. Our Async MCTS introduces three new parameters:

**Batch size $b$.** The number of simulations, each leading to an evaluation (MAV call), that are done asynchronously at any given time.

**Timeout $t_0$.** The amount of time to wait for evaluations. After the timeout expires, any evaluations that have not returned are discarded.

**Virtual Count Value $n_c$.** The "virtual" value to temporarily add to a visit count $N(s, a)$ during the simulation to discourage all simulations in a batch from choosing the same path.

In Async MCTS, simulations are instead performed in batches. First, $b$ downward passes are performed serially to queue $b$ evaluations. The $b$ evaluations are performed in parallel and the MAV outputs are returned asynchronously.

The evaluations are then processed one-by-one as they return. For each queued evaluation that was successful (*i.e.,* did not time out), its values are then backpropagated up the tree (serially). The state-action pairs that are visited before the queuing of evaluations are assigned temporary *virtual counts*: their visit counts are temporary set to $N(s, a) = N(s, a) + n_c$. The visit counts and virtual counts are then subtracted upon the upward pass, i.e., $N(s, a) = N(s, a) - n_c$.

Note that, since the simulations and processing of evaluations (backpropagation) are both done serially, there is no need for complex concurrency handling primitives such as mutexes or semaphores. In practice, the implementation is based on Python's `concurrent.futures` module and re-uses most of the serial MCTS subroutines, changing only the main loop.

The average amount of time taken per search is shown in Table 1.

### A.3. Hyper-Parameter Tuning and Values

For our external-MCTS agents, we tune hyper-parameters using a combination of manual and head-to-head comparisons. We found that setting $\tau = 0.1$, $k = 5$, and $\varepsilon = 0.05$ worked best for the prior (Equation 1).

For Async MCTS we used a batch size $b = 16$. We found that lower batch sizes were always preferred in terms of performance. We used a timeout of $t_0 = 60$ seconds, and (only as a baseline) static virtual loss parameter $n_c = 10$. In head-to-head comparisons, we found that a tuned dynamic virtual count would generally outperform static virtual counts. We use the robust child strategy for final move selection (Chaslot et al., 2009) (choosing the action with the highest number of visits), using the total reward as a tie-breaker.

We used dynamic virtual count values $n_{\min} = 2$ in all cases, and $n_{\max}$ varying with the number of simulations: $(M, n_{\max}) \in \{(100, 8), (250, 8), (500, 8), (1000, 16), (2000, 32)\}$.

### A.4. External MCTS: Algorithm Details

In this section, we describe the terminology and pseudo-code for external MCTS.

**Notation:**

- $i \in \{0, 1\}$: denotes the active player; e.g., in chess, 0 for white, 1 for black

- $M$: number of MCTS simulations

- $s_t$: a string description of a state at step $t$; e.g., a FEN in chess

- $a_t$: an action at step $t$

- $T(s_{t-1}, a_{t-1}) = s_t$: a (deterministic) transition function

- $\boldsymbol{a}_t^L = (a_t^1, \ldots, a_t^{L(s_t)})$: a list of legal actions at step $t$; note that $\boldsymbol{a}_t^L$ is state dependent, but we omit dependency for the simplicity of notation

- $\mathcal{N}_0$: a root node; stores internal objects
    - an active player $\mathcal{N}_0(i) \in \{0, 1\}$
    - a string description of the initial state $\mathcal{N}_0(s) = s_0$
    - a list of legal actions $\mathcal{N}_0(\boldsymbol{a}^L) = \boldsymbol{a}_0^L$

- $Q^{(i)}(s_t, a_t)$: a state-action value from the perspective of player $i$ in state $s_t$ given action $a_t$

- $\boldsymbol{Q}^{(i)}(s_t, \boldsymbol{a}_t^L) = (Q^{(i)}(s_t, a_t^1), \ldots, Q^{(i)}(s_t, a_t^{L(s_t)}))$: a state-action value from the perspective of player $i$ in state $s_t$ for all legal moves $\boldsymbol{a}_t^L$

- $P(s_t, a_t) = \mathbb{P}(a_t|s_t)$: a prior probability of action $a_t$ in state $s_t$

- $\boldsymbol{P}(s_t, \boldsymbol{a}_t^L) = (P(s_t, a_t^1), \ldots, P(s_t, a_t^{L(s_t)}))$: a prior distribution, i.e., a distribution over all legal moves $\boldsymbol{a}_t^L$ in state $s_t$

- $\mathcal{N}_t$: a node at step $t > 0$; stores internal objects
    - an active player $\mathcal{N}_t(i) = 1 - \mathcal{N}_{t-1}(i)$
    - a string description of a state $\mathcal{N}_t(s) \in \{s_t = T(s_{t-1}, a_{t-1}), \text{NONE}\}$
    - a list of legal actions $\mathcal{N}_t(\boldsymbol{a}^L) \in \{\boldsymbol{a}_t^L, \text{NONE}\}$ in state $s_t$
    - a cumulative value $\mathcal{N}_t(V^{(i)}(s)) \geq 0$ from child's perspective $i = \mathcal{N}_t(i)$
    - a visit count $\mathcal{N}_t(N(s, a)) = N(s_{t-1}, a_{t-1}) \geq 0$ given the parent state $s_{t-1}$ and action $a_{t-1}$
    - a prior probability $\mathcal{N}_t(P(s, a)) = P(s_{t-1}, a_{t-1}) \geq 0$ given the parent state $s_{t-1}$ and action $a_{t-1}$

- MAV: a model that supports a multi-action value interface
    - $(\boldsymbol{a}_0^L, \boldsymbol{Q}^{(i)}(s_0, \boldsymbol{a}_0^L)) = \text{MAV}(s_0)$: a state evaluator; returns a list of legal moves $\boldsymbol{a}_0^L$ and associated state-action values $\boldsymbol{Q}^{(i)}(s_0, \boldsymbol{a}_0^L)$ for the state $s_0$ in the root node $\mathcal{N}_0$; notice that the state evaluator is only necessary for the root node
    - $(s_t, \boldsymbol{a}_t^L, \boldsymbol{Q}^{(i)}(s_t, \boldsymbol{a}_t^L)) = \text{MAV}(s_{t-1}, a_{t-1})$: a state-action evaluator; for a given state-action parent tuple $(s_{t-1}, a_{t-1})$ returns the child state $s_t = T(s_{t-1}, a_{t-1})$, a list of child's legal actions $\boldsymbol{a}_t^L$ and the associated state-action values $\boldsymbol{Q}^{(i)}(s_t, \boldsymbol{a}_t^L)$ from child's perspective $i = \mathcal{N}_t(i)$

## B. External Engines and Game Data

In this section, we describe the details on the data generation for each game, while Table 4 reports training data statistics.

---

**Algorithm 2** EXPAND($\mathcal{N}_{t-1}, \boldsymbol{P}$)

---

1: **Input:** Parent node $\mathcal{N}_{t-1}$, prior $\boldsymbol{P}$
2: **Output:** NONE
3: **for** $a_{t-1} \in \mathcal{N}_{t-1}(\boldsymbol{a}^L)$ **do**
4:     Create an empty child node $\mathcal{N}_t$
5:     Connect parent node $\mathcal{N}_{t-1}$ and a child node $\mathcal{N}_t$ with edge $(\mathcal{N}_{t-1}, a_{t-1})$
6:     Initialize child node:
7:         $\mathcal{N}_t(i) \leftarrow 1 - \mathcal{N}_{t-1}(i)$
8:         $\mathcal{N}_t(s) \leftarrow$ NONE
9:         $\mathcal{N}_t(\boldsymbol{a}^L) \leftarrow$ NONE
10:         $\mathcal{N}_t(V^{(i)}(s)) \leftarrow 0$
11:         $s \leftarrow \mathcal{N}_{t-1}(s), a \leftarrow a_{t-1}$
12:         $\mathcal{N}_t(N(s,a)) \leftarrow 0$
13:         $\mathcal{N}_t(P(s,a)) \leftarrow P(s,a)$
14: **end for**
15: **return**

---

**Algorithm 3** SIMULATION($\mathcal{N}_{t-1}$, MAV, $k$)

---

1: **Input:** Parent node $\mathcal{N}_{t-1}$, multi-action value model MAV, top $k$ values
2: **Output:** NONE
3: $s_{t-1} \leftarrow \mathcal{N}_{t-1}(s)$
4: $a_{t-1} \leftarrow$ SELECT($\mathcal{N}_{t-1}$)
5: $\mathcal{N}_t \leftarrow$ MOVE-TO-CHILD($\mathcal{N}_{t-1}, a_{t-1}$)
6: **if** $\mathcal{N}_t(s)$ is NONE **then**
7:     $(s_t, \boldsymbol{a}_t^L, \boldsymbol{Q}^{(i)}(s_t, \boldsymbol{a}_t^L)) = $ MAV$(s_{t-1}, a_{t-1})$ from child's perspective $i = \mathcal{N}_t(i)$
8:     $\boldsymbol{P}(s_t, \boldsymbol{a}_t^L; \boldsymbol{Q}^{(i)}, k) \leftarrow$ Equation (1)
9:     EXPAND($\mathcal{N}_t, \boldsymbol{P}$)
10:     $\mathcal{N}_t(s) \leftarrow s_t$
11:     $\mathcal{N}_t(\boldsymbol{a}^L) \leftarrow \boldsymbol{a}_t^L$
12:     $Q^* = \max_{a_t \in \boldsymbol{a}_t^L} Q^{(i)}(s_t, a_t)$
13:     BACKPROPAGATE($\mathcal{N}_t, Q^*$)
14: **else**
15:     SIMULATION($\mathcal{N}_t$, MAV)
16: **end if**
17: **return**

---

**Stockfish** For chess and Chess960 we used the open source engine Stockfish 16 (Source Code Avalible on Github) for both annotations and as opponents in our internal Elo calculations. We compiled all Stockfish builds from source, using clang++, with the following compile flags `-O3 -DIS_64BIT -DUSE_PTHREADS -mprefer-vector-width=128` for all builds. We ran Stockfish over a wide array of hardware using two different configurations, for older hardware we added `-DUSE_POPCNT -DUSE_SSSE3 -DUSE_SSE41 -march=westmere -msse4.2` and for newer hardware we added `-DUSE_AVX2 -DUSE_PEXT -march=haswell`. These builds usually give us between 350k-700k nodes per second for single threaded runs of `stockfish -- bench 4096 1 130000 default nodes`.

**Ext-BoN** (Ruoss et al., 2024a) showed that a small (270M-parameter) transformer can play Blitz chess on Lichess at an Elo of 2895 against humans (i.e., Grandmaster level). They created a large-scale dataset of 10 million chess games with legal move and value annotations (15 billion data points) provided by Stockfish 16, the state-of-the-art chess engine. They then trained Transformers via supervised learning to predict action-values, i.e., no explicit search, and showed that their models achieve highly non-trivial generalization (e.g., solving challenging chess puzzles). They thus showed that a remarkably good approximation of Stockfish's search-based algorithm can be distilled into large-scale transformers via supervised learning, but that perfect distillation is still beyond reach.

**Algorithm 4** SELECT($\mathcal{N}_{t-1}$)

1: **Input:** Parent node $\mathcal{N}_{t-1}$
2: **Output:** Action $a^*$ determined by PUCT
3: Access child nodes values from parent's perspective and compute PUCT; if tied select randomly
4: $i = \mathcal{N}_{t-1}(i)$
5: $s = \mathcal{N}_{t-1}(s)$
6: $s' = \mathcal{N}_t(s)$
7: $\boldsymbol{a}^L = \mathcal{N}_{t-1}(\boldsymbol{a}^L)$
8: $N(s) = \sum_{a \in \boldsymbol{a}^L} \mathcal{N}_t(N(s,a))$
9: $A^* \leftarrow \arg\max_{a \in \boldsymbol{a}^L} \left( \frac{\mathcal{N}_t(V^{(i)}(s'))}{\mathcal{N}_t(N(s,a))} + c_{\text{puct}} \cdot \mathcal{N}_t(P(s,a)) \cdot \frac{\sqrt{N(s)}}{1 + \mathcal{N}_t(N(s,a))} \right)$
10: **if** $|A^*| > 1$ **then**
11: $\quad a^* \leftarrow \text{RAND}(A^*)$
12: **else**
13: $\quad a^* \leftarrow A^*$
14: **end if**
15: **return** $a^*$

**Algorithm 5** BACKPROPAGATE($\mathcal{N}_t, Q$)

1: **Input:** Child node $\mathcal{N}_t$, state-action value $Q$
2: **Output:** NONE
3: **while** $\mathcal{N}_t \neq \mathcal{N}_0$ **do**
4: $\quad \mathcal{N}_t(V^{(i)}(s)) \leftarrow \mathcal{N}_t(V^{(i)}(s)) + Q$
5: $\quad s = \mathcal{N}_{t-1}(s), a = \mathcal{N}_{t-1}(a)$
6: $\quad \mathcal{N}_t(N(s,a)) \leftarrow \mathcal{N}_t(N(s,a)) + 1$
7: $\quad \mathcal{N}_t \leftarrow \text{MOVE-TO-PARENT}(\mathcal{N}_t)$
8: **end while**
9: **return**

**Algorithm 6** MOVE-TO-CHILD($\mathcal{N}_{t-1}, a_{t-1}$)

1: **Input:** Parent node $\mathcal{N}_{t-1}$, parent action $a_{t-1}$
2: **Output:** Child node $\mathcal{N}_t$
3: Move from a parent node $\mathcal{N}_{t-1}$ to a child node $\mathcal{N}_t$ via the edge $(\mathcal{N}_{t-1}, a_{t-1})$
4: **return** $\mathcal{N}_t$

**Algorithm 7** MOVE-TO-PARENT($\mathcal{N}_t$)

1: **Input:** Child node $\mathcal{N}_t$
2: **Output:** Parent node $\mathcal{N}_{t-1}$
3: Move from a child node $\mathcal{N}_t$ to a parent node $\mathcal{N}_{t-1}$
4: **return** $\mathcal{N}_{t-1}$

**Chess/Chess960 Game Training Data**   Chess positions were drawn from two sources: Lichess and TCEC. TCEC data consisted of all 34,291 standard chess games played in seasons 1-25, from which 4,204,025 unique positions were extracted. Lichess standard rated chess games were drawn from the months October 2014, March 2017, and January 2020 for a total of 58,705,532 games. All 19,758,457 Chess960 games played on Lichess from August 2013 to June 2024 were included. All positions were annotated with Stockfish 16 for a duration of 1 second, with the exception of those drawn from TCEC and Lichess October 2014 games, where 15 seconds of analysis time was used.

**Hex Training Data**   12,279,645 Hex games were generated by self-play with neurobenzene (Gao & Pawlewicz), run with search time set to 10 seconds. In 10% of positions, a random move was chosen instead to encourage exploration of an otherwise narrow state space distribution.

---

**Algorithm 8** FINALMOVESELECTION($\mathcal{N}_0$)

---

1: **Input:** Root node $\mathcal{N}_0$
2: **Output:** Best final action $a^*$ given search statistics
3: Determine most visited child; if tied find maximum estimated value
4: $s \leftarrow \mathcal{N}_0(s)$
5: $\boldsymbol{a}^L \leftarrow \mathcal{N}_0(\boldsymbol{a}^L)$
6: $A^* \leftarrow \arg\max_{a \in \boldsymbol{a}^L} \mathcal{N}_1(N(s,a))$
7: **if** $|A^*| > 1$ **then**
8:     Access child nodes values from root's perspective and find maximum estimated value
9:     $i \leftarrow \mathcal{N}_0(i)$
10:     $s' \leftarrow \mathcal{N}_1(s)$
11:     $a^* \leftarrow \arg\max_{a \in A^*} \frac{\mathcal{N}_1\left(V^{(i)}(s')\right)}{\mathcal{N}_1(N(s,a))}$
12: **else**
13:     $a^* \leftarrow A^*$
14: **end if**
15: **return** $a^*$

---

*Table 4.* Multi-action-value (MAV) model training data statistics.

| Game | Positions | Action values |
|---|---|---|
| Chess | 3.1B | 54.3B |
| Chess960 | 1.2B | 20.9B |
| Connect Four | 21.8M | 110.5M |
| Hex | 125.6M | 537.0M |

**Connect Four Training Data** 2,458,293 Connect Four games were generated using an $\varepsilon$-greedy strategy with $\varepsilon = 0.4$. Greedy actions were chosen uniformly among all moves that yielded the best attainable minimax value, computed using the Fhourstones solver (Tromp).

## C. Games League

We conduct planning and reasoning experiments in games, similar to *GameBench* (Costarelli et al., 2024) and *GTBench* (Duan et al., 2024). Games present an opportunity to develop and test approaches in a controllable and safe environment, with ground truth that is readily verifiable through existing game models. We use *OpenSpiel* (Lanctot et al., 2019) as the game engine to drive our evaluations.

Decision-making with language agents inseparably fuses a *prompting strategy*, the *language model*, and the *response interpretation*. The language model essentially interfaces with a second modality – *the game*, for which prompting encodes the state observation as language input, and response interpretation maps from language output back to action. The language model performs the response computation in language space. Any given triplet can be evaluated on specific states, or by playing against another triplet or against benchmark bots (e.g., Random, search or reinforcement learning based agents) that operate directly on the game representation.

We evaluate agents *offline*, through fixed datasets of states, and *online*, in pairwise matches against reference opponents. Metrics such as action-choice quantiles and action-value regret that we compute for each dataset across agents are directly comparable, and exhibit different *skill dimensions* (tasks), such as doing well in opening, mid-game or endgame positions. In contrast, online matches directly measure relative playing strength as wins/draws/losses, and may traverse different distributions of states for each match-up of opponents. While the eventual objective is to increase playing strength across a wide range of opponents, the offline evaluation on datasets helps characterise each policy.

A tournament of multiple opponents results in square agent-vs-agent matrices indicating the number of wins and draws (losses inferable from transposed of wins) for each pair. Elo (Elo, 1967) finds the best single-dimensional *rating* for each contender, such that their mutual win probabilities are best explained by a ratio of their exponentiated rating. This implies a *ranking*, i.e., an ordinal preference over each of the contenders. We report average winrates (treating draws as half wins) for

matches, and Elo for tournaments.

## D. MAV Generalization

During the opening and endgame, human players often rely on memorized theory, while middlegame requires more calculation and intuition. In Figure 10, we observe that 10% of the positions played by MAV in evaluation appear in its training data, while between moves 20 and 50, virtually no position has been seen by MAV during training. These results show that in order to avoid losing games during middlegame, MAV is required to generalize.

Finally, we show an example of MAV's generalization capabilities into a position which does not appear in the training data. In the position shown in Figure 11, MAV has found a creative move, g5, which is the best move according to Stockfish. The move allows the White knight on f3 to capture the pawn but results in a long-term advantage for Black.

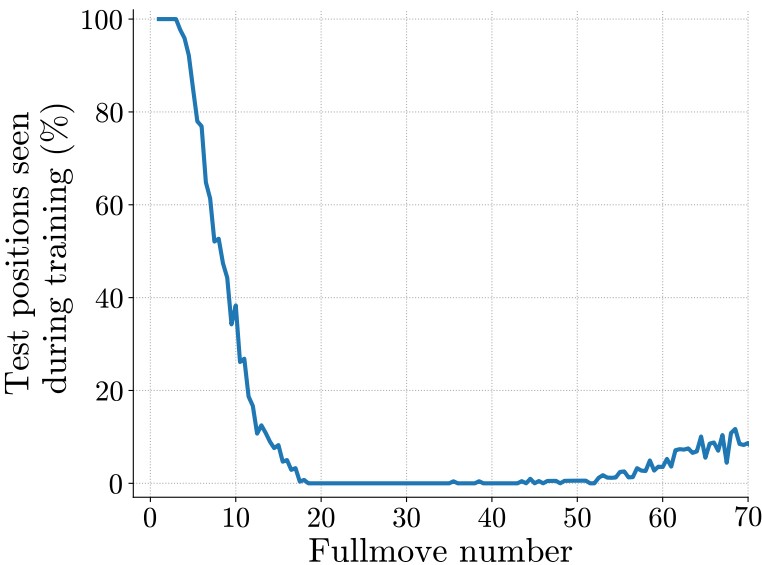

*Figure 10.* Percentage of chess positions encountered during evaluation that also appear in the MAV training data. Between moves 20 and 50, almost all of the encountered positions are unseen, requiring the model to generalize.

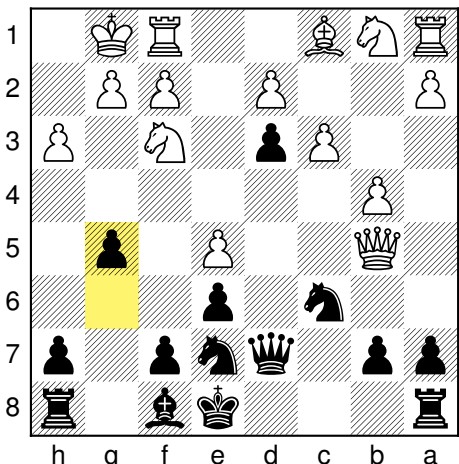

*Figure 11.* Example position where MAV plays a creative pawn sacrifice, g5 (best move according to Stockfish), to gain a long-term advantage.

**Out-of-Distribution test set positions** We programmatically generate out-of-distribution (OOD) positions by randomly sampling the number and type of pieces on the board while ensuring compliance with several rules such as: kings cannot be in check simultaneously, or the position is not a checkmate. Examples of OOD positions are shown in Figure 12. Notice that some positions are not reachable via a legal sequence of moves from the initial chess position, which showcases that MAV generalizes outside of the human games it was trained on by achieving near 100% accuracy as reported in Table 2.

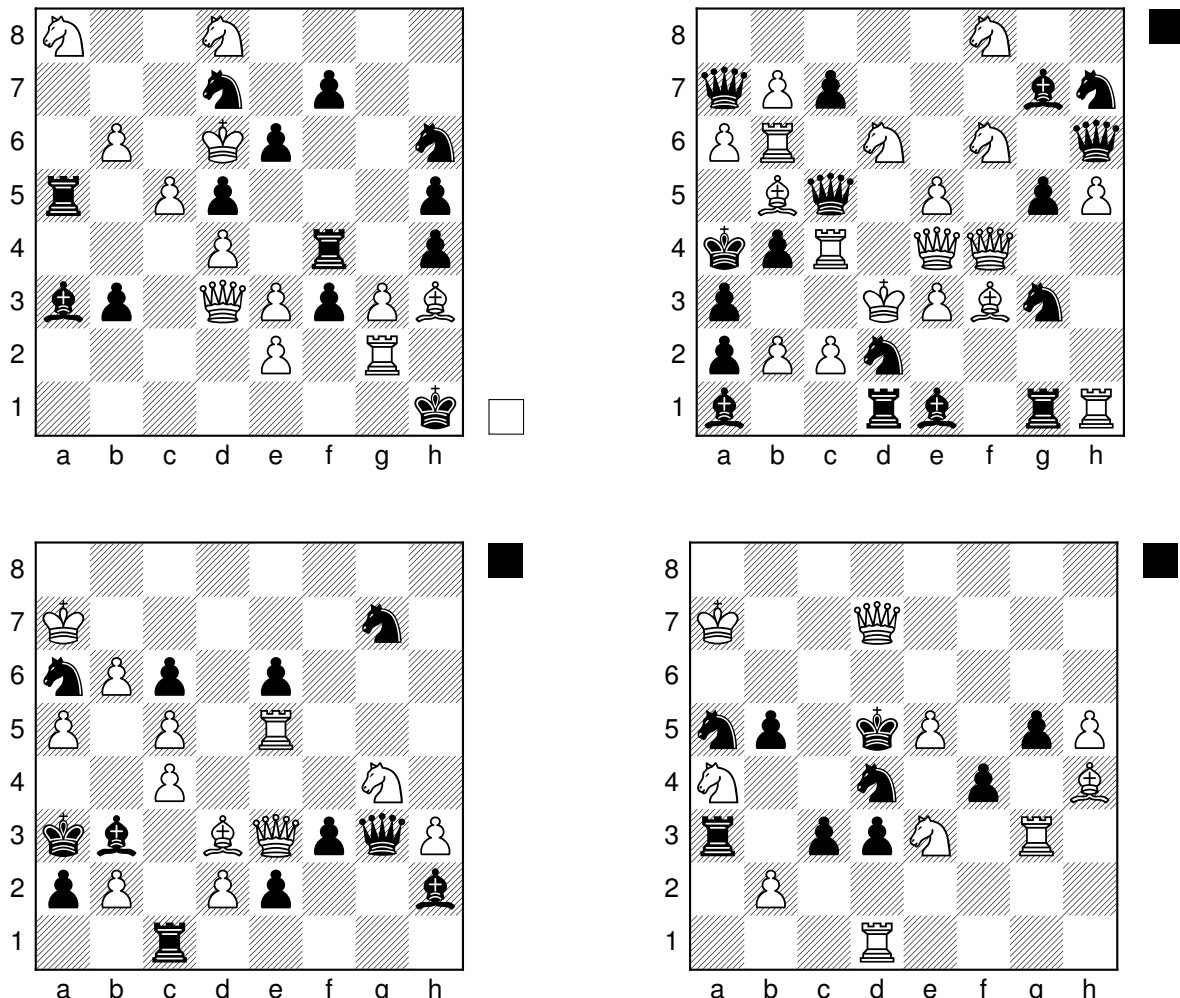

*Figure 12.* A sample from out-of-distribution set to test the legal move rate and state tracking accuracy of MAV. The squares next to the diagrams indicate active players.

# E. Related Work

Planning and reasoning with language models is both an open challenge and an active area of research. Given how widely relevant planning is as a capability to support decision making in complex, dynamic, and open-ended domains (Rasal & Hauer, 2024), there is, perhaps unsurprisingly, a correspondingly wide variety of planning approaches that have been explored to date, mapping onto these unmet needs. The remainder of this section first outlines general reasoning benchmarks, and then goes into game benchmarks in particular. Next, we outline adjacent works of in-context reasoning, which shares with internal search that plans are expanded by the language model itself, and reasoning as search, which sketches a broader scope around approaches similar to external search. Finally, we give pointers to specific works as chess, which takes a prominent role in our evaluation.

**Benchmarking reasoning**    Reasoning tasks can be divided into tasks of common sense reasoning (Zhao et al., 2024; Cui et al., 2024), structured reasoning (Wang et al., 2024b), mathematical reasoning (Shao et al., 2024b; Wu et al., 2024c; Chen et al., 2024a; Gu et al., 2024; Wang et al., 2024d; Zhang et al., 2024c; Tang et al., 2024b), algorithmic reasoning (Markeeva et al., 2024), symbolic reasoning (Fang et al., 2024), causal reasoning (Jin et al., 2024), temporal reasoning (Xiong et al., 2024), meta-reasoning (Zeng et al., 2024) and strategic reasoning (Zhang et al., 2024d). Multimodal reasoning may span any subset of these categories, with the added complexity of incorporating cross-modal tasks (Chia et al., 2024; Kil et al., 2024; Xie et al., 2024a; Haresh et al., 2024; Shao et al., 2024a; Shi et al., 2024). There are also multilingual benchmarks (Bang et al., 2023; Wang et al., 2024c), aimed at evaluating model reasoning across languages. Large language models can approach these reasoning tasks directly, or with help of external tools (Paranjape et al., 2023; Lu et al., 2024) or planners and solvers (Dagan et al., 2023; Borazjanizadeh & Piantadosi, 2024), and their performance can be evaluated in typical scenarios or *puzzles* (Giadikiaroglou et al., 2024; Li et al., 2024b), which tend to be more complex and more out-of-distribution. These evaluations can either be general or more domain-specific (Nori et al., 2024; Guo et al., 2024). There have also been efforts aimed at evaluating the utility of language models as world models, from the perspective of decision making and planning (Yang et al., 2024a).

**Reasoning in games**    Strategic reasoning (Gandhi et al., 2023), which naturally arises in game play, is particularly challenging due to a dynamic environment and the uncertainty in predicting the actions of the adversary. It is goal-oriented, interactive, adaptive, and requires a level of predictive ability. Outside of games, this type of reasoning may be valuable in economic (Horton, 2023; Li et al., 2023) and social (Gandhi et al., 2024a; Ziems et al., 2024) applications. The focus in this paper is primarily on strategic planning in games, which have been suggested as a an important set of benchmarks for developing and evaluating strategic reasoning (Tsai et al., 2023a; Duan et al., 2024; Costarelli et al., 2024) in language models, not unlike their historical role in the development of AI previously (McCarthy, 1990; Schaul et al., 2011; Ensmenger, 2012), and sequential decision making in particular. Three particular game benchmarks that have been concurrently developed with this paper are GameBench (Costarelli et al., 2024), GTBench (Duan et al., 2024), and LMAct (Ruoss et al., 2024b). All three adopt the same agentification of language models for games: an agent comprises a prompting strategy that generates a prompt from a game state, an LLM that generates the response, and an interpretation of the textual response as a game action. The resulting agent can be evaluated by playing matches in specific games, like tic-tac-toe, either against benchmark RL agents or other language model agents. This makes it possible to draw conclusions on strategic reasoning abilities from the pairwise winrates or rankings derived from Elo-rating the tournament results. Our specific evaluation is delineated in the experimental section of this paper.

**Reasoning *in-context***    It has been demonstrated that large language models encode a certain level of intrinsic reasoning ability in the natural language space, especially when prompted to do so (Plaat et al., 2024), which can be solicited by chain-of-thought (CoT) techniques (Kojima et al., 2022; Wei et al., 2022; Shi et al., 2022; Zhang et al., 2023; Wang et al., 2024e), least-to-most prompting (Zhou et al., 2022), plan-and-solve prompting (Wang et al., 2023), self-ask structured prompting (Press et al., 2022), multi-step problem decomposition and plan execution (Nguyen & Shareghi, 2024), few-shot prompting (Brown et al., 2020; Gao et al., 2020; Gramopadhye et al., 2024), and many-shot prompting (Agarwal et al., 2024), among others. Yet, CoT methods are not always reliable, may express inconsistent reasoning steps or traces that do not actually correspond to how the decisions are reached (Ye & Durrett, 2022; Turpin et al., 2024), may not be robust to typographical errors (Gan et al., 2024), and may not be the best approach for planning in more complex domains, though it has recently been shown that they can in principle handle e.g., dynamic programming tasks (Feng et al., 2024a). Without a world model capable of tracking state transitions, it is hard to evaluate the long-term consequences of earlier decisions (Hao et al., 2023). These emerging capabilities are heavily influenced by the data itself, as intermediate reasoning steps are only helpful in case of locally structured data, consisting of local clusters of variables with strong mutual influence (Prystawski et al., 2024). CoT rationales can be optimised through task-oriented feedback (Lee et al., 2024). Informative thought templates can be distilled from and reused across a number of different tasks, and compiled in a buffer of thoughts (Yang et al., 2024b). Diversity in CoT generations matters, and generating a larger number of diverse reasoning chains can lead to better performance (Zhang et al., 2022). Reasoning traces can be interleaved with actions, as in ReAct (Yao et al., 2022). Self-consistency techniques present another avenue for further improving model performance (Wang et al., 2022). To be useful, CoT prompts may need to be specific to their problem class (Stechly et al., 2024). Synthetic data generation can be used to improve the quality and the coverage of demonstrations (Shao et al., 2023). CoT has also been considered in the multimodal context (Lu et al., 2022; Mitra et al., 2024) and whiteboard-of-thought was recently shown to be helpful, in encouraging reasoning across modalities (Menon et al., 2024). Language models are capable of improving their performance on some types of problems merely by the extra computation granted by the additional tokens, even when those are filter

tokens rather than structured or interpretable CoT (Pfau et al., 2024). It is possible to consider instead generating *internal thoughts*, as rationales at the token level for predicting future tokens, demonstrated by the Quiet-STAR method (Zelikman et al., 2024). This method belongs to a class of approaches that utilise self-improvement or self-play for boosting the planning and reasoning ability in language models, which has become a widely used approach (Zelikman et al., 2022; Singh et al., 2024; Hosseini et al., 2024; Aksitov et al., 2023; Chen et al., 2024c; Tian et al., 2024; Saab et al., 2024). Self-improvement approaches often rely on the intrinsic ability for self-correction in LLMs, yet self-correcting reasoning without explicit external feedback remains difficult (Huang et al., 2024). Given that CoT reasoning is ultimately *greedy* and may have problems when there are many options worth exploring (Saparov & He, 2022), this was a motivation for exploring alternative, more structured search methods. Dynamic prompt-conditional strategy selection may help identify the best strategic approach to each particular problem (Parekh et al., 2024). Multi-agent approaches can be used to further develop this particular point, decoupling strategic reasoning and more specific step-wise planning (Wang et al., 2024a). Planning can also be done over associated knowledge graphs (Chen et al., 2024b), and graph learning may help with the overall planning capabilities in the more general case (Wu et al., 2024b). Finally, some problems may lend themselves more naturally to backward planning, potentially necessitating a custom approach (Ren et al., 2024).

**Reasoning as search**  Testing LLMs on especially sequential decision-making problems requires *agentification*, with a variety of proposed approaches to use LLMs as or within game agents (Tsai et al., 2023b; Hu et al., 2024b; Wu et al., 2024a; Hu et al., 2024c). MCTS-based planning, and self-play, have been also been explored in this context (Rebstock et al., 2024; Putta et al., 2024; Light et al., 2024). Rather than relying on individual reasoning chains, it is possible to instead construct a tree of thoughts (Long, 2023; Yao et al., 2024) a graph of thoughts (Besta et al., 2024), or a diagram of thoughts (Zhang et al., 2024e). There has also been recent interest in utilizing MCTS with LLMs for complex reasoning tasks (Zhang et al., 2024a). The everything-of-thoughts (XoT) (Ding et al., 2023) approach utilizes MCTS for incorporating external domain knowledge into thoughts, as does Tree-of-Traversals (Markowitz et al., 2024), enabling language models to effectively reason over facts encoded in knowledge graphs. Tree search has also proved valuable in completing realistic web tasks (Koh et al., 2024), as well as code generation (Li et al., 2024a), mathematical reasoning (Jiang et al., 2024), and tabular data handling (Ji et al., 2024). While most approaches rely on the prompted LLM itself acting as a value function, therefore bounding the utility of search by the quality of that value function approximation (Chen et al., 2024d), and searching at low depths, it is also possible to use tree search with a learned value function in guiding LLM decoding, as was demonstrated in TS-LLM (Feng et al., 2024b). Language Agent Tree Seatch (LATS) (Zhou et al., 2024) incorporates MCTS along with self-reflection and memory, achieving strong performance on programming tasks. MCTS has also been used to improve LLM reasoning ability through iterative preference optimization (Xie et al., 2024b). When applied in the language generation space, MCTS can be seen as a likelihood-based search on the direct preference optimization (DPO) policy (Rafailov et al., 2024). MCTS can also be used to strategically plan over the the problem-solving trajectories for zero-shot in-context learning (Tang et al., 2024a). Self-Rewarding Tree Search based on MCTS has been used for improving domain-specific retrieval-augmented generation (Hu et al., 2024a). In terms of other search strategies, *Searchformer* has recently been proposed (Lehnert et al., 2024), a Transformer model that was trained to mimic the dynamics of the $A^*$ algorithm, showing good performance on Sokoban puzzles and in maze navigation. Algorithm distillation has recently been proposed as a method for distilling reinforcement learning algorithms into neural networks, facilitating in-context reinforcement learning with language models (Laskin et al., 2022).

**Reasoning in chess**  As our paper explores the application of planning with language models to chess, which has been a separate topic of interest, here we also briefly review the current state of art and the approaches and evaluations relevant to chess in particular. Chess-GPT (Feng et al., 2023) was a custom model, integrating chess playing and language ability, that was shown to offer notable improvements in board state tracking and playing strength over a number of LLM baselines. More recently, it was shown to be possible to train Transformer models to Grandmaster-level strength in chess (Ruoss et al., 2024a), without search, looking just a single move ahead. There have been recent advances involving diffusion models as well (Anonymous, 2024). By training specialized Transformer models for chess, it has been shown that the models may demonstrate *transcendence* (Zhang et al., 2024b), at least at lower playing strengths – where the resulting model may outperform any of the human players from the training set, as the model learns a policy that effectively averages out some of the mistakes human players make. This is in line with the findings showing that *weak-to-strong supervision* is possible (Burns et al., 2023). It is also possible to develop *skill-compatible* models (Hamade et al., 2024), as skill compatibility is ultimately distinct from playing strength. There is also an interest in developing models that can act as chess coaches (Menon, 2023). There is evidence that chess-playing Transformers can learn and internalize *look-ahead* in relation to future optimal moves (Jenner et al., 2024). Hybrid convolutional transformer networks have been identified as a

promising avenue, yielding performance improvements (Czech et al., 2024). An in-depth analysis of chess playing strength of the most prominent language models (Acher, 2023) had identified *gpt-3.5-turbo-instruct* as the strongest model at the time, capable of completing longer games and winning against non-trivial Stockfish difficulty levels, yet still failing to finish a non-negligible percentage of games due to illegal moves.

## F. Featured Games

### F.1. Chess

We start games from the opening book described in Table 5. A 📚 symbol indicates the last move loaded from the opening book after which the agents continue the game. Once the game is finished, the agents reverse the colors and play the game with the same opening position. A square next to the diagrams represents the side to play. All games, except for Game 0 and Game 1, use Stockfish with 2 seconds of thinking time per move. Further details can be found in Section "External Engines and Game Data". We use MAV–MCTS to represent the external search MAV with mean scoring, MAV–MCTS - max scoring is self-explanatory, and MAV–IS represents the internal search MAV.



## Game 0
**A win against human player**
**Long-term piece activity**


**Date: 2024-07-24**
**White: Luka Rimanić, 1990 FIDE Elo**
**Black: MAV - max scoring**
**Result: 0 − 1**
**Online PGN: https://lichess.org/eU15TjI0**
📚 1. e4 e6 2. d4 d5 3. Nc3 dxe4 4. Nxe4 Nf6 5. Bg5 Be7 6. Bxf6 gxf6 7. g4

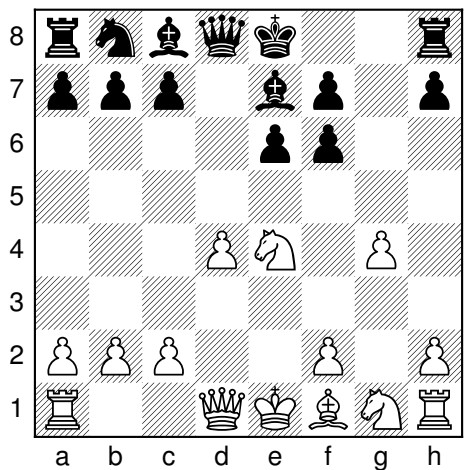

7...h5 **MAV sacrifices a pawn for the long-term piece activity.** 8. gxh5 f5 9. Ng3 c5 10. Nf3 Nc6 11. c3 cxd4 12. Nxd4 Nxd4 13. Qxd4 Qxd4 **MAV exchanges queens, but the bishop pair on the open board and rooks' activity secure the advantage for Black.** 14. cxd4 Bd7 15. O-O-O Bc6 16. Rg1 Rc8 17. Kb1 f4 18. Ne2 Be4+ 19. Ka1 Bf3 20. Rb1 Rd8 21. Nxf4 Rxd4 22. Bb5+ Kf8 23. Ne2 Rd2 24. Rbc1 Rxh5 **MAV regains the sacrificed pawn with interest after 17 moves; despite the even material, White is losing.** 25. Rc8+ Rd8 26. Rxd8+ Bxd8 27. Nc3 Rxh2 28. Rf1 Bb6 29. Nd1 Ke7 30. a3 f5 31. Kb1 Kf6 32. Kc2 a6 33. Bc4 Bd4 34. b4 b5 35. Bd3 e5 36. Kd2

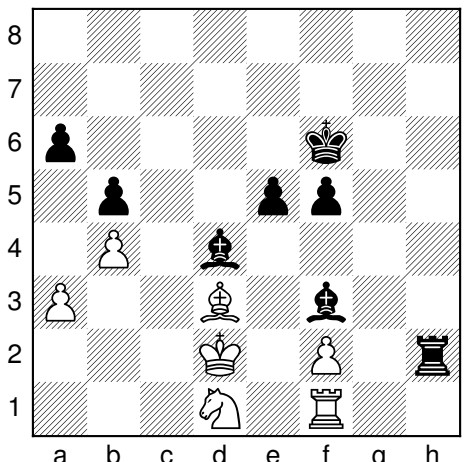

36...Bxd1 **MAV is not afraid to enter the endgame with opposite-colored bishops.** 37. Kxd1 e4 38. Be2 Bxf2 39. a4 bxa4 40. Bxa6 a3 41. Bc4 Be3 42. b5 Rd2+ 43. Ke1 Rb2 44. b6 Bxb6 45. Kd1 Ke5 46. Ke1 Be3 47. Kd1 f4 48. Ke1 f3 49. Kd1 Kd4 50. Bf7 Kc3 51. Ba2 Rxa2 52. Rxf3 exf3 53. Ke1 Ra1# 0 − 1



## Game 1
**The first win against Stockfish**
**Unclear long-term attack**


**Date: 2024-05-31**
**White: MAV–MCTS, development version**
**Black: Unoptimized Stockfish (50K-90K nps)**
**Result: 1 − 0**
**Online PGN: https://lichess.org/gVNnuSWn**
1. d4 f5 2. Bg5 g6 📚 3. e3 Bg7 4. Qd2 h6 5. Bh4 c5 6. f4 Qb6

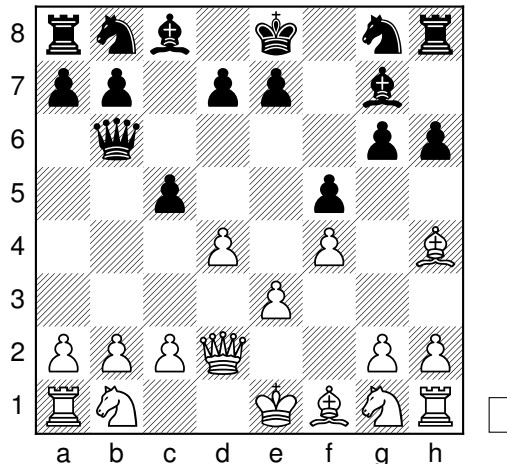

7. Nc3 **A prelude to a pawn sacrifice.** cxd4 8. exd4 Bxd4 **MAV–MCTS sacrifices a central pawn and likely loses castling rights for a compensation that is unclear to a**

**human eye.** 9. Nf3 Be3 10. Qd3 Nc6 11. Nd5 Qxb2 12. Qxe3

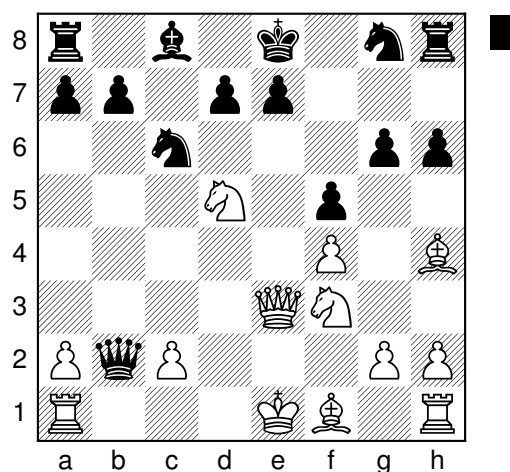

12...Qxa1+ **MAV–MCTS proceeds with sacrificing an exchange with a check.** 13. Kf2 Kf8

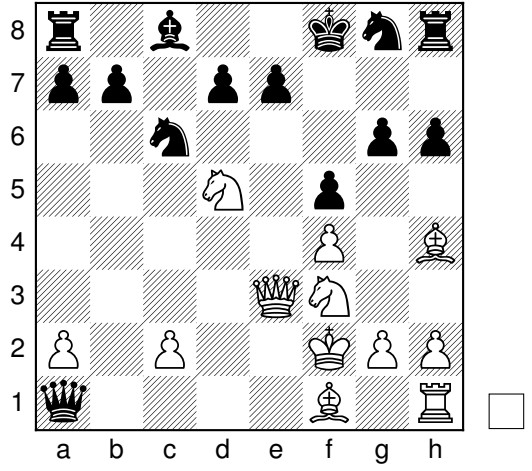

14. Bb5 **MAV–MCTS sacrifices the rook!** Qxh1 15. Bxc6 g5 16. fxg5 dxc6

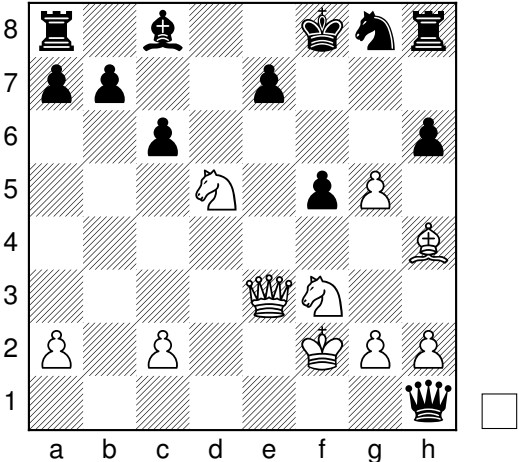

17. g6 **A nail in the coffin – Black is lost despite being two rooks up.** Qa1 18. Bxe7+ Kg7 19. Bd6 cxd5 20. Qe8 Be6 21. Qxe6 (game terminated by early termination). **MAV–MCTS's playing style resembles games of players such as GM Mikhail Tal and IM Rashid Nezhmetdinov who sacrificed material for unclear long-term attacks. Note that the game was played with limited resources during the development phase, which affected the evaluation of some moves.** 1 − 0

## Game 2
### Restricting opponent's activity
**Date: 2024-11-12**
**White: MAV–MCTS($M = 2000$) - max scoring**
**Black: Stockfish–L20**
**Result: 1 − 0**
**Online PGN: https://lichess.org/xLsaUTr8**
1. e4 e5 2. f4 exf4 3. Nf3 g5 4. Nc3 Nc6 5. g3 g4 6. Nh4 f3 7. d4 Bb4 8. d5 Qe7 📚 9. Be3 Ne5 10. Nf5 Bxc3+ 11. bxc3 Qa3

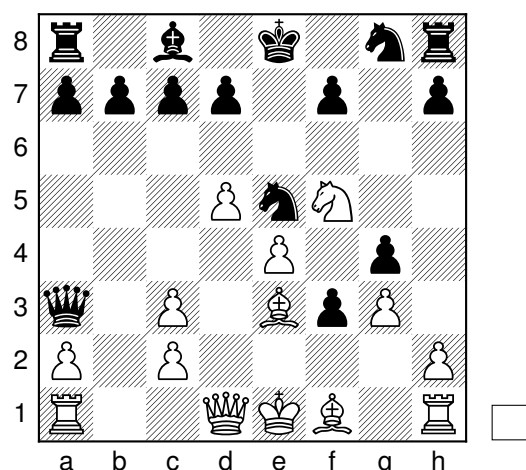

12. d6 **MAV–MCTS sacrificed the second pawn to paralyze Black's queenside. White will further rapidly develop its pieces by attacking the exposed Black's queen.** Qxc3+ 13. Kf2 Nf6 14. Bd3 cxd6 15. Bd4 Qc6

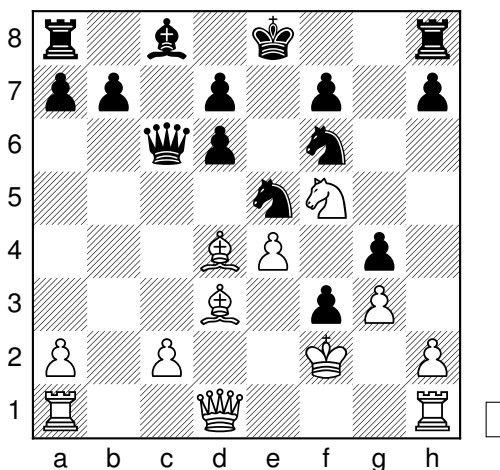

16. a4 **At the first glance, this move aims to stop b5 and queenside development, but the move also has a much deeper idea.** Rg8 17. Ra3 **Now the real idea behind 16.a4 becomes apparent; MAV–MCTS wants to activate its rook via c3!** Rg5 18. Qd2 Rxf5 **A knight paralyzes the entire Black's position, so Black desperately sacrifices the rook for some activity.** 19. exf5 Ne4+ 20. Bxe4 Qxe4 21. Re1 Qxf5 22. Rd3 f6 23. Bb2 Kd8 24. Rxd6 b6 25. Bxe5 fxe5 26. Rd5 Qf6 27. Rexe5 Kc7 28. Re3 h6 29. Rc3+ Kb8 30. a5 Qe7 31. Qf4+ d6 32. Re3 Qc7 33. Rxd6 Kb7 34. Qe4+ Kb8 35. Rc6 Bd7 36. Rxc7 Kxc7 37. Qxa8 bxa5 38. Qxa7+ Kc6 39. Rc3+ Kd6 40. Rd3+ Kc6 41. Qxd7+ Kb6 42. Rd6+ Kc5 43. Rc6+ Kb4 44. Qb7+ Ka3 45. Qb3# 1 − 0

# Game 3
### Precise pawn endgame
**Date: 2024-11-12**
**White: MAV–MCTS($M = 2000$) - max scoring**
**Black: Stockfish–L18**
**Result: 1 − 0**

1. e4 Nc6 2. d4 d5 3. Nc3 Nf6 4. e5 Nd7 📚 5. Nce2 e6 6. c3 f6 7. Nf4 Qe7 8. exf6 Nxf6 9. Be2 Qd7 10. Nf3 Bd6 11. O-O O-O 12. Nd3 Ne4 13. Nfe5 Bxe5 14. Nxe5 Nxe5 15. dxe5 a5 16. Be3 b6 17. Qc2 Bb7 18. Bd3 h6 19. Rad1 Qe8 20. f3 Nc5 21. Bh7+ Kh8 22. Bg6 Qe7 23. Qd2 Kg8 24. Bb1 Ba6 25. Qc2 g5 26. Bxc5 bxc5 27. Qg6+ Qg7 28. Qxe6+ Qf7

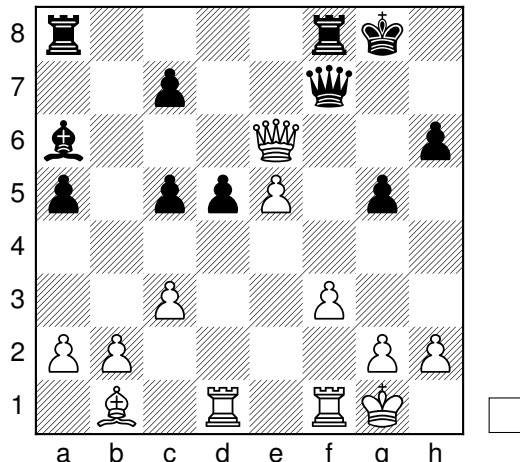

29. Qxh6 **MAV–MCTS correctly sacrifices an exchange to ruin Black's kingside.** Bxf1 30. Kxf1 c6 31. Re1 Rae8 32. Bg6 Qg7 33. Qxg7+ Kxg7 34. Bxe8 Rxe8 35. e6 Kf6 36. e7 c4 37. b3 cxb3 38. axb3 Rxe7 39. Rxe7 Kxe7 40. Ke1 Kd6 41. g3 c5 42. h4 gxh4 43. gxh4 Kd7 44. h5 Ke8 45. Kd1 Kf8

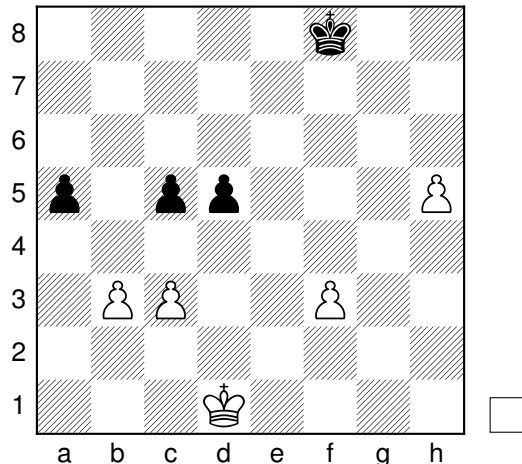

46. f4 **MAV–MCTS plays the only winning move in a seemingly simple pawn endgame. The idea is to timely set up the f5-h5 pawn formation.** Kg7 47. f5 Kg8 48. Kc2 Kh7 49. f6 Kh6 50. Kb1 Kh7 51. Kc2 Kh6 52. Kd3 Kh7 53. Ke3 Kh8 54. Kf4 Kg8 55. Kf5 d4 56. cxd4 Kh7 57. f7 Kg7 58. Ke6 c4 59. Ke7 Kh6 60. f8=Q+ Kh7 61. Kf6 cxb3 62. Qg7# 1 − 0

# Game 4
### Building a fortress
**Date: 2024-11-12**
**White: Stockfish–L18**
**Black: MAV–MCTS($M = 2000$) - max scoring**
**Result: 1/2 − 1/2**

**Online PGN: https://lichess.org/a5Rqsg8s**

1. d4 Nf6 2. Bg5 Ne4 3. Bf4 c5 4. f3 Qa5+ 5. c3 Nf6 6. d5 Qb6 7. e4 📚 Qxb2 8. Nd2 Qxc3 9. Bc7 g6 10. Rc1 Qe3+ 11. Ne2 d6 12. Rb1 Nfd7 13. f4

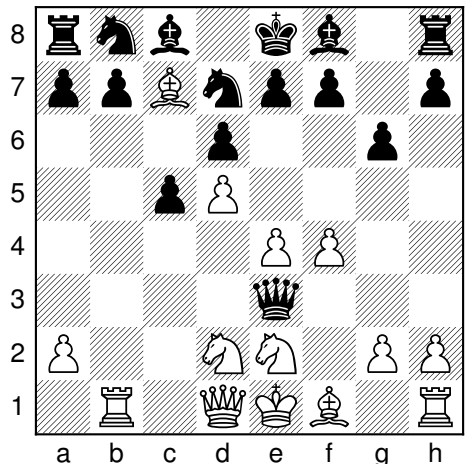

13...Na6 **Prelude to a queen sacrifice.** 14. Rb3 Qxb3 **MAV–MCTS sacrifices a queen hoping its position is solid enough to hold back White's pieces activity.** 15. axb3 Nxc7 16. Qa1 Rg8 17. h4 Bg7 18. Nc3 b6 19. h5 Bb7 20. hxg6 hxg6 21. Rh7 Bd4 22. Qc1 a6 23. Nc4 Nf6 24. Rh3 Bxc3+ 25. Rxc3 Nxe4 26. Re3 f5 27. Nxb6 Nxd5 28. Nxa8 Nxe3 29. Qxe3 Bxa8 30. g4 Rh8 31. gxf5 gxf5 32. b4 cxb4 33. Qb6 Kd7 34. Qd4 Rc8 35. Bxa6 Rc1+ 36. Ke2 Nc5 37. Bb5+ Bc6 38. Qxb4 Ra1 39. Bc4 Ra4 40. Qc3 Kc7 41. Kd1 Kb6 42. Qc1 e6 43. Qb2+ Kc7 44. Qc3 Bd7 45. Ke1 Ne4 46. Qc1 Ra5 47. Bf1+ Bc6 48. Be2 Nc5 49. Qc3 Ra4 50. Qg3 Ra1+ 51. Bd1 Kb6 52. Qc3 Rb1 53. Qd4 Kc7 54. Qg7+ Kb6 55. Qe7 Ba4 56. Qxd6+ Bc6 57. Qd8+ Kb7 58. Qf8 Nd7 59. Qa3 Kb6 60. Qd3 Rc1 61. Kd2 Rc5 62. Qb3+ Kc7 63. Be2 Bd5 **MAV–MCTS builds a fortress.** 64. Qb4 Bh1 65. Ba6 Bc6 66. Ke2 Kd8 67. Bc4 Bd5 68. Ba6 Bc6 69. Kd2 Bh1 70. Ke3 Bd5 71. Bb5 Rc7 72. Ba4 Ra7 73. Qd4 Rc7 74. Qh8+ Ke7 75. Kf2 Kd6 76. Qg7 Bh1 77. Bb5 Bc6 78. Qd4+ Ke7 79. Qb4+ Kd8 80. Bc4 Bd5 81. Bd3 Ke8 82. Ke3 Rb7 83. Qd4 Be4 84. Qh8+ Ke7 85. Bc4 Rc7 86. Qg7+ Kd6 87. Bb5 Bc6 88. Bc4 Bd5 89. Ba6 Be4 90. Qa1 Rc5 91. Kf2 Bd5 92. Be2 Bc6 93. Kg3 Rd5 94. Qa3+ Rc5 95. Bf1 Kc7 96. Be2 Bd5 97. Kh4 Kc8 98. Qb2 Kc7 99. Kg3 Bc4 100. Bf3 Bd5 101. Bd1 Rc4 102. Bh5 Bc6 103. Qg7 Rc1 104. Qe7 Rc3+ 105. Kf2 Rc2+ 106. Kg1 (threefold repetition). **MAV–MCTS proves its decision to sacrifice a queen on move 14 was correct.** 1/2 – 1/2

## Game 5
**Masterful middlegame maneuvering**
**Date: 2024-11-12**
**White: MAV–MCTS**($M = 2000$) **- max scoring**

**Black: Stockfish–L18**
**Result: 1 – 0**
**Online PGN: https://lichess.org/iJa6Nxlq**

1. e4 e6 2. d4 d5 3. e5 c5 4. c3 Ne7 5. Nf3 Nec6 📚 6. Bd3 Be7 7. O-O Nd7 8. Re1 f6 9. exf6 Nxf6 10. dxc5 a5 11. c4 O-O 12. Nc3 dxc4 13. Bxc4 Qxd1 14. Rxd1 Bxc5 15. Na4 Ba7 16. Be3 Nd5 17. Bxa7 Nxa7 18. Nc5 Nc6 19. Ng5 Re8 20. Rac1 Nb6 21. Bb5 a4 22. a3 Ra5 23. Bxc6 bxc6 24. Nge4 h6 25. Rd6 Rb5 26. Rd2 Kh7 27. Nd6 Rd8 28. Nce4 Rd5 29. Rxc6 Rxd2 30. Nxd2 Na8 31. N6c4 Bb7 32. Rxe6 Rd4 33. f3 Bd5

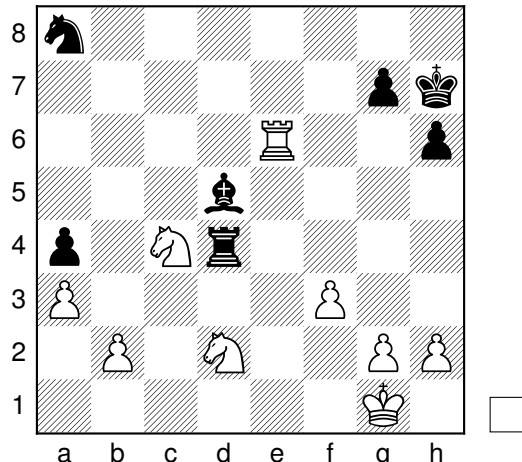

34. Rd6 **After dominating the key squares by masterful maneuvers, MAV–MCTS correctly evaluates that entering a rook versus minor pieces endgame is decisive.** Rxc4 35. Nxc4 Bxc4 36. Rc6 Bd5 37. Ra6 Nc7 38. Rxa4 Kg8 39. Ra7 Ne6 40. b4 Kf8 41. b5 Ke8 42. b6 Nd8 43. Rxg7 Nf7 44. a4 Kd7 45. a5 Kc6 46. Rg6+ Kb7 47. h4 h5 48. Rf6 Bc4 49. Rf5 Nd8 50. Rxh5 Bf7 51. Rh8 Nc6 52. Rh7 Ne5 53. f4 Nd7 54. Rxf7 Kc6 55. Rxd7 Kxd7 56. a6 Kd6 57. b7 Kc7 58. h5 Kd6 59. b8=Q+ Kc6 60. a7 Kd5 61. a8=Q+ Kc5 62. Qa4 Kd5 63. Qe5# 1 – 0

## Game 6
**Stunning piece sacrifice**
**Date: 2024-11-14**
**White: MAV–MCTS**($M = 100$)
**Black: Stockfish–L18**
**Result: 1/2 – 1/2**
**Online PGN: https://lichess.org/pTgHnsqV**

1. e4 c5 2. Nf3 d6 3. d4 cxd4 4. Nxd4 Nf6 5. Nc3 a6 6. Be3 e6 7. f3 Be7 8. Qd2 b5 9. g4 Nfd7 📚 10. O-O-O Ne5 11. g5 b4 12. Na4 O-O 13. f4 Ned7 14. Qxb4 Nc5 15. h4 Bd7

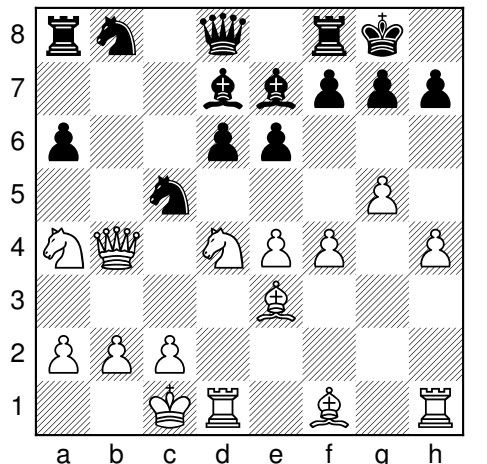

16. Nxe6 **MAV–MCTS sacrifices a knight on a square defended by three pieces.** Nxe6 17. f5 d5 **It is still unclear what MAV–MCTS gained for a sacrificed piece.** 18. Qb3 Bxa4 19. Qxa4 Bc5 20. Kb1 Bxe3 21. fxe6 d4 22. exf7+ Kh8 23. e5 Rxf7 24. Bg2 Raa7 25. Rhe1 Bf2 26. e6 Rf5 27. Re2 Re7 28. Rf1 Qf8 29. Qb4 Nd7 30. Bh3 d3 31. cxd3 Bc5 32. Rxf5 Bxb4

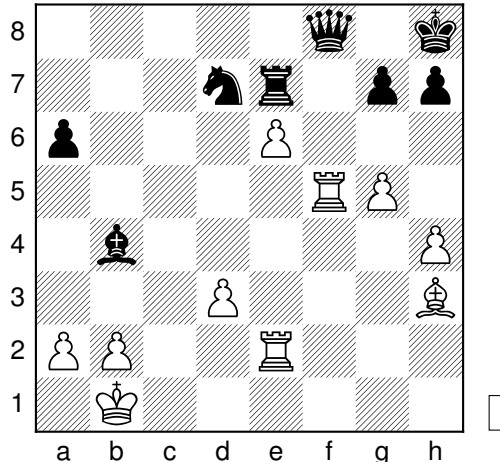

33. exd7 **It was likely better to grab the queen, but the position remain unclear.** Qd8 34. Rfe5 Rxe5 35. Rxe5 Be7 36. d4 h6 37. g6 a5 38. Re1 Kg8 39. d5 Qb6 40. h5 Kf8 41. a3 a4 42. Rc1 Qa5 43. d6 Bd8 44. Rc8 Qe1+ 45. Ka2 Qe6+ 46. Ka1 Qf6 47. Bg4 Qg5 48. Bf5 Qf6 49. Be6 Qf1+ 50. Ka2 Qa1+ 51. Kxa1 (stalemate).

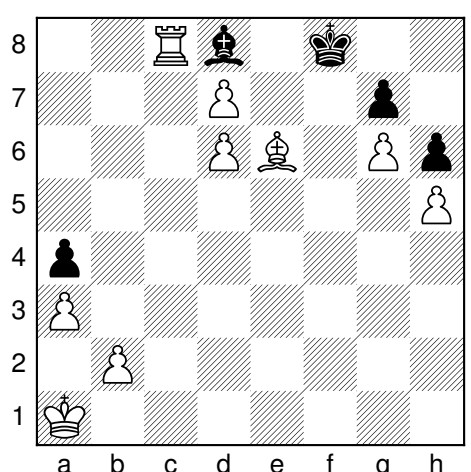

**Stunning end to an exciting game.** 1/2 – 1/2

# Game 7
## Exquisit active defense

**Date: 2024-11-14**
**White: Stockfish–L19**
**Black: MAV–MCTS**($M = 250$)
**Result: 0 – 1**
**Online PGN: https://lichess.org/nyMarqhl**

1. e4 c5 2. Nc3 Nc6 3. Nf3 g6 4. d4 cxd4 5. Nxd4 Bg7 6. Be3 d6 7. h3 Nf6 8. g4 O-O 9. g5 Ne8 10. h4 Nc7 11. f4 e5 12. Nde2 f5 📗 13. h5 fxe4 14. Qd2 d5 15. O-O-O d4 16. hxg6 **This is a start of an attack where a cost of a mistake is an immediate loss.** dxc3 17. Qxc3 Qe8 18. Qb3+ Be6 19. Qxb7 Nd5 20. Bc5 Rb8 21. gxh7+ Kh8

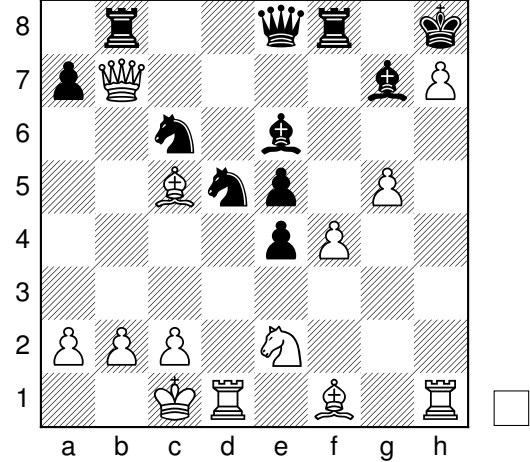

22. Qxg7+ **White sacrifices a queen as the last resource.** Kxg7 23. Bxf8+ Qxf8

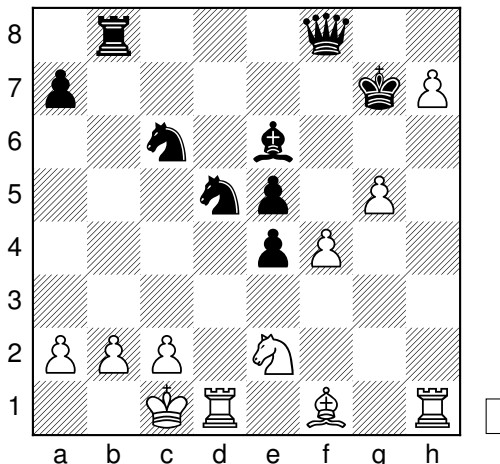

24. h8=R **White finds an opportunity to underpromote to a rook!** Qxh8 25. Rxh8 Rxh8 26. f5 Ne3 27. fxe6 Nxd1 28. Kxd1 Rh1 **After the dust has settled, MAV–MCTS has a decisive advantage.** 29. Ng3 Rg1 30. Nf5+ Kg6 31. e7 Kf7 32. g6+ Ke8 33. Ke1 Nd4 34. Ne3 Kxe7 35. c3 Nf3+ 36. Ke2 Rxg6 37. Nc4 Kf6 38. Ne3 Kg5 39. Kf2 Rd6 40. Be2 Nh4 41. b4 Rd2 42. a4 Nf5 43. Nxf5 Kxf5 44. Ke3 Rc2 45. c4 Ra2 46. c5 Rxa4 47. b5 Ra3+ 48. Kd2 e3+ 49. Kc2 Ra2+ 50. Kd3 Kf4 51. c6 Rd2+ 52. Kc3 Rd8 53. Kb4 Kg3 54. Ka5 Kf2 55. Bc4 Rd4 56. b6 Rxc4 57. bxa7 e2 58. a8=Q e1=Q+ 59. Kb6 Qb4+ 60. Ka7 Qa5+ 61. Kb7 Rb4+ 62. Kc8 Qxa8+ 63. Kd7 Qg8 64. Ke7 Qb8 65. Kf6 e4 66. c7 Qg8 67. Kf5 Qf7+ 68. Kg5 Qd7 69. c8=Q Qxc8 70. Kh5 Rb1 71. Kh4 Rg1 72. Kh5 Qh8# 0 − 1

## Game 8
**The strength of bishop pair**
**Date: 2024-11-14**
**White: MAV–MCTS($M = 500$)**
**Black: Stockfish–L18**
**Result: 1 − 0**
**Online PGN: https://lichess.org/pHzcWtOd**
1. d4 f5 2. g3 Nf6 3. Bg2 e6 4. Nf3 Be7 5. c4 O-O 6. O-O d6 7. Nc3 Qe8 📚 8. b4 Kh8 9. Qc2 Nbd7 10. a4 c6 11. a5 e5 12. b5 a6 13. bxc6 bxc6 14. dxe5 dxe5

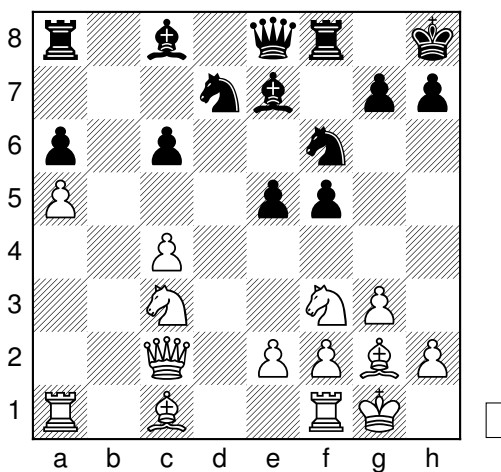

15. Qxf5 **MAV–MCTS grabs a pawn without fearing numerous discovered attacks.** Nd5 16. Qc2 Nxc3 17. Qxc3 e4 18. Ng5 **Already on move 15, MAV–MCTS needed to assess the exchange sacrifice correctly.** Bf6 19. Qa3 Bxa1 20. Qxa1 Nf6 21. Ba3 Rg8 22. Bb2 h6 23. Bxf6 hxg5 24. Bxg5 Bg4 25. Re1 Qh5 26. h4 Bxe2 27. Bxe4 Bxc4 28. Qd4 Bb3 29. Bxc6 Rab8 30. Kh2 Qg6 31. Bd7 Rgf8 32. Re5 Qc2 33. Be3 Qd1 34. Rg5 Qxd4 35. Bxd4

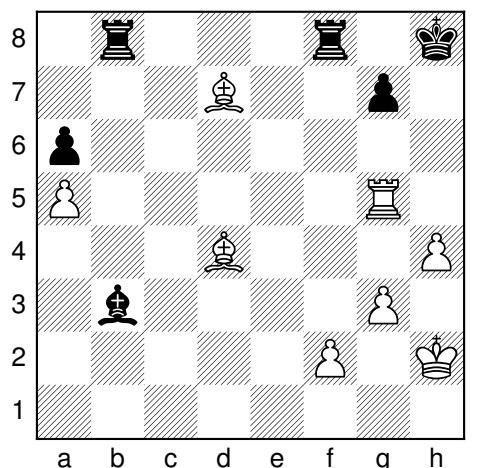

**A bishop pair dominates the board even in the endgame.**
35...Rf7 36. Bf5 Kg8 37. Bb6 Rf6 38. g4 Rfxb6 39. axb6 Rxb6 40. Bd3 Bd1 41. Bc4+ Kf8 42. Kg3 Rf6 43. Bd5 a5 44. Rh5 Ra6 45. Rh8+ Ke7 46. h5 Rd6 47. Be4 Rd8 48. Rh7 Kf6 49. f4 Rg8 50. g5+ Ke6 51. h6 gxh6 52. Rxh6+ Ke7 53. Ra6 a4 54. g6 Be2 55. Rxa4 Kf6 56. Rb4 Rxg6+ 57. Bxg6 Kxg6 58. Rb1 Bc4 59. Rb4 Be6 60. Rb1 Kf6 61. Rb4 Bc8 62. Rb1 Kf5 63. Rb8 Bd7 64. Rb7 Ba4 65. Re7 Kg6 66. Re2 Kf5 67. Ra2 Bb3 68. Ra7 Be6 69. Rb7 Bc4 70. Rg7 Be6 71. Rb7 Bc4 72. Rd7 Kg6 73. Kf3 Kf5 74. Rg7 Bd3 75. Rh7 Be4+ 76. Ke3 Bc6 77. Rh3 Be8 78. Rh7 Bg6 79. Re7 Kf6 80. Rc7 Be8 81. Rc4 Bb5 82. Rc8 Bd7 83. Ra8 Bh3 84. Ra2 Kf5 85. Kf3 Kf6 86. Ra8 Ke6

87. Rc8 Kf6 88. Rc4 Kf5 89. Kf2 Ke6 90. Rc2 Bf5 91. Rc7 Kd6 92. Rb7 Kd5 93. Kg2 Kc5 94. Rf7 Bd3 95. Kf3 Kc6 96. Kg4 Be2+ 97. Kh4 Bf1 98. f5 Bb5 99. Re7 Bc4 100. Kg5 Bg8 101. Kg6 Kc5 102. Kg7 Bb3 103. Rd7 Kc6 104. Ra7 Kd5 105. Ra6 Ke4 106. Rb6 Bd5 107. Rd6 Ke5 108. Rxd5+ Kxd5 109. Kh6 Ke4 110. Kg5 Kd5 111. f6 Ke6 112. Kg6 Kd6 113. f7 Ke7 114. Kg7 Kd7 115. f8=Q Kc7 116. Qe7+ Kb6 117. Kg6 Kb5 118. Qa3 Kc4 119. Kf7 Kd4 120. Qc1 Ke4 121. Qg5 Kd4 122. Qb5 Kc3 123. Kf6 Kd2 124. Qf1 Ke3 125. Kf5 Kd4 126. Qc1 Kd5 127. Qh6 Kc4 128. Qb6 Kd5 129. Qh6 Kc5 130. Qa6 Kb4 131. Qa2 Kc3 132. Ke4 Kb4 133. Kf3 Kb5 134. Qa7 Kc6 135. Ke4 Kb5 136. Qa3 Kb6 137. Kd5 Kb7 138. Qa5 Kb8 139. Kc6 Kc8 140. Qc7# **Note that MAV–MCTS was inefficient in delivering forced mates because of a very low simulation budget.** $1 - 0$

# Game 9
## Material quality over quantity
**Date: 2024-11-14**
**White: Stockfish–L19**
**Black: MAV–MCTS(** $M = 500$ **)**
**Result: 1/2 – 1/2**
**Online PGN: https://lichess.org/Mka8BrUO**

1. d4 Nf6 2. c4 c5 3. d5 e5 4. Nc3 d6 5. e4 Be7 6. Nf3 Nbd7 7. g3 O-O 8. Bg2 📚

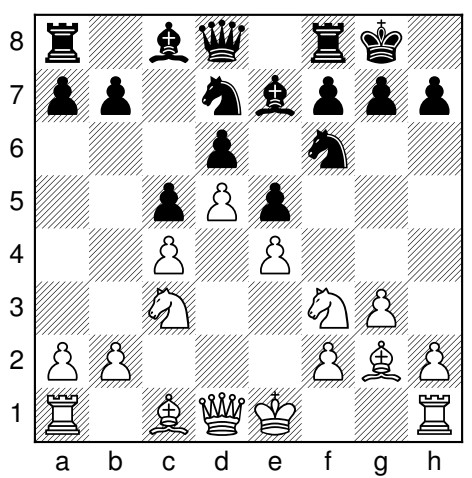

**8...b5 MAV–MCTS plays a pawn sacrifice similar to Benko gambit.** 9. cxb5 a6 10. Bf1 Ne8 11. Nd2 f5 **MAV–MCTS proceeds with shattering White's pawn structure.** 12. exf5

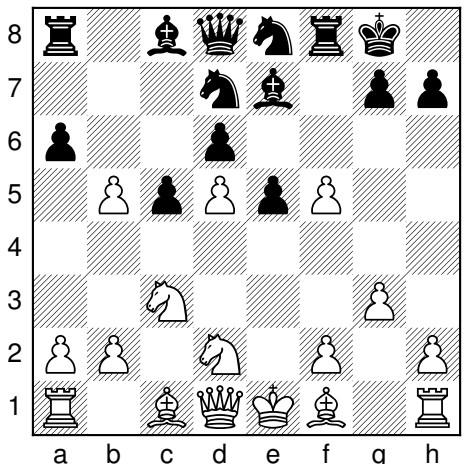

12...Nc7 13. a4 axb5 14. Nxb5 Nf6 15. Bc4 Bxf5 16. O-O Na6 17. f4 e4 **Despite being a pawn down, MAV–MCTS has a compact pawn structure without any weaknesses that offer an active piece play.** 18. Re1 Nb4 19. Ra3 Nd3 20. Re3 Nb4 21. Re2 Kh8 22. Nf1 Bg4 23. Rae3 Qe8 24. Qb3

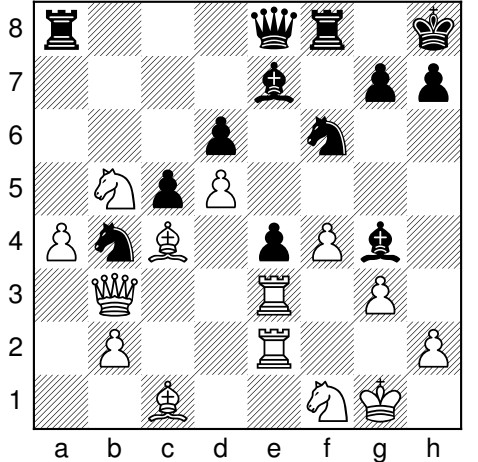

24...Qh5 **For two moves in a row MAV–MCTS estimates that the bishop is a stronger piece than the rook and declines to win an exchange.** 25. Re1 Rfb8 26. Bd2 Nbxd5 27. Rxe4 Bf8 28. R4e2 Nb6 29. Bf7 Qh3 30. Be6 Nxa4 31. Bxg4 Qxg4 32. Qf7 Qg6 33. Qxg6 hxg6 34. Nc7 Ra7 35. Ne6 Nxb2 36. Bc3 Nc4 37. Bxf6 gxf6 38. Rc1 d5 39. Rd1 Rd7 40. Rc2 Nb6 41. Rb1 Bd6 42. Rcb2 Re8 43. Rxb6 Rxe6 44. Kf2 c4 45. Rc6 Rc7 46. Rbb6 Rxc6 47. Rxc6 Kg7 48. Kf3 Kf7 49. Ne3 Bb8 50. Rc8 Rb6 51. Nxd5 Rb3+ 52. Ke4 f5+ 53. Kd4 c3 54. Kc4 Rb7 55. Nxc3 Ba7 56. Nd5 Bg1 57. h3 Bh2 58. g4 fxg4 59. Rh8 Kg7 60. Rh4 g3 61. Rg4 Rb2 62. Kc3 Rf2 63. Ne3 Rf3 64. Kd2 g2 65. Nxg2 Rxh3 66. Ke1 Bxf4 67. Kf2 g5 68. Nh4 Rxh4 69. Rxh4 gxh4 70. Ke1 h3 71. Kf2 Bh2 72. Kf3 Bd6 73. Kf2 Bh2 74. Kf3 Bd6 75. Kf2 Bh2 (threefold repetition). **This**

is a well known draw because the queening square is the opposite color of the bishop. 1/2 – 1/2

# Game 10
### Exemplary advantage conversion
**Date: 2024-11-14**
**White: MAV–MCTS($M = 2000$)**
**Black: Stockfish–L19**
**Result: 1 – 0**
**Online PGN: https://lichess.org/HLOwBsg9**

1. d4 d5 2. c4 c6 3. Nf3 Nf6 4. e3 e6 5. Nc3 a6 6. c5 Nbd7 7. b4 a5 📚 8. b5 g6 9. Be2 Bg7 10. O-O h5 11. Rb1 O-O 12. Na4 Qe7 13. Ne5 Nxe5 14. dxe5 Ng4 15. Nb6 Rb8 16. f4 Bd7 17. Bxg4 hxg4 18. Qxg4 cxb5 19. Nxd7 Qxd7 20. Bb2 b6 21. cxb6 Rxb6 **The material is equal, but MAV–MCTS has much more active bishop and upper-hands for a kingside attack with the minimal resources.** 22. h4 Rc6 23. h5 b4 24. Rf3 Rfc8 25. hxg6 fxg6 26. Qxg6 Qf7 27. Qg5 Qf5 28. Qxf5 exf5 29. Rg3 Kh7 **The conversion is not easy because MAV–MCTS's rooks are seemingly deprived of any activity.**

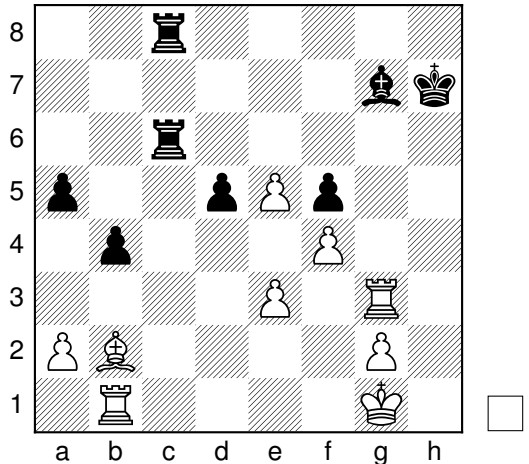

30. Kf2 **MAV–MCTS finds a beautiful maneuver which activates all its pieces at the cost of a pawn.** Rc2+ 31. Kf3 R8c4 32. Rh1+ Kg8 33. Bd4 Rxa2

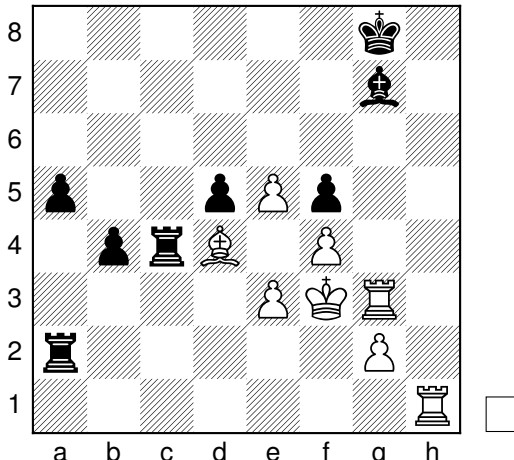

**The endgame looks scary for White because of the advanced Black's pawns, but MAV–MCTS correctly evaluates that Black's king is too weak.** 34. Rg5 Rd2 35. e6 Rdxd4 36. exd4 b3 37. g4 Rc3+ 38. Ke2 Kf8 39. e7+ Kxe7 40. Rxg7+ Kf6 41. Rb7 fxg4 42. Rh5 Rc4 43. Rxb3 Rxd4 44. Rb6+ Kf7 45. Rh7+ Kg8 46. Rd7 Re4+ 47. Kd3 Rxf4 48. Ra7 Rf8 49. Rg6+ Kh8 50. Kd4 a4 51. Rh6+ Kg8 52. Rg6+ Kh8 53. Ke5 d4 54. Rh6+ Kg8 55. Rg6+ Kh8 56. Ke6 Rf4 57. Rf6 Rf1 58. Rxf1 Kg8 59. Ke5 g3 60. Rg1 d3 61. Rxg3+ Kh8 62. Kf5 d2 63. Ra8+ Kh7 64. Ra7+ Kh8 65. Rd7 d1=B 66. Rxd1 Kh7 67. Rh1# 1 – 0

# Game 11
### Controlling key squares
**Date: 2024-11-12**
**White: MAV–MCTS($M = 2000$) - max scoring**
**Black: Stockfish–L18**
**Result: 1 – 0**
**Online PGN: https://lichess.org/l2F7vJ7q**

1. e4 c5 2. Nf3 d6 3. d4 cxd4 4. Nxd4 Nf6 5. Nc3 g6 6. Be3 Bg7 7. f3 O-O 8. Qd2 Nc6 9. O-O-O d5 10. Qe1 e5 11. Nxc6 bxc6 12. exd5 Nxd5 13. Bc4 Be6 14. Kb1 Rb8 📚 15. Ne4 Qc7 16. Bc5 Rfd8 17. g4 a5 18. a4 h6 19. Bb3 Kh8

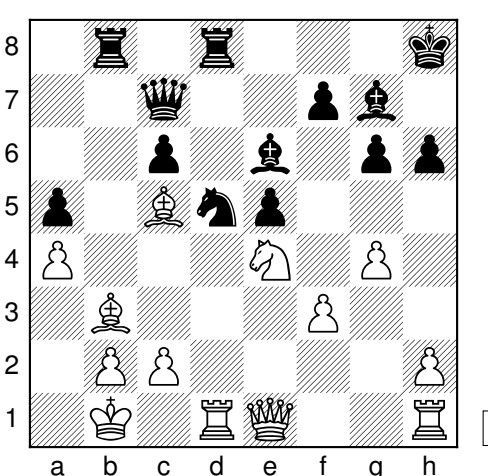

20. Ba3 **MAV–MCTS vacates c5 square for a knight and estimates that the control over a3-f8 diagonal is much more important than the control over g1-a7 diagonal. Additionally, it protects b2 square from any possible attack along the b line.** Nf4 21. Bxe6 Nxe6 22. h4 Rxd1+ 23. Qxd1 c5 24. Qd6 Qb7 25. Qd3 Qc6 26. g5 h5 27. Qc4 Qd7 28. Ka2 Rd8 29. Rf1 Qc8 30. Rf2 Rd4 31. Qb5 Qd8 32. Qc6 Rd1 33. Bxc5 Nxc5 34. Qxc5 Kh7 35. Re2 Rd5 36. Qc6 Rd7 37. Re3 Rc7 38. Qb6 Qd7 39. Qb5 Qh3 40. c4 Qxh4 41. c5 Qf4 42. Rd3 h4 43. Qxa5 Rc8 44. Qa6 Rc7 45. Qb6 Rc8 46. Qb7 Rf8 47. c6 h3 48. c7 h2 49. Rd8 h1=Q 50. c8=Q Rxd8 51. Qxd8

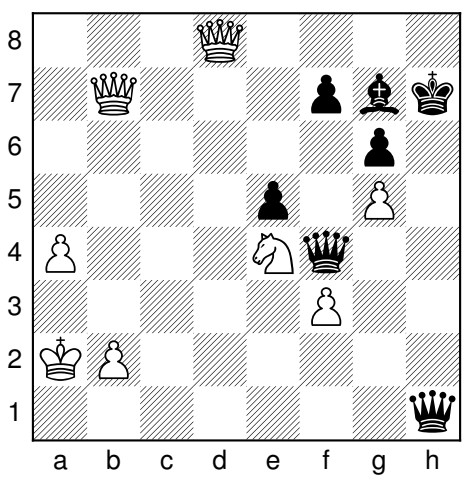

51...Qh3 **A long maneuvering phase of the game results in a four queens endgame! MAV–MCTS is completely winning because of a very weak Black's king.** 52. Qbb8 Qe6+ 53. b3 f5 54. gxf6 Bxf6 55. Qb7+ Be7 56. Qbxe7+ Qxe7 57. Qxe7+ Kh8 58. Qf6+ Kg8 59. Qxf4 exf4 60. a5 g5 61. a6 Kg7 62. a7 Kf7 63. a8=Q g4 64. Qd5+ Ke7 65. Qd6+ Kf7 66. Qf6+ Ke8 67. Nc5 gxf3 68. Qg7 f2 69. Qd7+ Kf8 70. Ne6+ Kg8 71. Qg7# 1 – 0

## Game 12
**Exploiting opponent's aggressive play**
**Date: 2024-11-14**
**White: MAV–MCTS($M = 2000$)**
**Black: Stockfish–L17**
**Result: 1 – 0**
**Online PGN: https://lichess.org/0vDAbIW8**

1. e4 e5 2. Nf3 Nc6 3. Bb5 a6 4. Ba4 Nf6 5. O-O Be7 6. Re1 b5 7. Bb3 O-O 8. a4 b4 9. a5 d5 📚 10. exd5 e4 11. dxc6 **Black sacrifices a knight and relies on aggressive piece play with a possible kingside attack.** Bd6 12. d4 **MAV–MCTS decides to immediately activate its pieces by returning the sacrificed knight.** exf3 13. Qxf3 h6 14. h3 Rb8 15. Nd2 Rb5 16. Nc4 Re8 17. Rxe8+ Qxe8

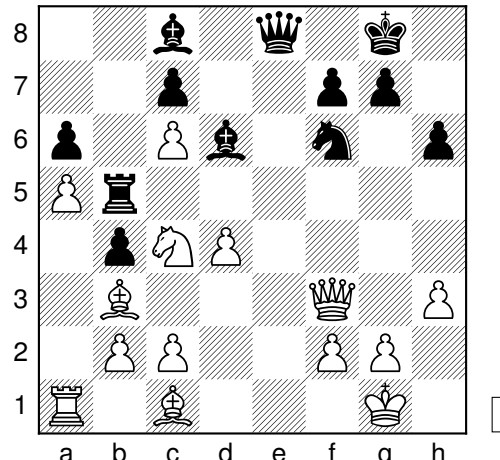

18. Bxh6 **MAV–MCTS enters a scary tactical sequence.** Rf5 19. Nxd6 cxd6 20. Qg3 Nh5 21. Qe3 Qxc6 22. g4 Rf3 23. d5 Rxe3 24. dxc6 Rxh3 25. Bg5 Nf6 26. Bxf6 gxf6 27. Ra4 **MAV–MCTS finishes the resulting winning endgame with a nice rook lift.** Kf8 28. Rxb4 Rh8 29. Rb8 Ke7 30. Bd5 Rg8 31. f3 Rg5 32. c4 Kd8 33. b4 Re5 34. Kf2 Kc7 35. Ra8 f5 36. f4 Rxd5 37. cxd5 Kd8 38. g5 Ke7 39. Rxc8 f6 40. Ke3 fxg5 41. fxg5 Kf7 42. Kf4 Ke7 43. Rg8 Kf7 44. Rc8 Kg7 45. Kxf5 Kh7 46. c7 Kg7 47. Rb8 Kf7 48. c8=Q Ke7 49. Qb7# 1 – 0

## Game 13
**Well executed caveman attack**
**Date: 2024-11-14**
**White: MAV–MCTS($M = 100$)**
**Black: Stockfish–L19**
**Result: 1 – 0**
**Online PGN: https://lichess.org/gvDiGfcP**

1. e4 g6 2. d4 Bg7 3. Nc3 d6 4. Be3 a6 5. Qd2 b5 6. h4 📚 h6 7. h5 b4 8. Nd5 c6 9. Nxb4 g5 10. Ne2 a5 11. Nd3 Nf6 12. f3 Qc7 13. Ng3 c5 14. c3 a4 15. Be2 O-O

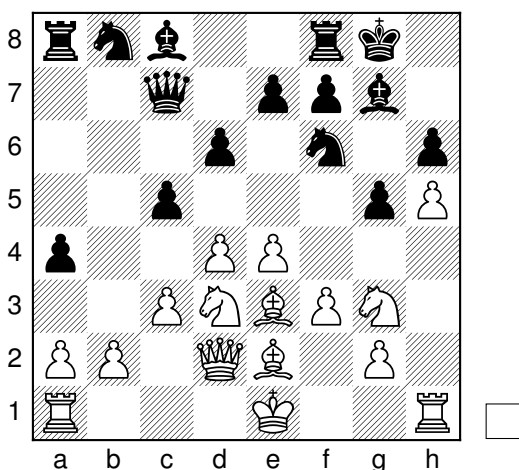

16. Bxg5 **MAV–MCTS sacrifices a bishop for a devastating attack.** hxg5 17. h6 Bh8 18. Qxg5+ Kh7 19. Nf5 Nh5 20. Rxh5 f6

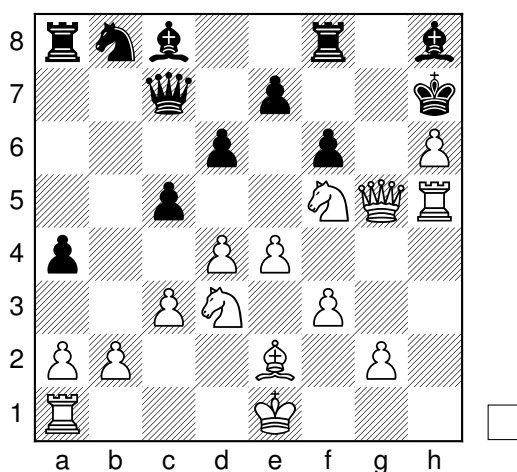

21. Qg7+ **A stunning queen sacrifice – the cherry on top.** Bxg7 22. hxg7+ Kg8 23. Rh8+ Kf7 24. Rxf8+ Ke6 25. Nxc5+ Qxc5 26. g8=Q+ Kd7 27. Rd8+ Kc7 28. Rxc8+ Kb6 29. dxc5+ Kb7 30. Bb5 Ra6 31. O-O-O Ka7 32. g3 a3 33. Nxe7 d5 34. b3 d4 35. f4 Re6 36. Re8 d3 37. Qh8 f5 38. Bxd3 Rf6 39. Bc4 Re6 40. g4 Rxe4 41. Rh1 Rxc4 42. Nc8+ Kb7 43. Qg7+ Nd7 44. bxc4 Kc6 45. Kb1 Kxc5 46. Kc1 Nb8 47. Qd4+ Kc6 48. Rf8 Nd7 49. c5 Nxf8 50. Re1 Nd7 51. Re2 Kc7 52. g5 Kxc8 53. Qh8+ Kc7 54. g6 Nb8 55. Re1 Na6 56. Re7+ Kc6 57. Qa8+ Kb5 58. g7 Nb8 59. c4+ Kxc4 60. Re1 Kb5 61. g8=Q Kxc5 62. Rg1 Na6 63. Kd1 Kb5 64. Rh1 Nc7 65. Rh5 Nxa8 66. Qd5+ Kb6 67. Qc4 Kb7 68. Qd5+ Kc8 69. Rxf5 Nb6 70. Qc5+ Kb7 71. Qc2 Nd5 72. Rxd5 Kb8 73. Re5 Kb7 74. Qf2 Kc7 75. Kc2 Kb8 76. f5 Kb7 77. Qe3 Kc7 78. Qg1 Kd8 79. Qe3 Kc7 80. f6 Kc6 81. Kd2 Kb7 82. Kc2 Ka6 83. Qg1 Kb7 84. f7 Kc7 85. Kd3 Kd8 86. Qf1 Kc7 87. f8=Q Kb7 88. Kd2 Kc7 89. Qe2 Kc6 90. Qe7 Kb6 91. Qb5# **Note that MAV–MCTS**

was inefficient in delivering forced mates because of the extremely low simulation budget. 1 – 0

# Game 14
### King in the center
**Date: 2024-11-21**
**White: MAV–IS**$(b = 4, d = 2)$
**Black: Stockfish–L15**
**Result: 1/2 – 1/2**
**Online PGN: https://lichess.org/XVqKocbp**
1. e4 e6 2. d4 d5 3. Nd2 Nf6 4. e5 Nfd7 5. Bd3 c5 6. c3 b5 📚 7. Ne2 Ba6 8. a3 Nc6 9. Nf3 cxd4 10. cxd4 Qc8 11. b4 Bb7 12. O-O a5 13. bxa5 Rxa5 14. Ng5 Be7 15. Bd2 Rxa3 16. Rxa3 Bxa3 17. Nf4 **MAV–IS starts a sequence that sacrifices two central pawns.** Nxd4 18. Qh5 Nxe5

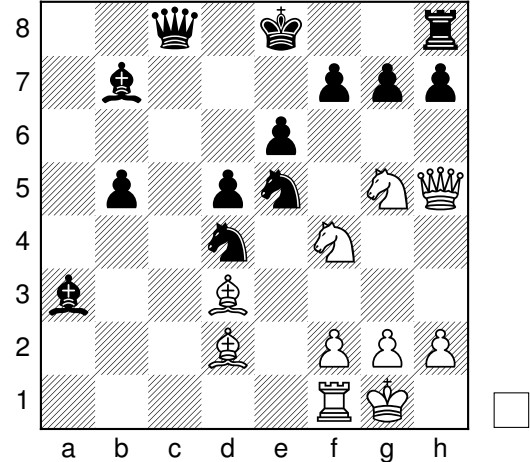

19. Ngxe6 **MAV–IS sacrifices a knight to keep Black's king in the center.** Nxe6 20. Bxb5+ Bc6 21. Nxe6 Bxb5 22. Re1 Bd6 23. Nxg7+ Kd7 24. Rxe5 Bxe5 25. Qxe5 **MAV–IS correctly estimated that the vulnerable Black's king and dark squares are worth the sacrificed exchange.** Qc5

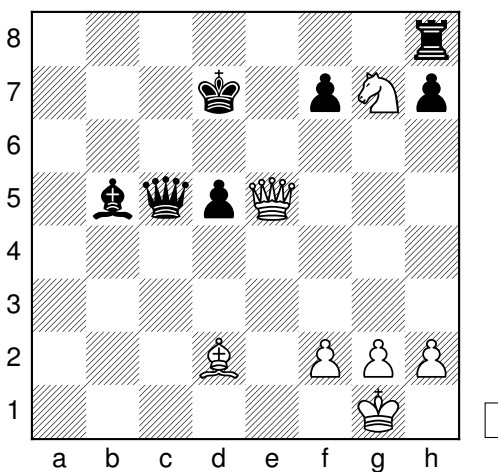

26. h3 **MAV–IS plays a quite prophylactic move which resolves a potential back rank issues before proceeding with the attack.** Ra8 27. Bf4 Bd3 28. Nh5 Rc8 29. Be3 d4 30. Nf6+ Kc6 31. Bxd4 Qd6 32. Qe3 Kb7 33. Be5 Qe6 34. Qd4 Rc4 35. Qb2+ Ka6 36. Qa3+ Kb6 37. Qb3+ Kc5 38. Qb8 Rc1+ 39. Kh2 Rc2 40. Ng4 Bf5 41. Ne3 Rxf2 42. Qa7+ Kb5 43. Qb7+ Kc5 44. Qa7+ Kb4 45. Qb7+ Ka4 46. Qa7+ Kb4 47. Qb7+ Kc5 48. Qa7+ (threefold repetition). 1/2 – 1/2

# Game 15
### Dominating the center
**Date: 2024-11-21**
**White: Stockfish–L10**
**Black: MAV–IS**($b = 4, d = 2$)
**Result: 0 – 1**
**Online PGN: https://lichess.org/8a0RINHx**
1. e4 c5 2. Nf3 Nc6 3. d4 cxd4 4. Nxd4 e5 5. Nb5 a6 6. Nd6+ Bxd6 7. Qxd6 Qf6 8. Qd1 Qg6 9. Nc3 Nge7 📚 10. f4 d5 11. f5

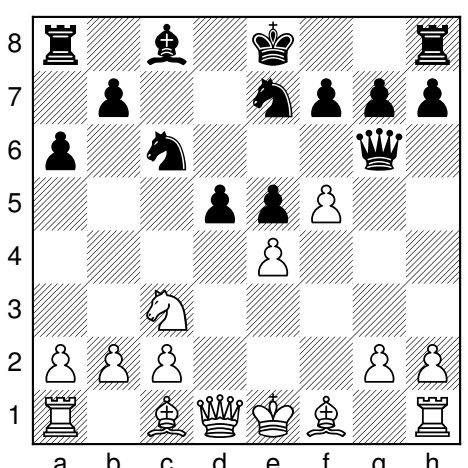

11...Nxf5 **To avoid a passive defense, MAV–IS sacrifices a piece for two central pawns.** 12. exf5 Bxf5 13. Bd3 e4 14. Bf1 d4 15. Nd5

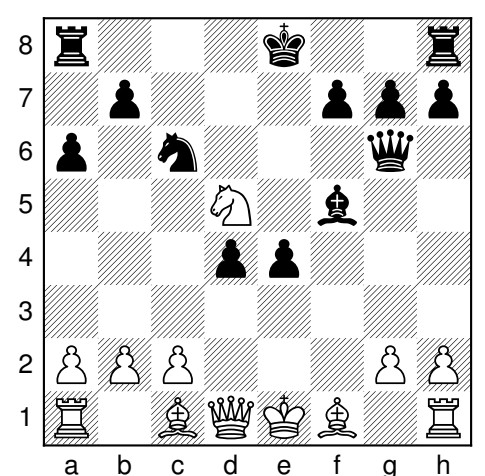

15...O-O-O **MAV–IS pushes backwards the only White's active piece, while all other pieces are still stuck at their original squares.** 16. Nf4 Qd6 17. Be2 Qc5 18. O-O d3+ 19. Kh1 dxe2 20. Qxe2 h5 21. c4 Bg4 22. Qe1 Rd1 23. Be3 Rxe1 24. Bxc5 Rxa1 25. Rxa1 h4 26. Re1 Bf5 27. Kg1 h3 28. b3 hxg2 29. Rd1 Bh3 30. Rd5 f6 31. Nxh3 Rxh3 32. Bd6 Rd3 33. Kxg2 Rxd5 34. cxd5 Nd4 35. Kf2 Nf5 36. Bb4 Kd7 37. Ba3 Kc7 38. Bc5 Kd7 39. a4 Kc7 40. Bb4 Kd7 41. Bf8 g6 42. b4 Ne7 43. Bxe7 Kxe7 44. a5 Kd6 45. Ke1 Kxd5 46. Ke2 Kd4 47. b5 axb5 48. Kf2 (game terminated by early termination). 0 – 1

# Game 16
### The power of zwischenzug
**Date: 2024-11-21**
**White: MAV–IS**($b = 4, d = 2$)
**Black: Stockfish–L10**
**Result: 1/2 – 1/2**
**Online PGN: https://lichess.org/cHJzIu8F**
1. e4 e5 2. f4 exf4 3. Nf3 d6 4. d4 g5 5. g3 g4 6. Nh4 f3 7. Nc3 📚 Nf6 8. Bg5 Be7 9. Qd2 Nc6 10. O-O-O h6 11. Bxh6 Nxd4

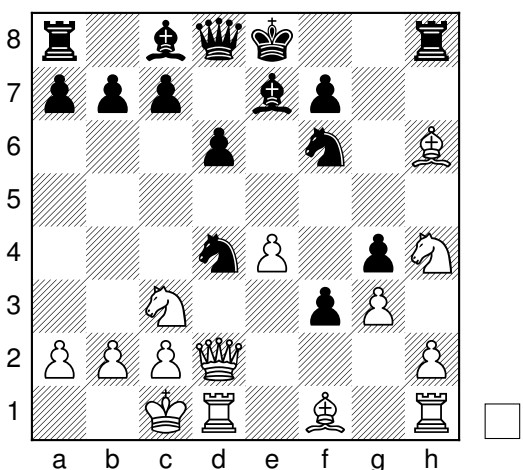

12. Bg7 **Instead of an immediate recapture, MAV–IS complicates the game by playing an in-between move.** Rg8 13. Qxd4 Rxg7 14. e5 Nh5 15. Bc4 Kf8 16. Kb1 d5 17. Nxd5 c6 18. Nxe7 Qxd4 19. Rxd4 Kxe7 20. Re1 b5

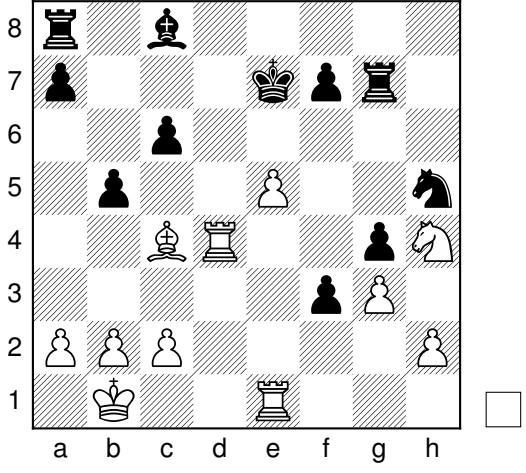

21. e6 **Instead of retreating the attacked piece, MAV–IS plays a strong in-between move that sacrifices a piece and leads to a dynamic equality.** bxc4 22. Nf5+ Kf6 23. Nxg7 Nxg7 24. Rxg4 Bxe6 25. Rf4+ Kg5 26. Rxf3 f5 27. b3 Rh8 28. Rf2 Rh3 29. bxc4 a5 30. Re5 a4 31. Rc5 Bd7 32. Ra5 Be6 33. Rxa4 Ne8 34. Re2 Rh6 35. c5 Ng7 36. Ra7 Kf6 37. Ra8 Bc4 38. Rf8+ Kg6 39. Rf2 Rh7 40. a4 Ne6 41. R8xf5 Ra7 42. Rf6+ Kg7 43. h4 Bd5 44. h5 Nxc5 45. Rg6+ Kh7 46. Rff6 Be4 47. Rg5 Bd5 48. Rgg6 Be4 49. Rg5 Bd5 50. Re5 Rb7+ 51. Kc1 Nxa4 52. g4 Nb6 53. g5 Ra7 54. Kd1 Nc4 55. Re8 Kg7 56. h6+ Kh7 57. Ke1 Rb7 58. Kf2 Rd7 59. Ke2 Na3 60. Kf2 Nxc2 61. Re5 Bg8 62. Kg3 Nb4 63. Re8 Bf7 64. Rb8 c5 65. Rc8 Re7 66. Rxc5 Nd3 67. Rcc6 Re3+ 68. Kh4 Ne5 69. Rc7 Kg8 70. Rc8+ Kh7 71. Rc7 Kg8 72. Rc8+ Kh7 73. Rc7  1/2 – 1/2

# Game 17
**Execution is stronger than a threat**
**Date: 2024-12-01**
**White: MAV–MCTS($M = 2000$)**
**Black: Stockfish–L20**
**Result: 1/2 – 1/2**
**Online PGN: https://lichess.org/Qoyj7u8h**

1. e4 g6 2. d4 Bg7 3. Nc3 d6 4. f4 Nc6 📚 5. Be3 e6 6. Nf3 a6 7. Bd3 b5 8. a4 b4 9. Ne2 Nf6 10. O-O a5 11. f5 **MAV–MCTS does not hesitate to start an immediate pawn break.** exf5 12. exf5 O-O 13. Bg5 Ne7 14. Ng3 Ba6 15. Bxa6 Rxa6 16. Qd3 Ra8 17. Nd2 Ned5 18. c3 bxc3 19. bxc3 Qd7 20. Nde4 Nxe4 21. Nxe4 Rfe8 22. h4 Rab8 23. Rf2 Qc6 24. Qf3 Qd7 25. Qd3 h6 26. Bd2 c5 27. Raf1 cxd4 28. fxg6 fxg6

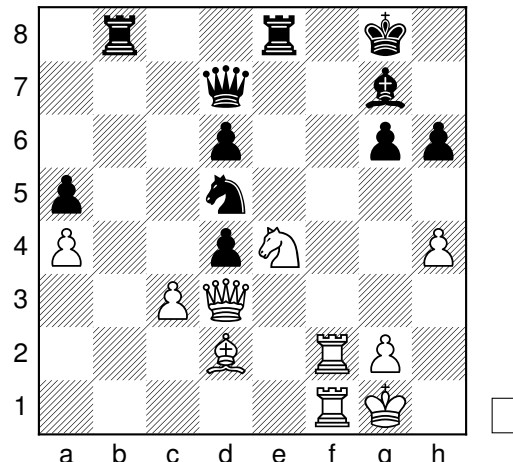

29. Nf6+ **MAV–MCTS does take time for the preparatory moves, but jumps with a knight to a square that is defended twice to open the diagonals and lines around Black's king.** Nxf6 30. Rxf6 Re6 31. Qxg6 Rxf6 32. Rxf6 Qe7 33. Rf3 Qe8 34. Qg4 Kh8 35. Rg3 Be5 36. Bf4 dxc3 37. Bxe5+ Qxe5 38. Qg6 Qd4+ 39. Kh1 Qxh4+ 40. Rh3 Qe1+ 41. Kh2 Qe5+ 42. Kg1 Qe1+ 43. Kh2 Qe5+ 44. Kg1 Qe1+ 45. Kh2 (threefold repetition). **Perpetual check saves Black from getting mated.**  1/2 – 1/2

# Game 18
**The dragon bishop**
**Date: 2024-12-01**
**White: Stockfish–L19**
**Black: MAV–MCTS($M = 2000$)**
**Result: 1/2 – 1/2**
**Online PGN: https://lichess.org/2HlG6ZaD**

1. e4 c5 2. Nf3 d6 3. d4 cxd4 4. Nxd4 Nf6 5. Nc3 g6 6. Be3 Bg7 7. f3 O-O 8. Qd2 Nc6 9. Bc4 Qa5 10. O-O-O Bd7 📚 11. Kb1 Rfc8 12. Nb3 Qd8 13. Be2 Ne5 14. h4 a5 15. Nd4 a4 16. g4 a3 17. b3 Qa5 18. Ncb5 Qxd2 19. Bxd2 Nc6 20. Bc1 Nxd4 21. Nxd4 Rc3 22. Rhe1 h5 23. g5 Ne8 **This**

**seemingly unimportant knight will play a pivotal role in what is to come.**

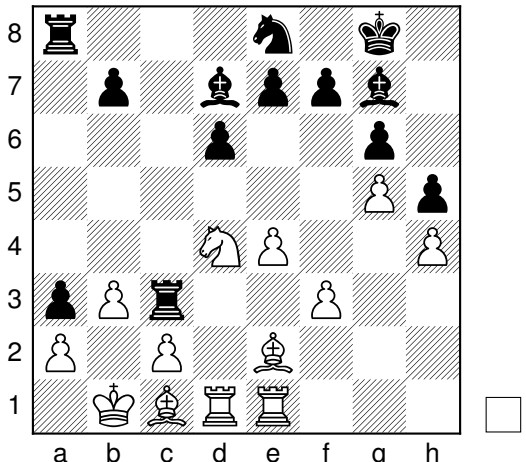

24. Bxa3 **Prelude to a very unusual tactic.** Rxa3

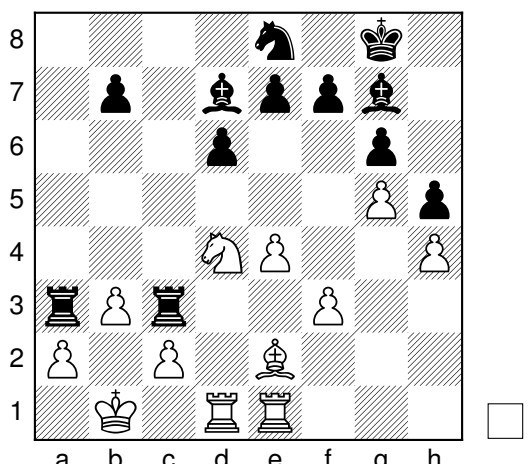

25. Kb2 **A lone king is "forking" two rooks.** Rxa2+ 26. Kxc3

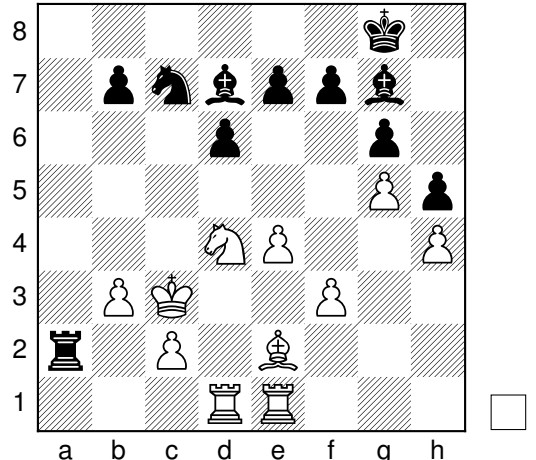

26...Nc7 27. b4 **White makes a "luft" to escape a pin from the dark-squared bishop.** Ne6 **The knight enters the game with a great effect!** 28. Kb3 Ra8 29. Nxe6

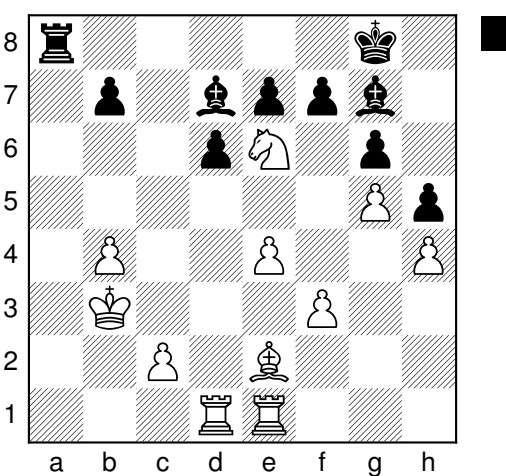

29...Ba4+ **This zwischenschach is the only move which secures sufficient compensation for the exchange because it pushes the king towards the center of the board.** 30. Kc4 fxe6 31. Rc1 Rc8+ 32. Kd3 Bc3 33. Rf1 Bxb4 34. c3 Ba5 35. Bd1 Bb5+ **The light-squared bishop takes over the role of a pinner.** 36. c4

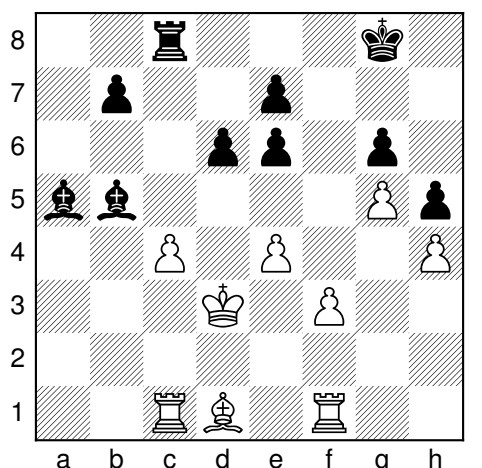

36...d5 **MAV–MCTS is pinning White from every angle!** 37. Bb3 Ba6 38. f4 Kg7 39. exd5 exd5 40. Kd4 Rd8 41. c5 **White returns the exchange to release the tension and enter a balanced endgame.** Bxf1 42. Rxf1 e6 43. Ra1 b6 44. Kd3 Rc8 45. cxb6 Bxb6 46. Ra6 Rb8 47. Kd2 Bf2 48. Bc2 Kf7 49. Ke2 Rb2 50. Kxf2 Rxc2+ 51. Kg3 Ke7 52. Kf3 Rc1 53. Kg2 d4 54. Ra4 Rd1 55. Kf3 Rh1 56. Rxd4 Rxh4 57. Ke3 e5 58. Re4 Rh3+ 59. Kd2 Rf3 60. Rxe5+ Kf8 61. Re6 Kf7 62. Rf6+ Kg7 63. Ke1 h4 64. Ke2 Rg3 65. Re6 h3 66. Kf2 Rg4 67. Re7+ Kg8 68. Ra7 h2 69. Ra8+ Kg7 70. Ra7+ Kg8 71. Ra1 Kf7 72. Re1 Rh4 73. Rh1 Rh3 74. Kf1 Ke6 75. Kg2 Re3 76. f5+ Kxf5 77. Ra1

Re8 78. Kh1 Re7 79. Ra5+ Re5 80. Ra1 Re8 81. Kxh2 Re7 82. Ra4 Re4 83. Ra3 Kg4 84. Ra1 Re3 85. Rg1+ Kf4 86. Rb1 Re2+ 87. Kh1 Re3 88. Rc1 Kg3 89. Kg1 Rb3 90. Kh1 Kh3 91. Kg1 Rb4 92. Kf2 Rf4+ 93. Ke1 Rf5 94. Rc2 Rxg5 95. Kf1 Rg4 96. Re2 g5 97. Re6 Kh2 98. Rh6+ Kg3 99. Rb6 Rc4 100. Rb3+ Kh2 101. Rb2+ Kh1 102. Kf2 Kh2 103. Kf1+ Kh1 104. Kf2 Kh2 105. Kf1+ (threefold repetition). **A peaceful outcome fo a wild game.** 1/2 − 1/2

# Game 19
### Taming the dragon bishop
**Date: 2024-12-01**
**White: MAV–MCTS($M = 2000$)**
**Black: Stockfish–L19**
**Result: 1 − 0**
**Online PGN: https://lichess.org/eKR8u32g**

1. e4 c5 2. Nf3 d6 3. d4 cxd4 4. Nxd4 Nf6 5. Nc3 g6 6. Be3 Bg7 7. f3 O-O 8. Qd2 Nc6 9. Bc4 Qa5 10. O-O-O Bd7 📚 11. Bb3 **MAV–MCTS chooses the different approach to the Sicilian Dragon than Stockfish in Game 18.** Ne5 12. Bh6 **MAV–MCTS immediately exchanges the favorite piece of every Dragon player.** Bxh6 13. Qxh6 Rac8 14. Nd5 Nxd5 15. exd5 Qc5 16. a3 a5 17. h4 f6 18. Rhe1 a4 19. Ba2 b5 20. Qd2 Rfe8 21. h5 gxh5 22. Rh1 Rf8 23. Rxh5 Rf7 24. Kb1 Nc4 25. Qe1 Rg7 26. g4 Ne5 27. Rh2 Rf8 28. Ka1 Nc4 29. Rh5 Ne5 30. c3 Nxf3 31. Nxf3 Bxg4

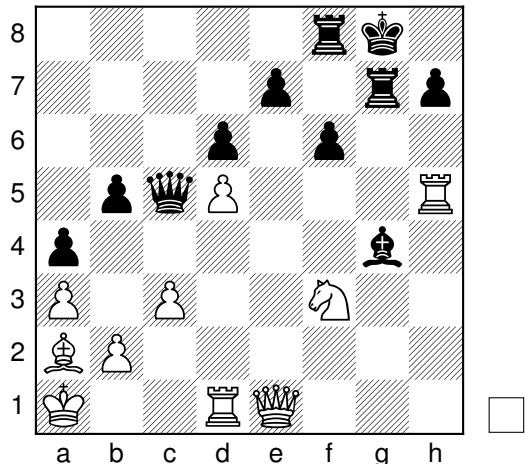

32. Qh1 **MAV–MCTS finds a beautiful geometry to secure a huge advantage.** Qe3

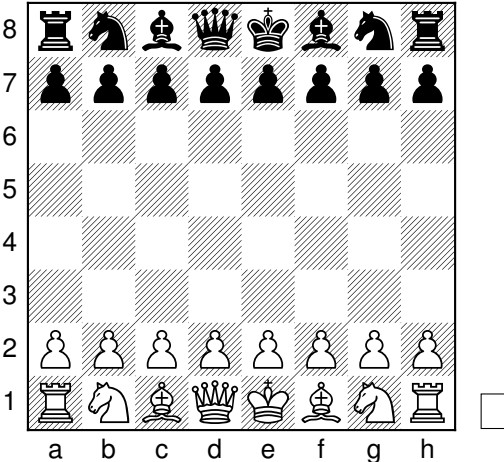

33. Rh3 **Another beautiful move by MAV–MCTS. Notice that a symmetric move Rf1 is the only alternative which secures the advantage.** Kh8 34. Re1 Qf4 35. Bb1 Bxh3 36. Qxh3 Qg4 37. Qh1 f5 38. Rg1 Qxf3 39. Qxf3 Rxg1 40. Qh5 Re1 41. Qh4 Re2 42. Bd3 Re5 43. Bxb5 Rf6 44. Qf4 Re4 45. Qf3 Rg4 46. Ka2 h5 47. Qe2 Rf7 48. Be8 Rfg7 49. Qe6 f4 50. Bf7 Kh7 51. Qf5+ R4g6 52. Qxh5+ Rh6 53. Qf5+ Kh8 54. Qc8+ Kh7 55. Qf5+ Kh8 56. Qxf4 Rf6 57. Qh4+ Rh7 58. Bh5 Kg7 59. Qg4+ Kh8 60. Qxa4 Kg8 61. Qe8+ Rf8 62. Qg6+ Kh8 63. a4 Rg8 64. Qf5 Rh6 65. a5 Rf6 66. Qh3 Kg7 67. Qe3 Kf8 68. a6 e5 69. a7 Ke7 70. Qb6 Rff8 71. Qc7+ Kf6 72. Qxd6+ Kf5 73. Qe6+ Kf4 74. d6 Kg5 75. Bf7 Ra8 76. Qxe5+ Kh4 77. Qh2+ Kg4 78. Qg2+ Kh4 79. Qh2+ Kg5 80. Qg3+ Kf5 81. Bxg8 Rxa7+ 82. Kb1 Ra8 83. d7 Ke4 84. Qg4+ Ke3 85. Qd4+ Kf3 86. Bd5+ Ke2 87. Bxa8 Kf1 88. b4 Ke2 89. b5 Ke1 90. Bg2 Ke2 91. d8=Q Ke1 92. Qe8# 1 − 0

# Game 20
### Endgame masterclass
**Date: 2024-12-01**
**White: MAV–MCTS($M = 2000$)**
**Black: Stockfish–L19**
**Result: 1 − 0**
**Online PGN: https://lichess.org/WjsKWo1E**

1. d4 d5 2. c4 c6 3. Nc3 e6 4. e3 f5 📚 5. Bd3 Nf6 6. Nge2 a5 7. O-O Bd6 8. c5 Bc7 9. f3 Nbd7 10. e4 dxe4 11. fxe4 fxe4 12. Nxe4 Nxe4 13. Bxe4 Qh4 14. Nf4 O-O 15. g3 Qe7 16. Bc2 Nf6

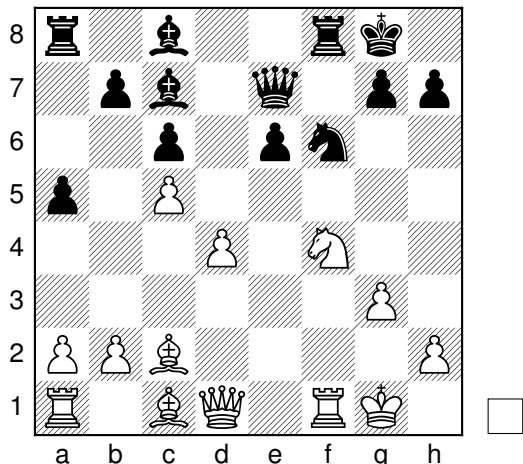

17. Nh5 **A strong move that secures the initiative.** e5 18. Bg5 a4 19. Qd3 e4 20. Qe3 Bh3 21. Nxf6+ gxf6 22. Bxf6 Rxf6 23. Qg5+ Qg7 24. Qxf6 Re8 25. Qxg7+ Kxg7

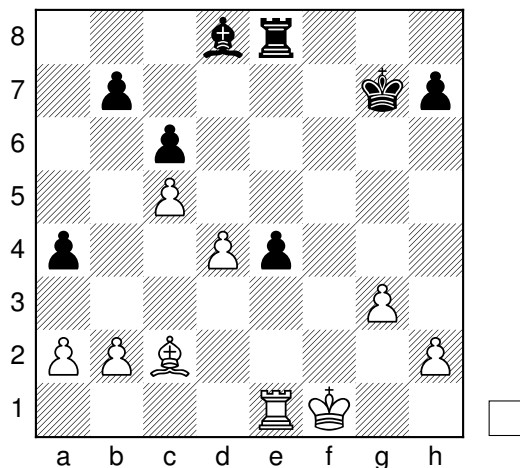

28. Rxe4 **MAV–MCTS decides to enter the opposite color bishops endgame which are notorious for their high margin of a draw.** Rxe4 29. Bxe4 Bf6

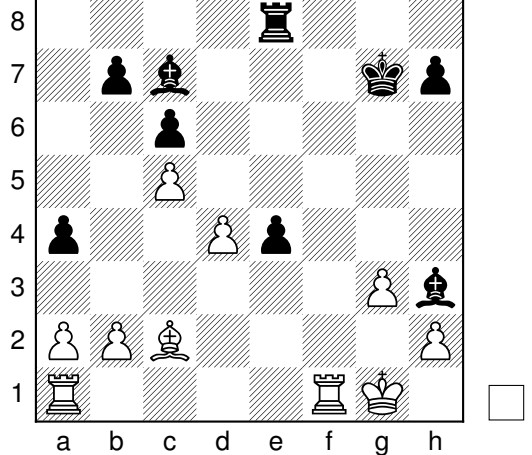

26. Rae1 **An unintuitive exchange sacrifice.** Bxf1 27. Kxf1 Bd8

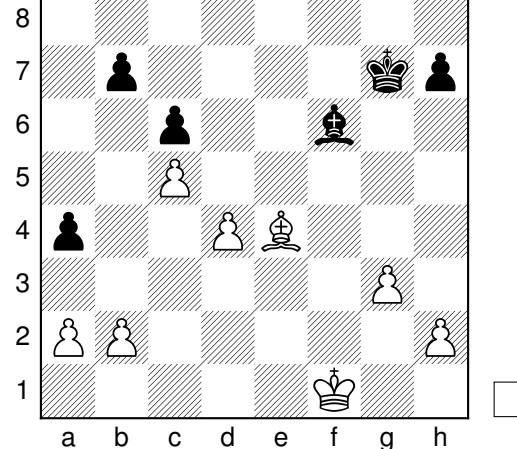

30. d5 **The pinnacle behind MAV–MCTS's play, and the only winning move! When sacrificing the exchange on move 26, MAV–MCTS correctly evaluated that the resulting semi-forcing sequence will result in the winning opposite color bishops endgame.** Bxb2 31. dxc6 bxc6 32. Bxc6 Ba3 33. Kg2 Bxc5 34. Bxa4 **It is known that the opposite-colored bishops endgames are winning if the distance between the passed pawns is four or more squares. MAV–MCTS proceeds to display the winning technique.** Kf6 35. Bc2 h6 36. Kf3 Bg1 37. a4 Bb6 38. Kg4 Bg1 39. h3 Bb6 40. Kh5 Kg7 41. Be4 Bf2 42. g4 Be1 43. Bg2 Bg3 44. a5 Be1 45. a6 Bf2 46. Be4 Bd4 47. Bd3 Bc5 48. Bc4 Be3 49. h4 Bc5 50. Bd3 Bg1 51. Bc2 Ba7 52. Be4 Bf2 53. Bb1 Bg1 54. Bc2 Bc5 55. Bb1 Bb6 56. Bc2 Bf2 57. Bb1 Bd4 58. Bc2 Bb6 59. Ba4 Bg1 60. Bd7 Be3 61. Ba4 Kf6 62. Bd1 Kf7 63. Ba4 Kf6 64. Bd1 Ke6 65. g5 Bxg5 66. hxg5 Kd6 67. Ba4 hxg5 68. Kxg5 Kc7 69. Kf5 Kb8 70. Ke4 Ka8 71. Ke5 Kb8 72. Bc2 Kc8 73. Kd6

Kb8 74. Bg6 Ka7 75. Bd3 Kb8 76. Bg6 Ka7 77. Bd3 Kb6 78. Bf1 Ka5 79. a7 Kb4 80. a8=Q Kc3 81. Qd5 Kb2 82. Qa5 Kb1 83. Qd2 Ka1 84. Bg2 Kb1 85. Bd5 Ka1 86. Qa2# 1 − 0

# Game 21
### A menace knight
**Date: 2024-12-01**
**White: Stockfish–L19**
**Black: MAV–MCTS($M = 2000$)**
**Result: 0 − 1**
**Online PGN: https://lichess.org/wuIGT2Q5**

1. d4 d5 2. c4 c6 3. Nc3 e6 4. e3 f5 📚 5. Qc2 Nf6 6. Bd3 a5 7. b3 Ne4 8. Bb2 Nd7 9. Nh3 Bd6 10. O-O Nxc3 11. c5 Bxh2+ 12. Kxh2 Ne4 13. Rfe1 Ndf6 14. Kg1 g5 15. f3 g4 16. Nf4 gxf3 17. gxf3 Rg8+ 18. Kf1 Ng3+ 19. Kg1 Ngh5+ 20. Kf1 Ng3+ 21. Kg1 Ngh5+ 22. Kf1 Nxf4 23. exf4 Kf7 24. Qh2 b6 25. Ke2 Ba6 26. Bxa6 Rxa6 27. a4 Ra8 28. Rg1 Rb8 29. Rxg8 Qxg8 30. Rg1 Qf8 31. Bc3 b5 32. Kd3 h5 33. Qg2 Qh6 34. axb5 Rxb5 35. Kc2 Rb8 36. Qg5 Qh7 37. Qh4 Rg8 38. Rg5 Ra8 39. Kb2 a4 40. b4 Qh6 41. Ka2 Rh8 42. Ka3 Qf8 43. b5 Qa8 44. Rg1 Qb7 45. Rb1 cxb5 46. Be1 Qc8 47. Rb2 Qa8 48. Bc3 Qa7 49. Rxb5 Rb8 50. Rxb8 Qxb8 51. Kxa4 Qb1 52. Qh2 Qd3 53. Qd2 **Even some extremely strong engines do not immediately realize that this is a losing move.**

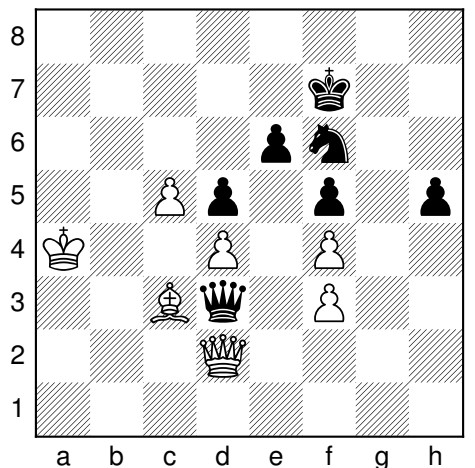

53...Qxd2 **MAV–MCTS correctly recognizes that the knight will dominate the bishop in this static pawn structure.** 54. Bxd2 h4 55. Be3 Ke8 56. Kb5 Kd7 57. Bf2 h3 58. Bg1 Kc7 59. Bh2 Nd7 60. Kb4 Nb8 61. Kc3 Kb7 62. Kd2 Kc6 63. Bg1 Kb5 64. Kc3 Ka4 65. c6 Nxc6 66. Kd2 Kb4 67. Kd3 Kb3 68. Bh2 Kb4 69. Bg3 Kb3 70. Bh2 Nb4+ 71. Ke3 Kc3 72. Ke2 Nc2 73. Kf2 Nxd4 74. Ke3 Nc2+ 75. Kf2 d4 76. Kg3 d3 77. Kxh3 d2 78. Bg1 Nd4 79. Kh4 d1=Q 80. Be3 Qe1+ 81. Kh5 Qxe3 82. Kh6 Qc1 83. Kg5 Kc2 84. Kf6 Qa3 85. Kg7 Qe7+ 86. Kh6 Kd2 87. Kh5 Qg7 88. Kh4 Ke1 89. Kh5 Kf2 90. Kh4 Qh6# 0 − 1

# Game 22
### Dominating the queen
**Date: 2024-12-01**
**White: MAV–MCTS($M = 2000$)**
**Black: Stockfish–L18**
**Result: 1 − 0**
**Online PGN: https://lichess.org/UMurXJiP**

1. d4 Nf6 2. c4 e6 3. Nf3 c5 4. d5 d6 5. Nc3 exd5 6. cxd5 g6 7. h3 Bg7 8. e4 O-O 9. Bd3 Re8 10. O-O Nbd7 📚 11. Bf4 c4 12. Bc2 Nc5 13. Nd2 Qd7 14. Nxc4 Ncxe4 15. Nxe4 Nxe4 16. f3 Nf6 17. Nxd6 Rd8 18. Bg3 Nh5 19. Bh2 Qc7 20. Ne4 Qb6+ 21. Kh1 f5 22. Ng3 Nf6 23. Bb3 Be6

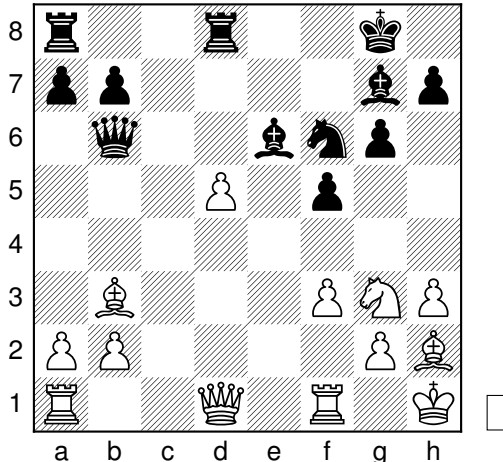

24. dxe6 **MAV–MCTS recognizes that the only way to fight for the advantage is by sacrificing the queen.** Rxd1 25. Raxd1 Re8 26. e7+ Kh8 27. Rfe1 Qb4 28. Nf1 f4 29. Bg1 h6 30. Re5 Rxe7

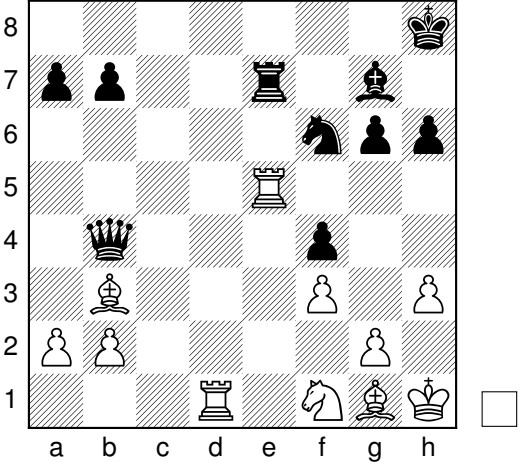

31. Rd4 **MAV–MCTS plays a zwischenzug that traps the opponent's queen in the middle of the board!** Qxb3

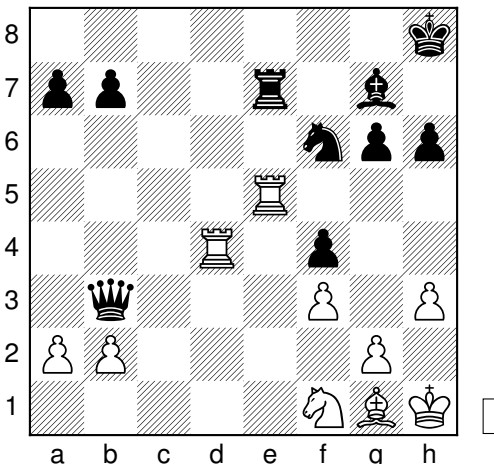

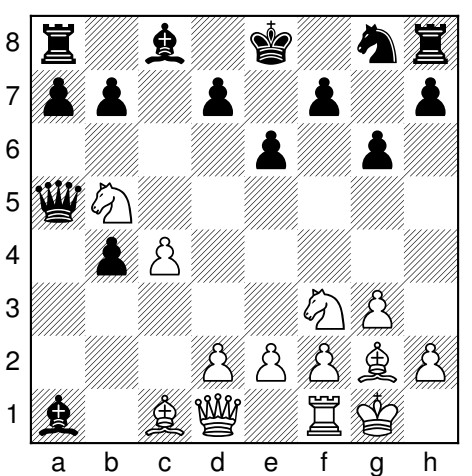

32. Rxe7 **MAV–MCTS correctly evaluates that taking the queen only leads to equality. Instead, by taking the rook, MAV–MCTS pieces continue to dominate Black's queen until the rest of the game.** Qg8 33. Nd2 g5 34. Ne4 Nd5 35. Rxb7 g4 36. hxg4 Qd8 37. Rd1 Qh4+ 38. Bh2 a6 39. Rc1 Ne7 40. b4 Bb2 41. Rd1 Bg7 42. b5 axb5 43. Rxb5 Ng8 44. Rh5 Qe7 45. Bxf4 Kh7 46. Rd6 Bb2 47. Bxh6 Nxh6 48. Rdxh6+ Kg8 49. Rg6+ Kf8 50. Rf5+ Ke8 51. Nd6+ Kd8 52. Rg8+ Kc7 53. Rf7 **The queen is gone.** Kxd6 54. Rxe7 Kxe7 55. g3 Ba3 56. Kg2 Kf7 57. Rb8 Kf6 58. Rb5 Ke7 59. Kh3 Kd7 60. Rb7+ Kc6 61. Rb3 Be7 62. f4 Bd8 63. g5 Ba5 64. Kg4 Kd5 65. Rb5+ Ke6 66. Rxa5 Kd6 67. Ra7 Ke6 68. Rb7 Kd6 69. g6 Ke6 70. a4 Kd6 71. g7 Kd5 72. a5 Kd6 73. g8=Q Kc5 74. f5 Kd4 75. Qg5 Kc5 76. Qe3+ Kc6 77. Rh7 Kb5 78. Rc7 Ka4 79. Rb7 Kxa5 80. Qa3# 1 − 0

## Game 23
### Underpromotion for the win
**Date: 2024-11-14**
**White: Stockfish–L17**
**Black: MAV–MCTS**($M = 2000$)
**Result: 0 − 1**
**Online PGN: https://lichess.org/GzFmze7T**
1. c4 c5 2. g3 g6 3. Bg2 Bg7 4. Nc3 Nc6 5. a3 e6 6. b4 Nxb4 7. axb4 cxb4 8. Nb5 📚 Bxa1 9. Nf3 Qa5 10. O-O

10...a6 **MAV–MCTS finds the only move which secures a sizable advantage in a very irrational position.** 11. Nd6+ Ke7 12. c5 Qxc5 13. d4 Qxd6 14. Bf4 Qb6 15. Qxa1 d5 16. e4 Nf6 17. Rc1 Bd7 18. Ne5 Rhd8 19. Rc5 Nh5 20. Qc1 Rac8 21. exd5 exd5 22. Nxd7 Kxd7 23. Bd2 b3 24. Qb2 Rxc5 25. dxc5 Qxc5 26. Qxb3 Kc8 27. Qb2 Qb5 28. Qa1 d4 29. Bf1 Qc5 30. Qa2 Rd7 31. Bh3 f5 32. g4 Nf6 33. gxf5 Kb8 34. fxg6 Rg7 35. Qb3 Rxg6+ 36. Kf1 Ne4 37. Bf4+ Ka8 38. Be6 d3 39. Be3 Qxe3 40. fxe3 Nd2+ 41. Kf2 Nxb3 42. Bxb3 Rb6 43. Bc4 Rb2+ 44. Ke1 d2+ 45. Ke2 a5 46. Bd3 a4 47. Bxh7 a3 48. Bg8 a2 49. Bxa2 d1=Q+ 50. Kxd1 Rxa2 51. Kc1 Rxh2 52. Kb1 Ka7 53. Kc1 b5 54. Kd1 Rg2 55. Ke1 Ra2 56. e4 Kb7 57. e5 Kc7 58. Kf1 Kd7 59. e6+ Kxe6 60. Kg1 b4 61. Kf1 Rc2 62. Ke1 Kd5 63. Kd1 b3 64. Ke1 Kc6 65. Kd1 Kd5 66. Ke1 b2 67. Kf1

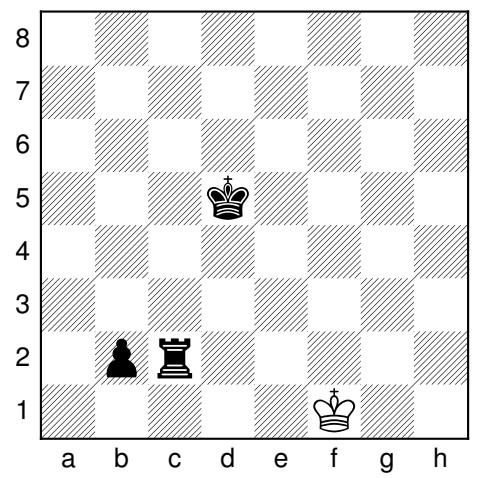

67...b1=R#

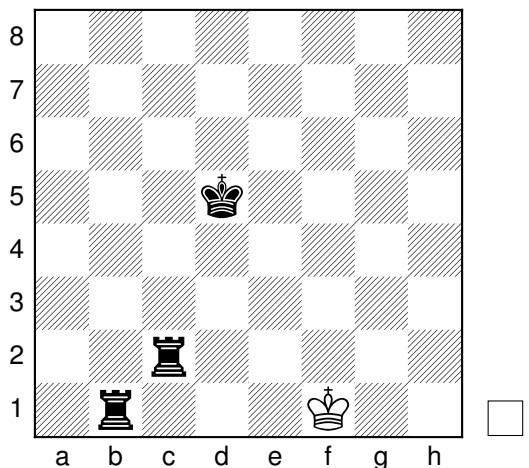

**MAV–MCTS earns bonus points for finishing the game in style!** 0 – 1

### F.2. Chess960

Each game starts with a randomly sampled initial position from a set of chess960 legal permutations. Once the game is finished, the agents reverse the colors and play the game with the same initial position. In the header of each game we show the starting order of pieces. A square next to the diagrams represents a side to play. All games use Stockfish with 2 seconds of thinking time per move. We use MAV–MCTS to represent the external search MAV with mean scoring.

# Game 24
### Setting the board on fire
**Date: 2024-11-14**
**White: MAV–MCTS($M = 100$)**
**Black: Stockfish–L19**
**Result: 1/2 – 1/2**
**Online PGN: https://lichess.org/0cs1dk1D**

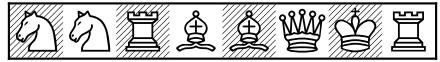

1. d4 d5 2. c4 dxc4 3. Rxc4 Nc6 4. Nb3 e5 5. dxe5 Nxe5 6. Rc1 c5 7. f4

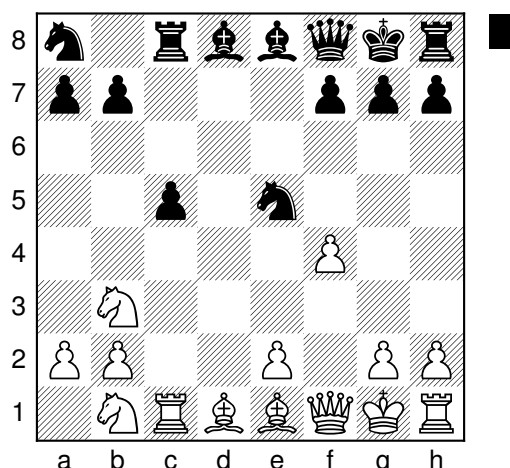

7...c4 **Instead of retreating, Black counterattacks MAV–MCTS's knight and the craziness starts.** 8. fxe5

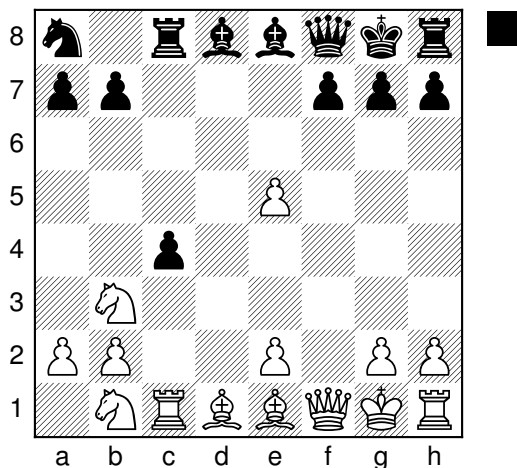

8...cxb3 **Black sacrifices the rook!** 9. Rxc8 Bb6+ 10. Bf2

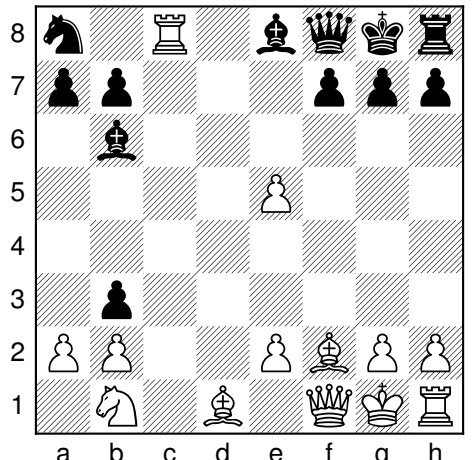

10...bxa2

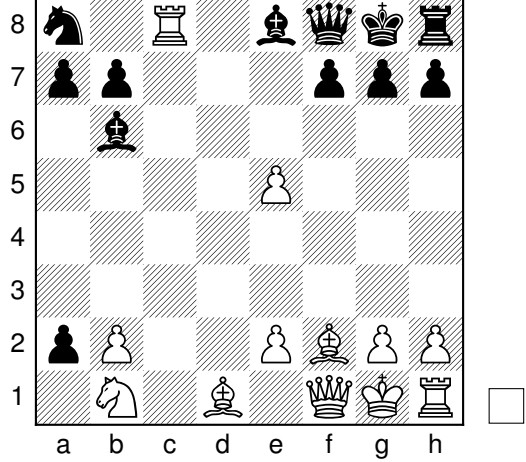

**MAV–MCTS cannot prevent Black's pawn from promoting on move 10! But...** 11. Ba4 a1=Q 12. Bxe8 Bxf2+ 13. Kxf2 Qa5

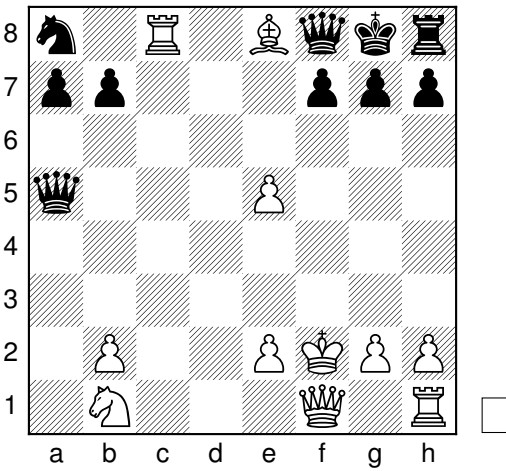

**Despite being a queen up, Black is not better!** 14. Qc1 Nb6 15. Qc7 Nxc8 16. Qxa5 Qxe8 17. Rd1 b6 18. Qb5 O-O 19. Qxe8 Rxe8 20. Rd7 a5 21. Na3 Kf8 22. Nc4 Re7 23. Rd8+ Re8 24. Rd7 Re6 25. Rc7 Ne7 26. Rb7 Ng6 27. Rb8+ Ke7 28. Rb7+ Kd8 29. Rxf7 Nxe5 30. Nxe5 Rxe5 31. Rxg7 h5 32. h4 Rf5+ 33. Ke3 Re5+ 34. Kf3 Ke8 35. Rg5 Rxg5 36. hxg5 Kf7 37. Ke4 b5 38. Kd5 b4 39. Kc4 Kg6 40. Kb5 b3 41. Kxa5 Kxg5 42. Ka4 h4 43. Kxb3

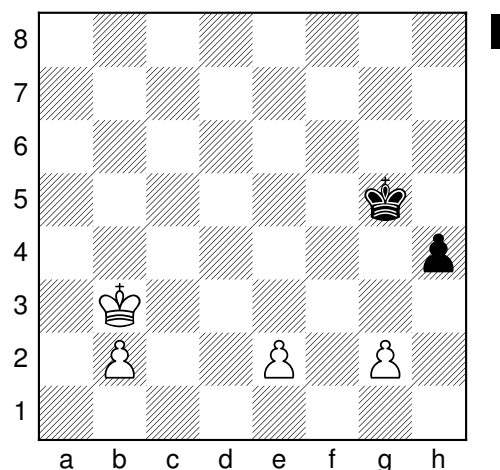

**MAV–MCTS must settle for a draw despite being two pawns up in the pawn endgame!** 43...Kf4 44. Kc2 Ke3 45. b4 Kxe2 46. b5 Kf1 47. b6 Kxg2 48. b7 h3 49. b8=Q h2 50. Qb7+ Kg1 51. Qb1+ Kg2 52. Qb7+ Kg1 53. Qb1+ Kg2 54. Qb7+ (threefold repetition). **After all the craziness the game ends peacefully.** 1/2 – 1/2

## Game 25
### Accelerated Benko gambit
**Date: 2024-11-14**
**White: Stockfish–L20**
**Black: MAV–MCTS($M = 2000$)**

**Result: 1/2 – 1/2**
**Online PGN: https://lichess.org/yJXI9zcc**

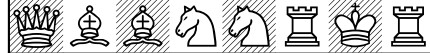

1. d4 h5 2. c4

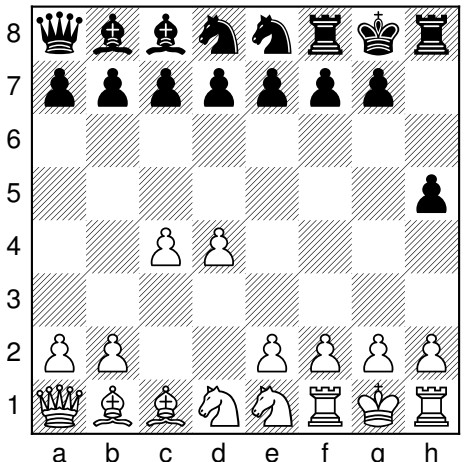

2...b5 **Even Stockfish gives the same evaluation as for the traditional Benko gambit.** 3. b3 Nf6 4. h4 Qb7 5. Nf3 c6 6. e4 d6 7. Ne3 a6 8. Re1 Ng4 9. e5 dxe5 10. Nxe5 Nxe5 11. dxe5 g6 12. O-O Ba7 13. f4 bxc4 14. Kh1 cxb3 15. axb3 Ne6 16. f5 **The position seems dire for MAV–MCTS.**

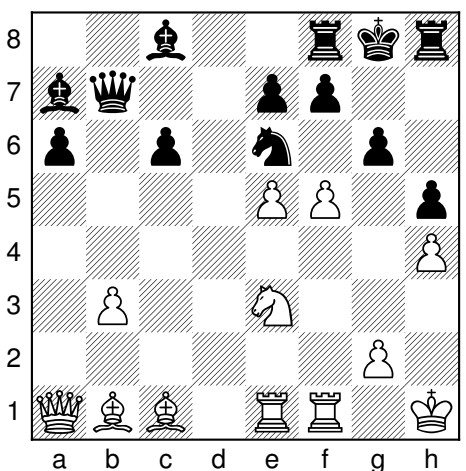

16...Qb4 **But, MAV–MCTS finds astonishing resources to keep the game alive!** 17. Qa4 Qxa4 18. bxa4 Bxe3 19. Rxe3 gxf5 20. Bxf5 Rd8 21. Rg3+ Ng7 22. e6 Kf8 23. Bg6 Bxe6 24. Bb2 f6 25. Ba3 f5 26. Re1 Rb8 27. Kg1 Rb7 28. Rge3 Rh6 29. Rxe6 Nxe6 30. Rxe6 Kg7 31. Bxf5 Rxe6 32. Bxe6 Kf6 33. Bc4 Rb1+ 34. Kf2 e5 35. Bd6 Ra1 36. Bb3 Rb1 37. Bg8 Ra1 38. Bb3 Rb1 39. Bg8 Ra1 40. Bb3

(threefold repetition). **After surviving some terrifying moments in the middlegame, MAV–MCTS lives to fight another day.** 1/2 – 1/2

# Game 26
**Delayed Benko gambit**
**Date: 2024-11-14**
**White: Stockfish–L19**
**Black: MAV–MCTS($M = 2000$)**
**Result: 1/2 – 1/2**
**Online PGN: https://lichess.org/MBeKZuLV**

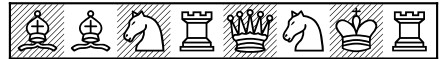

1. b3 c5 2. e4 h5 3. Ne3 b5 4. c4

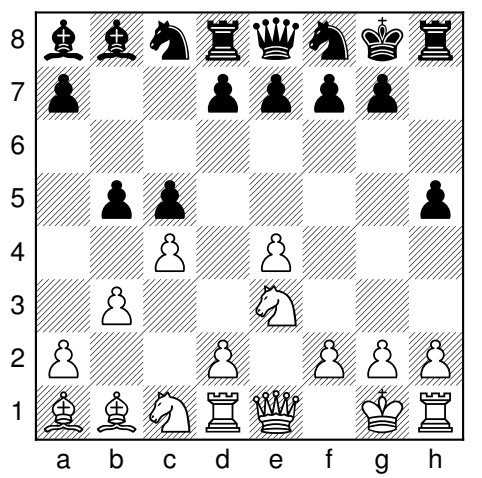

4...h4 **Similar to Game 2, MAV–MCTS pushes b and h pawns.** 5. O-O e5 6. Nd5 Nb6 7. Bd3 Ne6 8. Qe3 Qf8 9. Ne2 Re8 10. Rde1 Rh6 11. Qf3 Nd4 12. Qh3 Bb7 13. Ne3 bxc4 14. bxc4 Bc8 15. Nxd4 exd4 16. Ng4 Rh8 17. f4 d6 18. f5 Ba6 19. f6 Bc8 20. e5 Rxe5 21. Qf3 g6 22. Bb2 Rg5 23. h3 Bxg4 24. hxg4 Nd7 25. Qb7 Ne5 26. Be2 h3 27. g3 d5 28. d3 Nxg4 29. Bxg4 Rxg4 30. Qd7 Rxg3+ 31. Kh1

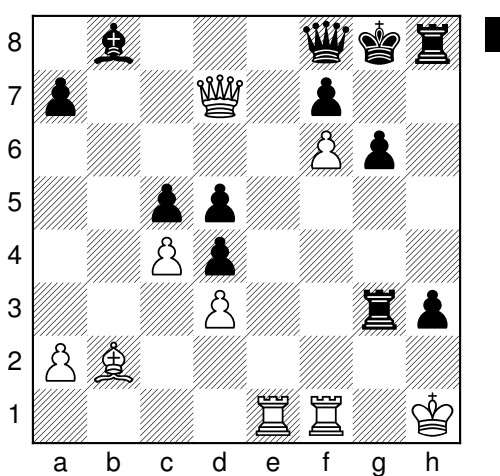

31...Rg2 **MAV–MCTS again finds a saving resource in a difficult position.** 32. Re8 Rh2+ 33. Kg1 Rg2+ 34. Kh1 Rh2+ 35. Kg1 Rg2+ 36. Kh1 (threefold repetition). 1/2 – 1/2

# Game 27
**Not fearing ghosts**
**Date: 2024-11-14**
**White: MAV–MCTS($M = 1000$)**
**Black: Stockfish–L19**
**Result: 1 – 0**
**Online PGN: https://lichess.org/4EMehaRm**

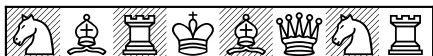

1. c4 f6 2. f4 c6 3. Nb3 O-O-O 4. d4 Bh5 5. Bb4 g5 6. f5 Qg7 7. Bd3 Bf4 8. Rc3 Nh6 9. g3 Ng4 10. Ke1 Bb8 11. Kd2 e5 12. d5 e4 13. Bxe4 Rhe8 14. Bf3 Be5 15. Rc2 Ne3

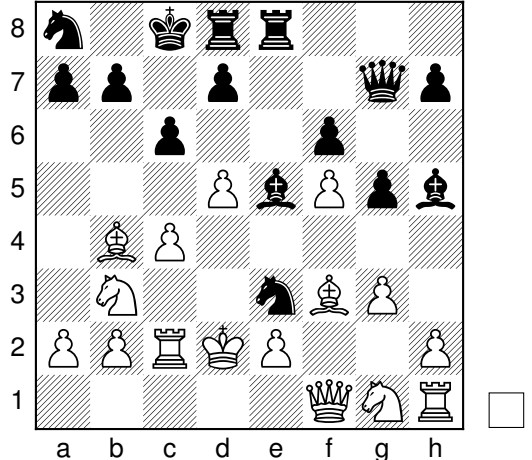

16. Kxe3 **MAV–MCTS correctly evaluates that it is safe**

to capture the knight and to expose its king to a various discovered attacks in the center of the board. Bd6+ 17. Kf2 Bxf3 18. Nxf3 Bxb4 19. c5 Nc7 20. a3 Qe7 21. Qc1 Nxd5 22. axb4 Nxb4 23. Rd1 Qe4 24. Nbd4 Nxc2 25. Qxc2 Qxc2 26. Nxc2 Re4 27. Nd2 Ra4 28. b3 Ra5 29. Ne4 d5 30. Nd6+ Kc7 31. Nd4 Rxd6 32. cxd6+ Kd7 33. g4 Ra6 34. h4 gxh4 35. g5 fxg5 36. f6 Ke8 37. Ne6 b5 38. Nc7+ Kd7 39. f7 Kxd6 40. Nxd5 c5 41. f8=Q+ Ke5 42. Qe8+ Re6 43. Qh8+ Kf5 44. Qxh7+ Ke5 45. Qg7+ Ke4 46. Qh7+ Ke5 47. Qg7+ Kf5 48. e4+ Kg4 49. Rg1+ Kh5 50. Rxg5# **Once MAV–MCTS obtained the advantage in the middlegame, it never let it slip.** 1 – 0

# Game 28
**Finishing games in style**
**Date: 2024-11-14**
**White: Stockfish–L18**
**Black: MAV–MCTS($M = 100$)**
**Result: 0 – 1**
**Online PGN: https://lichess.org/kaZI3lB6**

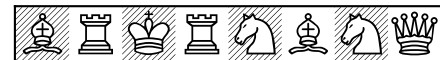

1. e4 g5 2. g3

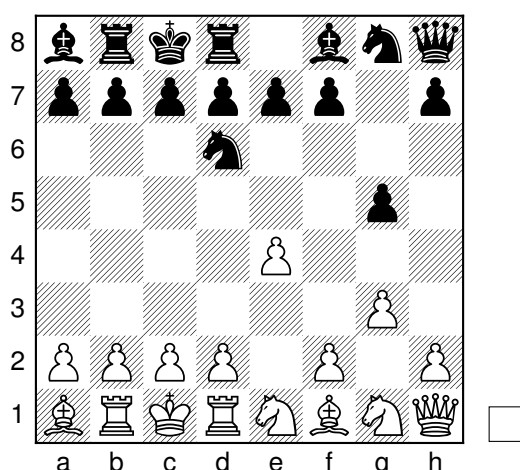

2...Nd6 **A move that breaks a traditional wisdom not to block central pawns with pieces.** 3. Bg2 f5 4. exf5 Bg7 5. Nh3 g4 6. Nf4 h5 7. Ned3 Nxf5 **MAV–MCTS finds a stable outputs for the knight.** 8. h3 e6 9. hxg4 hxg4 10. Nh5 Bd4 11. a4 Nf6 12. Nhf4 e5 13. Ne2 d6 14. a5 Qxh1 15. Rxh1 b6 16. Bxa8 Rxa8 17. c3 e4 18. cxd4 exd3 19. Nf4 Kd7 20. b4 Rh8 21. Re1 Rae8 22. Rxe8 Rxe8 23. axb6 Re1+ 24. Kb2 Rxb1+ 25. Kxb1 cxb6 26. Nxd3 Kc6 27. Kc2 Ne4 28. Kd1 Kb5 29. d5 Kc4 30. Ke2 Ng5 31. Bc3 Ne4 32. Bb2 Ng5 33. Ne1 Kxb4 34. Kd3 Kc5 35. Bc3 Kxd5 36. Bc3 Kxd5 37. Bh8 Ng5 38. Ng2 Ne6 39. Nh4 Nxh4 40.

gxh4 Nf4+ 41. Kc2 Ke4 42. Bf6 b5 43. Kb3 a5 44. Be7 d5 45. Kc2 a4 46. Bd6 d4 47. d3+ Nxd3 48. h5 Ne1+ 49. Kd2 Nf3+ 50. Kc2 Kf5 51. h6 Kg6 52. h7 Kxh7 53. Kd3 Kg8 54. Bb4 Ne5+ 55. Kxd4 Nc6+ 56. Kc3 Nxb4 57. Kxb4 Kg7 58. Kc3 Kf6 59. Kb4 Kg5 60. Kc3 a3 61. Kc2 b4 62. Kb3 Kf4 63. Ka2 Kf3 64. Kb1 b3 65. Kc1 Kxf2 66. Kd1 a2 67. Kd2 a1=Q 68. Kd3 g3 69. Kc4 Qe5 70. Kb4 Qg7 71. Kc4 b2 72. Kb4 g2 73. Kc4 b1=Q 74. Kc5 Qd7 75. Kc4

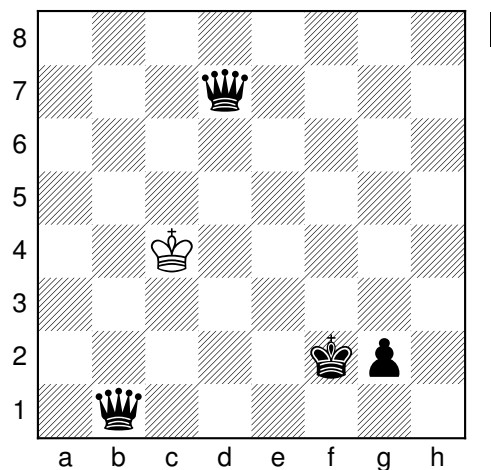

75...g1=R **Like in the traditional chess game 23, MAV–MCTS once again shows its preference for rook underpromotions.** 76. Kc3 Rc1# 0 − 1

# Game 29
### A knight on the rim...
**Date: 2024-11-14**
**White: MAV–MCTS($M = 1000$)**
**Black: Stockfish–L17**
**Result: 1 − 0**
**Online PGN: https://lichess.org/zurqClYv**

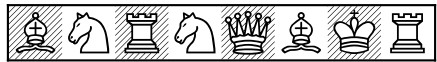

1. d4 d5 2. h4 b6 3. h5 Nd7 4. c4 e6 5. cxd5 exd5 6. h6 g6

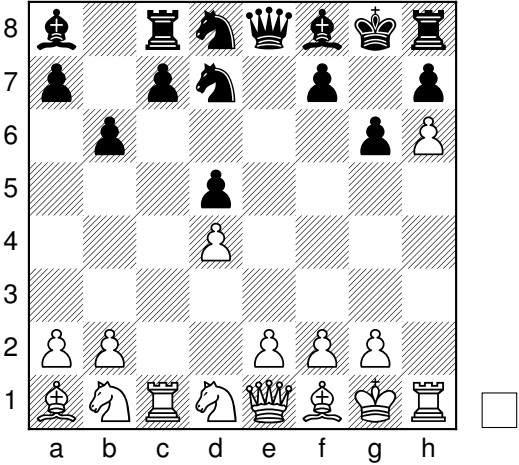

7. e4 **MAV–MCTS pushes a pawn on the most protected square on the board to secure the piece activity.** Qxe4 8. Qxe4 dxe4 9. Nbc3 Bd6 10. Ne3 Bb7 11. Nb5 Bf4 12. g3 Bxe3 13. fxe3 Ne6 14. b3 O-O 15. a4 a6

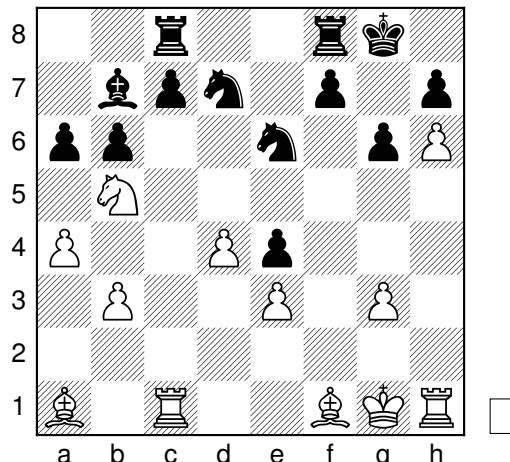

16. Na7 **Not your everyday knight route.** Rce8 17. Nc6 Nf6 18. Ne5 Rd8 19. Bc4 Ng5 20. Kf2 c6 21. Bb2 b5 22. Be2 Nd5 23. Nxc6 Rd6 24. Ne5 Rf6+ 25. Ke1 Rd8 26. axb5 axb5 27. Rc5 Nxe3 28. Rxb5 Ba8 29. Kd2 Nf5 30. Ng4 Rfd6 31. Rc1 e3+ 32. Ke1 Bf3 33. d5 Bxg4 34. Bxg4 Nxh6 35. Be2 Ne4 36. Be5 R6d7 37. Rb4 Nd6 38. Bf6 Re8 39. g4 Kf8 40. Rb6 Ng8 41. Bb2 Red8 42. Ba3 Kg7 43. Bf3 Ne7 44. Bb2+ Kf8 45. Ba3 Kg8 46. Rc3 h5 47. gxh5 g5 48. Rxe3 Nxd5 49. Bxd5 Nf5 50. Bc6 Nxe3 51. Bxd7 Nc2+ 52. Kd2 Nxa3 53. Rd6 Rb8 54. Ba4 Nb1+ 55. Kc2 Na3+ 56. Kc1 Kh8

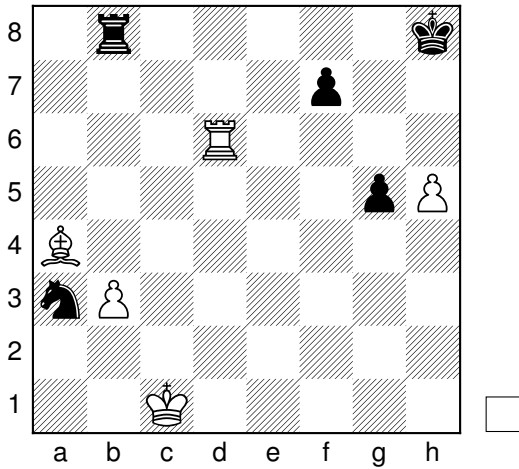

**57. Rd5 The Black's knight is trapped on the edge of the board!** f5 58. Rxf5 g4 59. Rg5 Rg8 60. Rxg8+ Kxg8 61. Kd2 Kf8 62. Ke3 Kg7 63. Kf4 Nc2 64. Kxg4 Kf6 65. Be8 Ne3+ 66. Kf4 Nd5+ 67. Ke4 Nc7 68. Bg6 Na6 69. Kd4 Kg5 70. Kc4 Kh6 71. Kb5 Nc7+ 72. Kc6 Na6 73. Kb5 Nc7+ 74. Kc5 Kg7 75. Bd3 Kh6 76. Kd6 Ne8+ 77. Ke7 Ng7 78. Bg6 Nxh5 79. Bxh5 Kg5 80. Bd1 Kh4 81. Ke6 Kg5 82. Ke5 Kg6 83. b4 Kg5 84. b5 Kh6 85. Kf6 Kh7 86. b6 Kg8 87. b7 Kh7 88. b8=R Kh6 89. Rh8#  1 − 0

# Game 30
### Complications never end
**Date: 2024-11-14**
**White: MAV–MCTS($M = 100$)**
**Black: Stockfish–L19**
**Result: 1/2 – 1/2**
**Online PGN: https://lichess.org/iSEAtCyD**

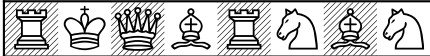

1. e4 Nhg6 2. d4 c5 3. dxc5 Qxc5 4. Nhg3 Qa5 5. c3 Bc7 6. Ne3 Nf4 7. Bc2 e6 8. a4 O-O-O 9. f3 Qa6 10. Rd1 f5 11. a5 d5 12. exd5 g6 13. Ba4 Re7 14. Nc2 Nxd5 15. Na3 e5 16. Nb5 Ne6 17. Nxa7+ Kb8 18. Bb5

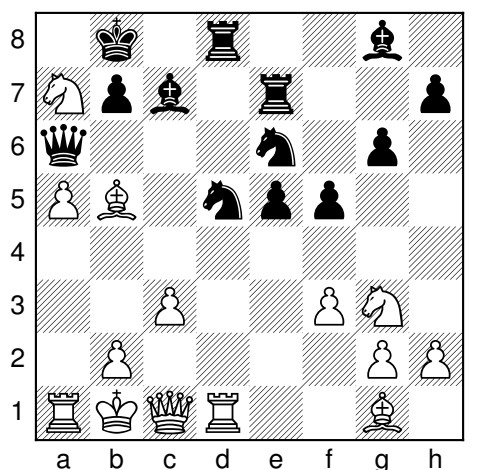

**Black's queen is nearly trapped.** 18...Qd6 19. Ne2 Nc5 20. Ra3 **MAV–MCTS finds the time for a prophylactic move in the middle of the complications.** e4 21. fxe4 fxe4 22. a6 Bb6 23. axb7 Bxa7 24. c4 Rxb7

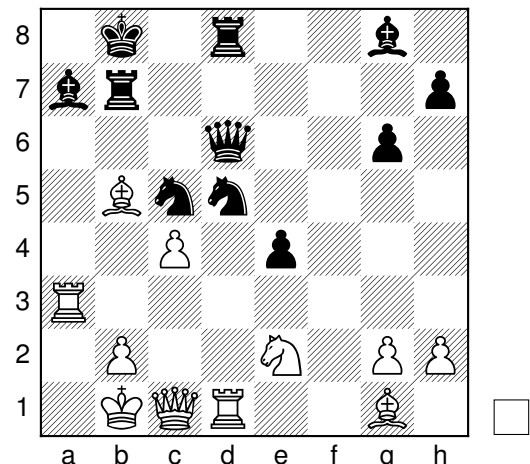

**25. Nd4 MAV–MCTS adds fuel to the fire.** Rc7 26. cxd5 Bxd5 27. Qg5 Bb6 28. Be3 Kb7 29. Bf4 Qe7 30. Qxe7 Rxe7 31. Bg5 Red7 32. Bxd8 Rxd8 33. b4 Nd3 34. Nc2 Nf2 35. Ba6+ Kc7 36. Rd2 e3 37. Nxe3 Be4+ 38. Kc1 Nd3+ 39. Bxd3 Bxe3 40. Kc2 Bxd2 41. Bxe4 **The dust has finally settled.** Bxb4 42. Ra7+ Kb6 43. Rxh7 g5 44. Rb7+ Ka5 45. Bc6 Rd2+ 46. Kb3 Rd3+ 47. Kc4 Rc3+ 48. Kd5 Ba3 49. Rb5+ Ka6 50. Rb1 g4 51. Bd7 Rc2 52. Ra1 Rd2+ 53. Ke5 Kb6 54. Rxa3 Rxg2 55. Rb3+ Kc7 56. Ba4 Rxh2 57. Rc3+ Kb8 58. Kd6 Rd2+ 59. Kc6 Rd8 60. Rb3+ Kc8 61. Bb5 Rd1 62. Re3 Rc1+ 63. Kb6 Rd1 64. Rc3+ Kb8 65. Re3 Rc1 66. Re7 Rc2 67. Re1 Rc3 68. Re4 Rc1 69. Bc4 Rb1+ 70. Bb5 Rc1 71. Bc4 Rb1+ 72. Bb5 Rc1 (threefold repetition). **MAV–MCTS decides to call it a day.** 1/2 − 1/2

## Game 31
### Only active pieces count

**Date: 2024-11-14**
**White: Stockfish–L20**
**Black: MAV–MCTS($M = 100$)**
**Result: 1/2 – 1/2**
**Online PGN: https://lichess.org/Nwy9BFqS**

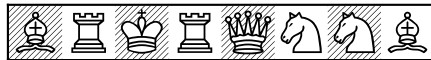

1. e4 e5 2. b3 b6 3. d3 d6 4. f4 exf4 5. Ne2 Ne6

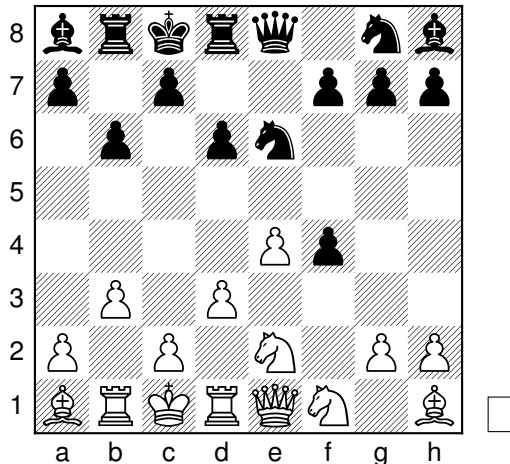

6. g3 **MAV–MCTS sacrifices a pawn for a rapid development.** fxg3 7. Qxg3 Ne7 8. Ne3 d5 9. exd5 Nxd5 10. Bxd5 Bxd5 11. Rf1 Bb7 12. Kd2 Rd7 13. Rbe1 g6 14. Bxh8 Qxh8 15. Nc3 O-O-O 16. Ng4 Qg7 17. Nf6 Rd4 18. Nb5 Rb4

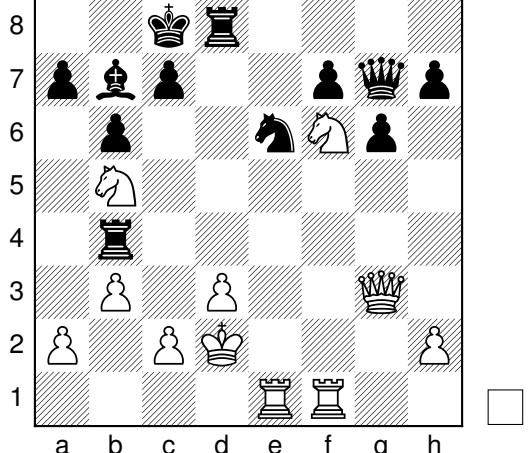

19. Nxc7 **A stunning piece sacrifice that is hard to grasp.** Nxc7 20. Re7 Qh6+ 21. Kd1 Ne6

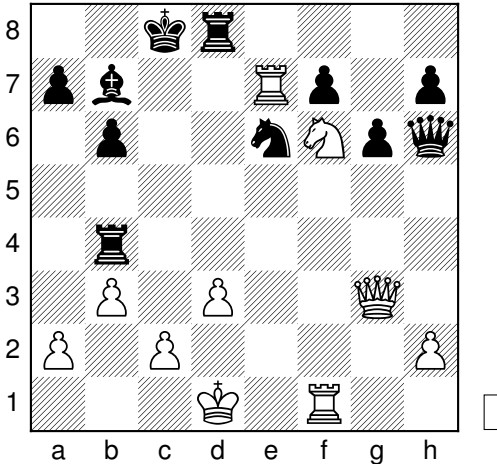

22. c4 **MAV–MCTS point is finally revealed – the Black's rook is isolated from the rest of the board!** Qg5 23. Qxg5 Nxg5 24. Kc2 Nf3 25. Kc3 a5 26. Rxb7 Nxh2 27. Rf2 Kxb7 28. Rxh2 b5 29. a3

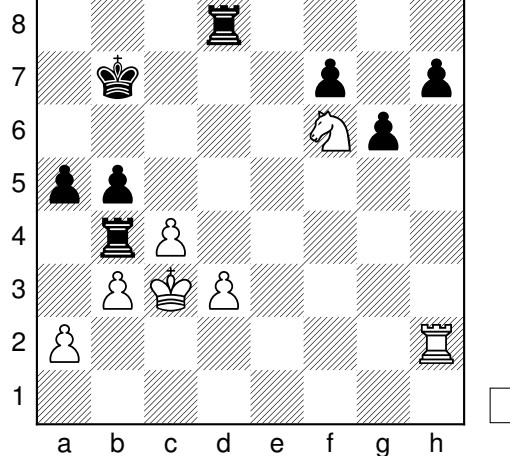

**The Black's rook gets trapped by a pawn.** 29...Rxd3+ 30. Kxd3 Rxb3+ 31. Kd4 bxc4 32. Rxh7 Kc6 33. Ne4 Rd3+ 34. Kxc4 Rxa3 35. Nc3 Ra1 36. Rxf7 Kd6 37. Rf6+ Ke5 38. Rxg6 Kf5 39. Rg8 Ke5 40. Rg4 Ke6 41. Rh4 a4 42. Rh2 Rg1 43. Ne2 Rb1 44. Rh5 a3 45. Ra5 Ra1 46. Nd4+ Kd6 47. Rd5+ Ke7 48. Kb3 a2 49. Nc6+ Ke6 50. Rh5 Rg1 51. Kxa2 Rg2+ 52. Ka3 Kf6 53. Rh1 Rg3+ 54. Ka4 Kf5 55. Ka5 Ra3+ 56. Kb6 Ra8 57. Rh5+ Ke4 58. Rh6 Ra1 59. Rh8 Rb1+ 60. Ka6 Rb2 61. Na7 Ra2+ 62. Kb7 Rb2+ 63. Kc8 Rb1 64. Rh6 Kf4 65. Ra6 Rh1 66. Kd8 Ke3 67. Ra5 Rh7 68. Ra1 Kd3 69. Nc8 Kc3 70. Rg1 Rh5 71. Kd7 Rh7+ 72. Ke6 Kc2 73. Rg8 Kd2 74. Rg5 Rh6+ 75. Kd7 Rh7+ 76. Kd6 Rh4 77. Rg8 Ke3 78. Na7 Kf4 79. Kc6 Rh6+ 80. Kb5 Rh1 81. Kc6 Rh6+ 82. Kb5 Ke4 83. Kb4 Kd5 84. Nc8 Rh1 85. Rg5+ Kd4 86. Kb5 Rh8 87. Na7 Ke3 88. Kb6 Rh1 89. Rg8 Ra1 90. Rg4 Rc1 91. Rg6 Rb1+ 92. Kc6 Ra1 93. Rg7 Ke2 94. Rg2+ Kf1 95. Rg7 Rb1 96. Rg4 Rc1+

97. Kb7 Rb1+ 98. Ka6 Ra1+ 99. Kb5 Rxa7 100. Rg6 Ra8
101. Kc5 Kf2 102. Rb6 Ke3 103. Rb1 Ra5+ 104. Kd6 Ke4
105. Rb4+ Kf5 106. Rb6 Ra1 107. Kd5 Rd1+ 108. Kc4
Ra1 109. Kd4 Ra4+ 110. Kd5 Ra5+ 111. Kd4 Ra1 112.
Rh6 Ra4+ 113. Kd3 Kg5 114. Rh7 Ra1 115. Ke4 Kg6 116.
Rh2 Re1+ 117. Kd3 Ra1 118. Ke4 Kf7 119. Rh7+ Kg6
120. Rh2 (threefold repetition). **An extraordinary game
despite the outcome!** 1/2 – 1/2

## Game 32
**Returning knight**
Date: 2024-11-14
White: Stockfish–L19
Black: MAV–MCTS($M = 500$)
Result: 1/2 – 1/2
Online PGN: https://lichess.org/4VVNz4wX

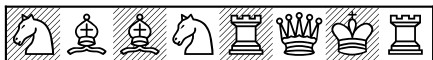

1. e4 e5 2. f4 exf4 3. Qxf4 d5 4. O-O Ne6 5. Qh4 d4 6. c3
c5 7. b4 dxc3 8. dxc3 Nb6 9. Ne3 cxb4 10. cxb4 Qxb4 11.
Nb3 h5 12. Qf2 f6 13. e5 Bxe5 14. Nf5 Nc4 15. Bd3 Bd7
16. Bxc4 Qxc4 17. Rxe5 fxe5 18. Nd6 Qg4 19. Qf7+ Kh7

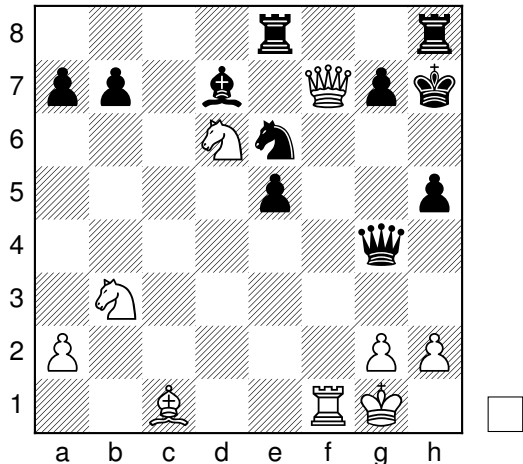

20. Ne4 **White finds a cute resource to force the draw.**
Qxe4 21. Qxh5+ Kg8 22. Qf7+ Kh7 23. Qh5+ Kg8 24.
Qf7+ Kh7 25. Qh5+ (threefold repetition). 1/2 – 1/2

## Game 33
**Harrys' deadlock**
Date: 2024-11-14
White: Stockfish–L19
Black: MAV–MCTS($M = 1000$)
Result: 0 – 1
Online PGN: https://lichess.org/yBlOcUgM

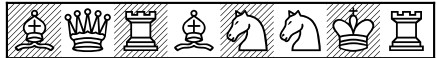

1. h4 h5

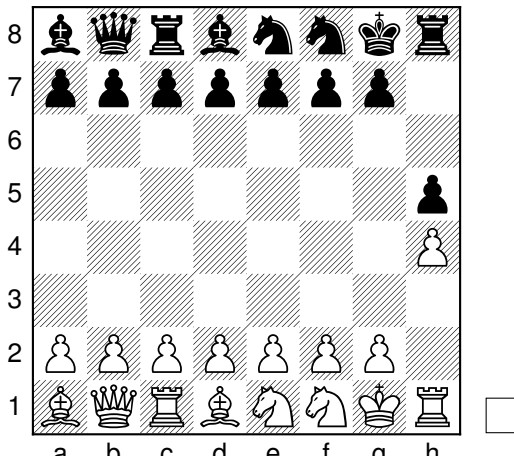

**Pushing h-pawns is one of the best ways to start the
game with this piece configuration according to strong
engines.** 2. b4 b5 3. d3 a5 4. c3 e6 5. Nd2 Ng6 6. e3 axb4
7. cxb4 Bxh4 8. Rh3 Qb6 9. g3 Bg5 10. Bd4 Qd6 11. Rxh5
e5 12. Ba1 Rxh5 13. Bxh5 Qe6 14. Bf3 Bxf3 15. Nexf3
Be7 16. Qb2 Ra8 17. Rc2 Qd5 18. Qb3 Qxb3 19. Nxb3 f6
20. Bc3 Ra4 21. d4 Bxb4 22. dxe5 Bxc3 23. Rxc3 Rxa2 24.
Rc5 Nxe5 25. Nxe5

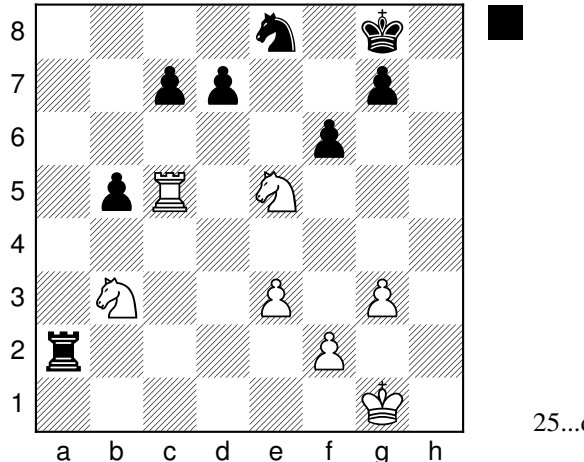

25...d6

**This zwischenzug, instead of immediately recapturing
the knight, is the only way to secure the advantage.** 26.
Rxb5 fxe5 27. f4 exf4 28. exf4 c5 29. Na5 Kf7 30. Rb7+
Ke6 31. Nc4 Kd5 32. Ne3+ Ke4 33. Re7+ Kf3 34. Nf5
Ra1+ 35. Kh2 Nf6 36. Nh4+ Kf2 37. Nf5 d5 38. Rc7 d4
39. Rxc5 d3 40. Rc3 d2 41. Rc2 Ne4 42. Nxg7 Kf3 43.
Rxd2 Nxd2 44. Nf5 Nf1+ 45. Kh3 Ne3 46. Kh4 Nxf5+ 47.

Kg5 Nxg3 48. Kf6 Kxf4 49. Ke7 Nf5+ 50. Kd7 Rg1 51.
Kc7 Ke5 52. Kc6 Ne7+ 53. Kc5 Rc1+ 54. Kb4 Kd4 55.
Kb3 Re1 56. Kc2 Nc8 57. Kb3 Ra1 58. Kc2 Nb6 59. Kb2
Rg1 60. Ka3 Kc3 61. Ka2 Nc8 62. Ka3 Ra1# 0 – 1

# Game 34
**Fianchettoing the rook**

**Date: 2024-11-14**
**White: Stockfish–L20**
**Black: MAV–MCTS($M = 2000$)**
**Result: 1/2 – 1/2**
**Online PGN: https://lichess.org/S9V6YA1i**

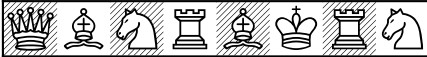

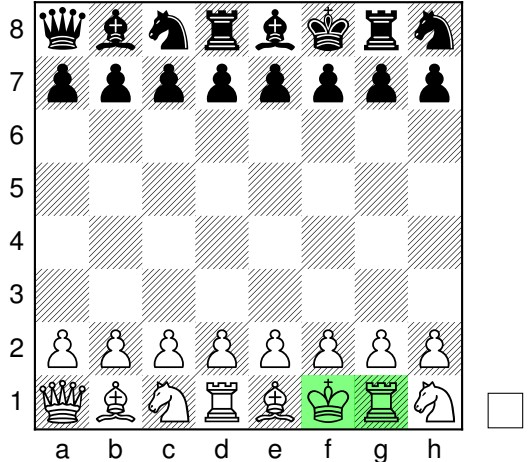

**1. O-O Once in a lifetime opportunities should be taken.**
Ng6 2. c4 c5 3. Ng3 Nb6 4. d4 cxd4 5. Ba5 Ne5 6. c5 Nc6
7. b4 Nxa5 8. bxa5 Nc4 9. a6 d5 10. cxd6 Nxd6 11. Rxd4
Bc7 12. Rh4 h6 13. a4 bxa6 14. Nd3 Qd5 15. Ba2 Qa5 16.
Rc1 Bb6 17. Bb3 g6 18. Rf4

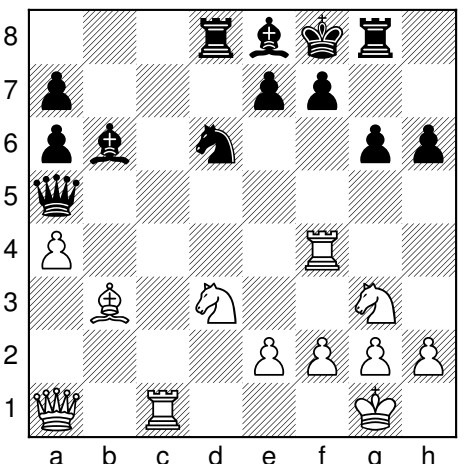

**18...Rg7 MAV–MCTS decides to fianchetto the rook.** 19.
Ne5 Kg8 20. e3 Kh7 21. h4 h5 22. Nc6 Bxc6 23. Rxc6 e6
24. Bc2 Kg8 25. Ne4 Nxe4 26. Rxe4 Qd2 27. Qf6 Kh7 28.
Re5 Kg8 29. Qf4 a5 30. Qf6 Qd7 31. Rc4 Rc8 32. Ree4
Kh7 33. g3 Rc7 34. Rxc7 Qxc7 35. Bd3 Qd7 36. Bb5 Qd5
37. Rc4 Qd8 38. Qxd8 Bxd8 39. Rd4 Bb6 40. Rd7 Kg8
41. Rb7 Kf8 42. Rb8+ Ke7 43. Rb7+ Kf8 44. Rb8+ Ke7
45. Re8+ Kf6 46. Rh8 Ke7 47. Re8+ Kf6 48. Kh2 Rh7 49.
Rg8 Rg7 50. Rh8 Ke7 51. Re8+ Kd6 52. Kh3 Ke5 53. Kg2
Kd6 54. Rh8 Ke5 55. Bc4 Kd6 56. Bd3 Ke7 57. e4 f6 58.
f4 e5 59. Kf3 exf4 60. gxf4 Bd4 61. Bb5 Bb2 62. Rc8 Ba1
63. Ke3 a6 64. Rc7+ Kf8 65. Rc8+ Ke7 66. Rc7+ Kf8 67.
Rc8+ Ke7 (threefold repetition). 1/2 – 1/2

# Game 35
**Bishops vs knights**

**Date: 2024-11-14**
**White: Stockfish–L20**
**Black: MAV–MCTS($M = 2000$)**
**Result: 1/2 – 1/2**
**Online PGN: https://lichess.org/zYVQjKBA**

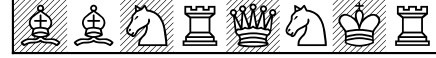

**1. d4 d5 2. b3 h5 3. h4 b6 4. c4 e6 5. Nd3 dxc4 6. bxc4**
c5 7. dxc5 Qc6 8. Ne3 bxc5 9. O-O Bc7 10. Ne5 Qe8 11.
Rxd8 Qxd8 12. g3 f6 13. Nf3 Kf7 14. Bc2 Nd6 15. Qc3
Qc8 16. Ng5+ Ke7 17. Nh3 Ne4 18. Qa3 Nd7 19. Nf4
Bxf4 20. gxf4 Qc7 21. f5 e5 22. Nd5+ Bxd5 23. cxd5 Nd6

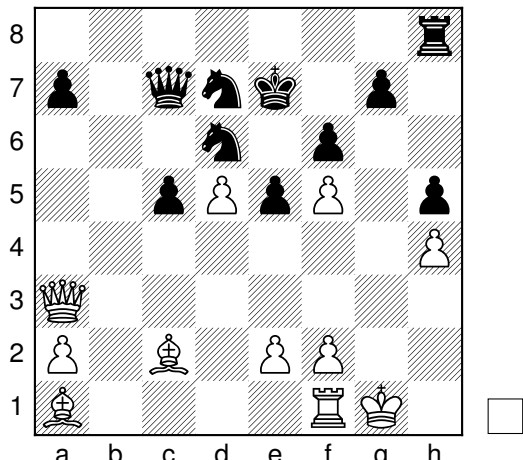

**Bishop pair against two knights imbalance guarantees a complex middlegame.** 24. Kh2 Rg8 25. e4 g5 26. fxg6 Rxg6 27. Qh3 Kd8 28. Bd1 Rg7 29. Qe3 Qa5 30. a3 Rg8 31. Re1

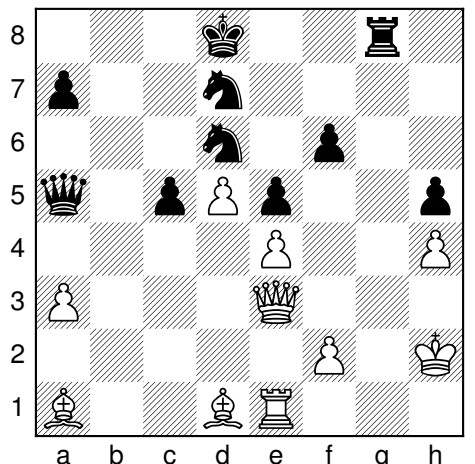

**31...f5 MAV–MCTS breaks the rule of thumb that the side with knights should keep the position closed because open positions favour a bishop pair.** 32. Bxh5 Nxe4 33. Bf3 Nd2 34. Bd1 Ne4 35. Rg1 Rxg1 36. Kxg1 c4 37. Bf3 Qxd5 38. h5 Qd6 39. Kg2 Qf6 40. Bxe4 fxe4 41. Bc3 Qh4 42. h6 Ke7 43. Qh3 Qxh3+ 44. Kxh3 Kf6 45. h7 Kg7 46. Kg4 Kxh7 47. Kf5 Kg8 48. Kxe4 Kf7 49. Kd5 Ke7 50. Bb4+ Kf7 51. Kxc4 Ke6 52. Kb5 Kd5 53. Bd2 Nf6 54. Be3 Ne4 55. Bxa7 Nc3+ 56. Kb4 Ne4 57. Be3

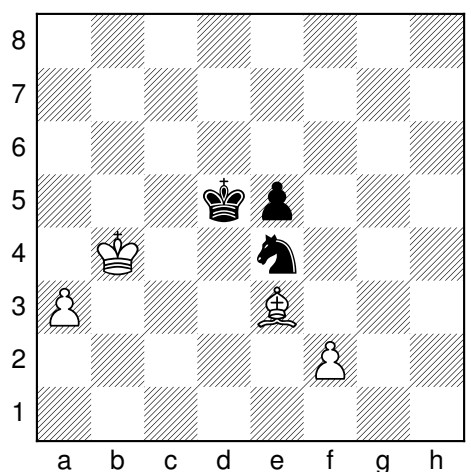

**57...Nxf2 MAV–MCTS correctly decides to sacrifice a knight to steer the game towards a theoretically drawn endgame.** 58. Bxf2 Kd6 59. Ka5 Kc7 60. Ka6 Kb8 61. Be3 e4 62. Bf4+ Ka8 63. a4 e3 64. Bxe3 Kb8 65. Kb6 Ka8 66. Bc5 Kb8 67. a5 Ka8 68. Bf2 Kb8 69. Ka6 Ka8 70. Kb6 Kb8 71. Ka6 Ka8 72. Bd4 Kb8 73. Bg1 Ka8 74. Kb6 Kb8 75. Be3 Ka8 76. a6 Kb8 77. Bc1 Ka8 78. Ba3 Kb8 79. Bc5 Ka8 80. Kc6 Kb8 81. Bg1 Ka8 82. Bc5 Kb8 83. a7+ Ka8 84. Kd5 Kb7 85. Ke5 Ka8 86. Kf6 Kb7 87. Bf2 Ka8 88. Ke5 Kb7 89. Kd6 Ka8 90. Bg1 Kb7 91. Bb6 Ka8 92. Ke6 Kb7 93. Kf5 Ka8 94. Kg4 Kb7 95. Bc5 Ka8 96. Kf3 Kb7 97. Bb6 Ka8 98. Kf4 Kb7 99. Be3 Ka8 100. Ke5 Kb7 101. Kd6 Ka8 102. Ke6 Kb7 103. Kd5 Ka8 104. Ke5 Kb7 105. Bf2 Ka8 106. Kf6 Kb7 107. Bd4 Ka8 108. Bf2 Kb7 109. Be3 Ka8 110. Bf2 (threefold repetition). **This a draw because the queening square is the opposite color of the bishop.** 1/2 – 1/2

## Game 36
### The last resource

**Date: 2024-11-14**
**White: Stockfish–L17**
**Black: MAV–MCTS**($M = 100$)
**Result: 1/2 – 1/2**
**Online PGN: https://lichess.org/QMfa5J52**

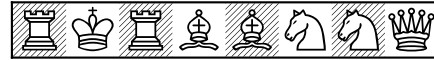

1. a4 a5 **Like in Game 10, pushing the flank pawns on the first move seems to be the preferred choice when rooks are on the a/h files with kings on the adjacent b/g files.** 2. e4 e5 3. Ne3 Ne6 4. g3 g6 5. f4 exf4 6. gxf4 Nxf4 7. Bg3 Ne6 8. Nc4 Ra6 9. Bf3 Ne7 10. e5 Nc6 11. Ne2 f5 12. d4 Ng5 13. d5 Nb4 14. Bg2 h5 15. Nc3 h4 16. Bf4 h3 17. Bf1 Nf7 18. Ra3 g5 19. Bg3 f4 20. Bf2 Nxe5 21. Nxe5

Qxe5 22. Bxa6 bxa6 23. Rb3 c5 24. dxc6 Bf7 25. cxd7
Rc7 26. Na2 Bxb3 27. Nxb4 axb4 28. cxb3 Bf6 29. d8=Q+
Bxd8 30. Re1 Qf5+ 31. Ka1

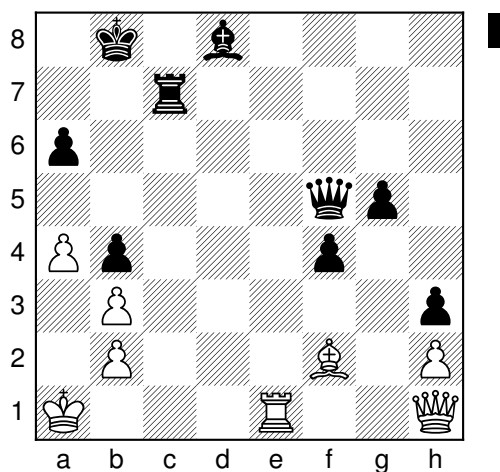

**It is hard to believe that this position in equal despite
White effectively being two pawns down and and all the
pieces stuck at the first two rows.** 31...Rc2 32. Bd4 f3 33.
Re8 Qd7 34. Rxd8+ Qxd8

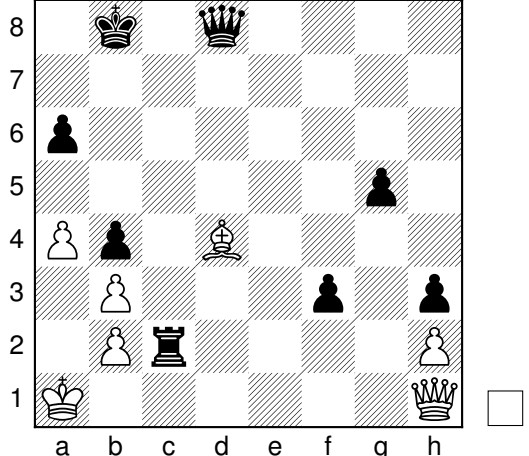

**It is even harder to understand the equality in this
position.** 35. Be5+ Ka7 36. Qxf3 Rc1+ 37. Ka2 Qd1 38.
Qf2+ Kb7 39. Qf7+ Kb6 40. Qf6+ Ka5

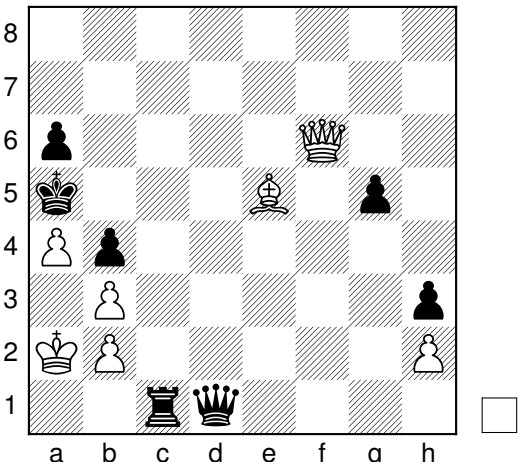

**The drawing mechanism is mesmerizing.** 41. Bc7+ Rxc7
42. Qb6+ Kxb6 43. a5+ Ka7 (stalemate).

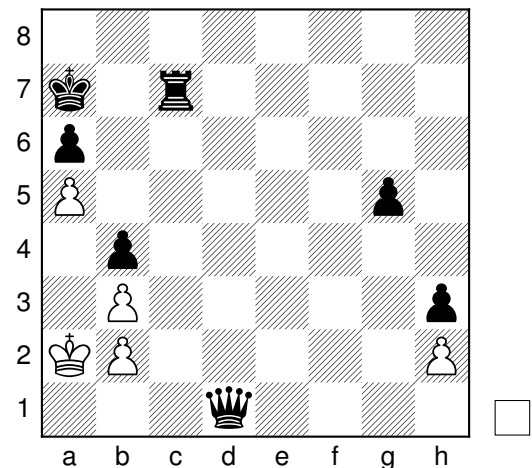

**The final position speaks for itself!** 1/2 – 1/2

## G. Opening Book

In each match-up between two agents, a specific opening is used, and agents swap seats to ensure each agent plays each opening both as black and as white. The openings are listed in Table 5.

Table 5: Table of the opening book used for chess. Openings were collected from various TCEC (Top Chess Engine Championship) seasons. The scores were computed with Stockfish 16 with 1.5B nodes per position. The scores are all from white's perspective (so negative values mean black is better) and were computed via: $\frac{\text{centipawns}}{100}$

| Opening | FEN | Score |
|---|---|---|
| QGD | rnbqkbnr/pp3ppp/4p3/3P4/3p4/2N5/PP2PPPP/R1BQKBNR w KQkq - 0 5 | 0.47 |
| Sicilian | rnbqkb1r/5ppp/p2ppn2/1p6/3NP3/P1N1B3/1PP2PPP/R2QKB1R w KQkq - 0 8 | 0.67 |
| King's Indian | r1bq1rk1/ppp2pbp/n2p2p1/3Pp2n/2P1P3/2N2NPP/PP3P2/R1BQKB1R w KQ - 1 9 | 0.62 |
| KGA | rnbqkb1r/pppp1ppp/8/4P2n/5p2/5N2/PPPP2PP/RNBQKB1R w KQkq - 1 5 | -0.28 |
| Catalan opening | rnbqkb1r/p2p1ppp/5n2/1ppP4/8/6P1/PP2PP1P/RNBQKBNR w KQkq - 0 6 | 0.71 |
| French | r1bqkb1r/pppn1ppp/2n1p3/3p3P/3P4/1N3N2/PPP2PPP/R1BQKB1R b KQkq - 2 6 | 0.63 |
| Nimzo-Indian | r2q1rk1/pb1p1ppp/1pn1p3/2b4n/1PP2B2/P1N2N2/2Q1PPPP/3RKB1R w K - 1 11 | 0.57 |
| Robatsch (modern) defence | rnbqk1nr/1pp1pbbp/p2p2p1/8/3PP3/2N1B3/PPPQ1PPP/R3KBNR b KQkq - 1 5 | 0.52 |
| Budapest | r1bqk2r/pppp1ppp/2n5/4P3/1bP2Bn1/2N2N2/PP2PPPP/R2QKB1R b KQkq - 6 6 | 0.36 |
| Ruy Lopez | r1bq1rk1/2p1bppp/pnnp4/1p2p3/3PP3/1BP2N1P/PP3PP1/RNBQR1K1 w - - 1 11 | 0.63 |
| English | rnbqkb1r/p1pp1ppp/5n2/1p4N1/2P1p3/2N5/PP1PPPPP/R1BQKB1R w KQkq - 0 5 | 0.59 |
| Sicilian defence | r1b1kb1r/pp1p1ppp/1qn1pn2/8/4P3/1NN5/PPP2PPP/R1BQKB1R w KQkq - 0 7 | 0.67 |
| QGD | rnbqkbnr/pp4pp/2p1p3/3p1p2/2PP4/2N1P3/PP3PPP/R1BQKBNR w KQkq - 0 5 | 0.62 |
| Pirc defence | r1bqkb1r/pppnppp1/3p1n1p/8/3PP1P1/2N5/PPP2P1P/R1BQKBNR w KQkq - 0 5 | 0.63 |
| Dutch | rnbqkbnr/ppp1p2p/6p1/3p1pB1/3P4/2N5/PPP1PPPP/R2QKBNR w KQkq - 0 4 | 0.82 |
| Caro-Kann | rn1qkbnr/pp1bppp1/2p4p/3pP3/3P2PP/8/PPP2P2/RNBQKBNR w KQkq - 1 6 | 0.82 |
| King's Indian | r1bq1rk1/pp2ppb1/3p1np1/2pPn3/2P1P2p/2N1BP2/PP2B1PP/R2QK1R1 b KQ - 1 11 | 0.24 |
| French | r1bqkbnr/ppp2ppp/2n1p3/3p4/3PP3/2N5/PPP2PPP/R1BQKBNR w KQkq - 2 4 | 0.77 |
| Trompovsky attack (Ruth, Opovcensky opening) | r1b1kb1r/pp2pppp/2np1n2/q5B1/7Q/2N5/PPP1PPPP/2KR1BNR w kq - 2 7 | 0.55 |
| KGA | r1bqkbnr/pppp1ppp/2n5/8/2B1Pp2/8/PPPP2PP/RNBQK1NR w KQkq - 2 4 | -0.52 |
| QGD | r1bqkbnr/ppp1ppp4/2n5/3P4/2p5/2N5/PP2PPPP/R1BQKBNR b KQkq - 0 4 | 0.73 |
| Sicilian | r1b2rk1/1pq1bppp/p1nppn2/8/3NP3/1BN1B3/PPP1QPPP/2KR2R1 b - - 5 11 | 0.25 |
| English | r2qk2r/pb1nbppp/1p1ppn2/8/2PQP3/2N2NP1/PP3PBP/R1B2RK1 w kq - 1 10 | 0.33 |
| Scandinavian defence | rn2kb1r/pp2pppp/2p2n2/q3Nb2/3P4/2N5/PPP2PPP/R1BQKB1R w KQkq - 0 7 | 0.71 |
| Benoni defence | rn2k2r/ppqb1pbp/3p1np1/2pP4/5B2/1QN2N2/PP2PPPP/R3KB1R w KQkq - 6 10 | 0.70 |
| Giuoco Pianissimo | r1bqk2r/ppp2p2/2np1n2/2b1p1pp/2B1P3/3P1NB1/PPP2PPP/RN1Q1RK1 w kq - 0 9 | 0.55 |
| QGD Slav | rn2kb1r/ppq2ppp/2p1pn2/3p4/2PP2bN/1QN1P3/PP3PPP/R1B1KB1R w KQkq - 4 8 | 0.81 |
| Pirc defence | rnbqkb1r/pp2pp1p/2pp1np1/8/3PP3/2N1B3/PPP2PPP/R2QKBNR w KQkq - 0 5 | 0.74 |
| King's Indian | r1bq1rk1/pppnppb1/3p1np1/7p/2PPP3/2N3N1/PP2BPPP/R1BQK2R w KQ - 0 8 | 0.55 |
| Sicilian | 3q1rk1/p2bppb1/3p1np1/1pr1n1Bp/3NP1PP/1BN2P2/PPPQ4/1K1R3R b - - 0 15 | 0.05 |
| Queen's pawn | rnbqkbnr/p1pp1ppp/1p2p3/8/2PP4/8/PP2PPPP/RNBQKBNR w KQkq - 0 3 | 0.43 |
| French | rnbq1k1r/ppp2pp1/4p2p/3pP3/1b1bPn1Q1/2N1B3/PPP2PPP/R3KBNR w KQ - 4 8 | 0.33 |
| Queen's Indian | rn1qk2r/p1p2ppp/bp1bp3/3n4/2NP4/1P4P1/P3PBBP/RNBQ1K1R w kq - 1 10 | 0.57 |
| KGA | rnbqkbnr/ppp2ppp/3p4/8/2B1Pp2/5N2/PPPP2PP/RNBQK2R b KQkq - 1 4 | -0.42 |
| Bird | rnbqk1nr/ppp2ppp/3b4/8/8/5N2/PPPPP1PP/RNBQKB1R b KQkq - 1 4 | 0.52 |
| Robatsch defence | rnbqk1nr/pp2pbbp/2pp2p1/8/2BPP3/2N5/PPP2PPP/R1BQK1NR w KQkq - 0 5 | 0.69 |
| Benko gambit | rnbqkb1r/p2ppp1p/5np1/1ppP4/2P5/5N2/PP2PPPP/RNBQKB1R w KQkq - 0 5 | 0.55 |
| Ruy Lopez | r1bq1rk1/2p1bppp1/p1np1n1p/1p2p3/4P3/1BP2N1P/PP1P1PP1/RNBQR1K1 w - - 0 10 | 0.53 |
| Benoni | r1bqr1k1/pp3pbp/3p1np1/2pPn3/P3P3/R1N5/1P1NBPPP/2BQ1RK1 b - - 2 12 | 0.62 |
| Sicilian | rnb1k2r/1pq2pbp/p2ppnp1/8/2PNP3/2NB4/PP2QPPP/R1B2RK1 w kq - 2 10 | 0.84 |
| QGD Slav | rnbqk2r/p3bppp/2p1p3/1p1nP3/P1pP4/2N2N2/1P2BPPP/R1BQK2R w KQkq - 2 9 | 0.00 |
| Caro-Kann | rn1qkbnr/pp4pp/2p1ppb1/3pP3/3P2PP/2N5/PPP1NP2/R1BQKB1R b KQkq - 0 7 | 0.63 |
| King's Indian | rnb2rk1/ppp2pbp/3p2p1/3Pp3/2n1P2Q/2N2P2/PP2K2P/R5NR w - - 0 13 | 0.51 |
| Scandinavian | rnb1kbnr/ppp1pppp/3q4/8/8/2N5/PPPP1PPP/R1BQKBNR w KQkq - 2 4 | 0.81 |
| Dutch, 2.Bg5 variation | rnbqkbnr/ppppp2p/6p1/5pB1/3P4/8/PPP1PPPP/RN1QKBNR w KQkq - 0 3 | 0.70 |
| French | r1b1k1r1/ppq1np1Q/2n1p3/3pP3/5P1P/P1p5/2P1N1P1/R1B1KB1R b KQq - 0 12 | 0.14 |
| English | rnbqkb1r/pp2pppp/4n3/8/1pB1P3/2N2N2/P2P1PPP/R1BQ1K1R w kq - 0 10 | 0.58 |
| KGA | rnbqkbnr/ppp2p1p/3p4/8/3PP1pN/2N2pP1/PPP4P/R1BQKB1R b KQkq - 1 7 | -0.55 |
| Old Benoni defence | rnbqkb1r/pp1ppppp/5n2/2pP4/8/5N2/PPP1PPPP/RNBQKB1R b KQkq - 0 3 | 0.39 |
| Sicilian | r1b1k2r/2q1bppp/p2pp3/1pn2PP1/3NP3/2N2Q2/PPP4P/2KR1B1R w kq - 1 14 | 1.21 |
| King's Indian | rnbq1rk1/ppp2pbp/3p1np1/3Pp3/2P1P3/2N1B3/PP2BPPP/R2QK1NR b KQ - 0 7 | 0.57 |
| Sicilian | 1rb1r1k1/2q1bppp/ppnppn2/8/P3PPP1/1NN1BB2/1PPQ3P/R4R1K w - - 1 16 | 0.53 |
| Dutch, 2.Bg5 variation | rnbqkbnr/ppppp1pp/8/5pB1/3P4/8/PPP1PPPP/RN1QKBNR b KQkq - 1 2 | 0.41 |
| KGA | rnbqkbnr/ppp2p1p/3p4/6p1/3PPp2/5N2/PPP3PP/RNBQKB1R w KQkq - 0 5 | -0.26 |
| Benoni defence | rnbqkb1r/pp1p1ppp/4pn2/2pP4/2P5/8/PP2PPPP/RNBQKBNR w KQkq - 0 4 | 0.78 |
| French | r1b1k2r/pp1nbppp/1qn1p3/2ppP3/3P1P2/2P2NP1/PP3K1P/R1BQ1BNR b kq - 2 9 | 0.37 |
| English | r3k2r/1bqnbppp/pp1ppn2/8/2PNP3/2N1B1P1/PP3PBP/R2QR1K1 w kq - 3 12 | 0.37 |
| Robatsch defence | r1bqk1nr/ppp1ppbp/2np2p1/8/3PPP2/2N5/PPP3PP/R1BQKBNR w KQkq - 1 5 | 0.95 |

| | | |
|---|---|---|
| King's Indian | r1bq1rk1/ppp1n1bp/3p1np1/2PPpp2/1P2P3/2N5/P2NBPPP/R1BQ1RK1 w - - 2 12 | 1.14 |
| Sicilian | r2qkb1r/pp1bppp/2np1n2/6B1/3NP3/2N5/PPPQ1PPP/R3KB1R b KQkq - 3 7 | 0.36 |
| QGD | r1bqk2r/pp1n2pp/2pbp2n/3p1p2/2PP4/2N1PN2/PPQ1BPPP/R1B1K2R w KQkq - 4 8 | 0.38 |
| Ruy Lopez | r2qr1k1/1bpnbppp/p1np4/1p6/3PP3/1B3N1P/PP1N1PP1/R1BQR1K1 w - - 1 13 | 0.85 |
| Queen's Indian | rn1q1rk1/1bnbppp/bpp1p3/3p4/2PP4/1PBN2P1/P3PPBP/RN1Q1RK1 b - - 5 11 | 0.53 |
| Scandinavian defence | rnbqkb1r/ppp1pppp/5n2/3P4/8/8/PPPP1PPP/RNBQKBNR w KQkq - 1 3 | 0.77 |
| Kevitz-Trajkovich defence | r1bqkb1r/pppppppp/2n2n2/8/2PP4/8/PP2PPPP/RNBQKBNR w KQkq - 1 3 | 0.50 |
| Evans gambit | r1b1k1nr/pppq1ppp/1bnp4/4p3/2BPP3/1QP2N2/P2N1PPP/R1B2RK1 b kq - 5 9 | -0.07 |
| King's Indian | r1bq1rk1/ppp1ppbp/2np1np1/8/2PPP3/2N1BP2/PP4PP/R2QKBNR w KQ - 3 7 | 0.60 |
| Sicilian | rnr3k1/1pq1bppp/p2pbn2/4p3/4P1P1/1NN1BP2/PPPQ3P/2KR1B1R w - - 1 12 | 0.75 |
| English opening | r1bq1rk1/ppp3bpp/2np1np1/4pp2/2P5/2NP1NP1/PP2PBBP/1RBQ1RK1 w - - 6 9 | 0.47 |
| French | r1bq1rk1/pp2nppp/2n1p3/2ppP3/3P2Q1/P1PB4/2P2PPP/R1B1K1NR w KQ - 5 9 | 1.13 |
| King's Indian | r1bqkb1r/pppn1ppp/3p1n2/4p3/2PP4/2N2N2/PP2PPPP/R1BQKB1R w KQkq - 2 5 | 0.80 |
| two knights defence | r1bqkb1r/ppp2ppp/2n5/3np1N1/2B5/8/PPPP1PPP/RNBQK2R w KQkq - 0 6 | 0.69 |
| Queen's Indian accelerated | r1bqkb1r/p1pppppp/1pn2n2/8/2PP4/5P2/PP2P1PP/RNBQKBNR w KQkq - 1 4 | 0.51 |
| Caro-Kann | r2qkb1r/pp1nnppb/2p1p2p/3pP3/3P4/2P2N2/PP1NBPPP/R1BQ1RK1 w kq - 2 9 | 0.52 |
| King's Indian | rnbq1rk1/ppp3bp/3p1pp1/4p3/2PPP1nB/2N2N2/PP2BPPP/R2QK2R b KQ - 1 9 | 0.51 |
| Sicilian | r1bqkbnr/3p1ppp/p1n1p3/1p6/4P3/1BN2N2/PP3PPP/R1BQ1RK1 b kq - 1 8 | -0.43 |
| Dutch | rnbq1rk1/ppppb1pp/4pn2/5p2/2PP4/5NP1/PP2PPBP/RNBQK2R w KQ - 3 6 | 0.53 |
| Robatsch (modern) defence | rnbqk1nr/pp1pppbp/6p1/2p5/3PP3/5N2/PPP2PPP/RNBQKB1R w KQkq - 0 4 | 0.91 |
| Benoni | r1bqr1k1/1pp3pbp/3p1np1/2pPn3/P3P3/2N5/1P1NBPPP/R1BQ1RK1 w - - 1 12 | 0.87 |
| Danish gambit | rnbqkbnr/pppp1ppp/8/8/2B1P3/8/PB3PPP/RN1QK1NR b KQkq - 0 5 | -0.49 |
| Trompovsky attack (Ruth, Opovcensky opening) | rnb1kb1r/pp1ppppp/1q3n2/2pP4/8/2P2P2/PP2P1PP/RNBQKBNR b KQkq - 2 7 | 0.57 |
| French | r1bqk2r/pp1nbppp/2n1p3/2ppP3/3P4/2PB1N2/PP1N1PPP/R1BQ1RK1 b kq - 2 8 | 0.32 |
| King's Indian | rnbq1rk1/ppp2bpp/3p1np1/3Pp3/2P1P3/2N2N1P/PP3PP1/R1BQKB1R b KQ - 0 7 | 0.33 |
| Sicilian | r1b2rk1/1pqp1ppp/p1n1pn2/8/1b1NP3/2N1BP2/PPPQ2PP/2KR1B1R w - - 1 10 | 0.65 |
| QGD semi-Slav | rnbqkb1r/p4p2/2p1p3/4N2p/NppPn1pP/6B1/PP2BPP1/R2QK2R w KQkq - 0 13 | 0.16 |
| Vienna gambit | r1bqkbnr/pppp1ppp/2n5/4Pp2/2N5/PPPPP2PP/R1BQKBNR w KQkq - 0 4 | -0.53 |
| Benko gambit | rn1qkb1r/3ppp1p/b4np1/2pP4/8/2N5/PP2PPPP/R1BQKBNR w KQkq - 0 7 | 0.80 |
| Pirc | rnbqkb1r/pp2pp1p/2pp1np1/6B1/3PP3/2N5/PPPQ1PPP/R3KBNR b KQkq - 1 5 | 0.49 |
| QGD semi-Slav | rnbqkbnr/pppppppp/8/8/8/8/PPPPPPPP/RNBQKBNR w KQkq - 0 1 | 0.27 |
| Ruy Lopez | r2qkbnr/1ppb1ppp/p1np4/4p3/B3P3/2P2N2/PP1P1PPP/RNBQK2R w KQkq - 1 6 | 0.58 |
| King's Indian | rnbq1rk1/ppp2pbp/3p1np1/4p3/2PPP3/2N1BP2/PP4PP/R2QKBNR w KQ - 0 7 | 0.77 |
| Sicilian | 1r1q1rk1/p4pbp/2p1b1p1/3np3/2B5/2N1BP2/PPP3PP/1K1RQ2R w - - 4 15 | 0.75 |
| Dutch | r1bq1rk1/ppp1p1bp/2np1np1/5p2/2PP4/2N2NP1/PP2PPBP/R1BQ1RK1 w - - 2 8 | 0.91 |
| Modern defence | rnbqk1nr/pppppbp/3p2p1/8/2PPP3/8/PP3PPP/RNBQKBNR w KQkq - 0 4 | 0.60 |
| Benoni defence | rnbq1rk1/1p3pbp/p2p1np1/2pP4/P4B2/2N1PN1P/1P3PP1/R2QKB1R b KQ - 0 10 | 0.62 |
| KGA | rnb1kbnr/pppp1ppp/8/8/2B1Pp1q/8/PPPP2PP/RNBQ1KNR b kq - 3 4 | -0.62 |
| Queen's pawn game, Chigorin variation | r1bqkbnr/ppp1pppp/2n5/3p4/3P4/5N2/PPP1PPPP/RNBQKB1R w KQkq - 2 3 | 0.54 |
| French | rnbqk2r/pppnbppp/4p3/3pP1B1/3P3P/2N5/PPP2PP1/R2QKBNR b KQkq - 0 6 | 0.17 |
| King's Indian | r1bq1rk1/pppnn1bp/3p2p1/3Ppp2/2P1P3/2NN4/PP2BPPP/R1BQ1RK1 w - - 0 11 | 0.50 |
| Sicilian | r1bqkbnr/1p3pp1/p1npp2p/8/3NP1PP/2N5/PPP2P2/R1BQKB1R w KQkq - 0 8 | 0.93 |
| King's Indian | r1bq1rk1/pppnnpbp/3p2p1/3Pp3/2P1P3/2N5/PP2BPPP/R1BQNRK1 w - - 3 10 | 0.57 |
| Sicilian | r1b1k2r/1pq1bppp/p1nppn2/8/3NPP2/P1N1BQ2/1PP3PP/2KR1B1R b kq - 3 10 | 0.72 |
| Dutch | rnbqkb1r/ppppp2p/5np1/5p2/3P1B2/2N1PN3/PPP2PPP/R2QKBNR b KQkq - 1 4 | 0.68 |
| KGA | rnbqkbnr/ppp2p1p/3p4/8/3PPppP/5N2/PPP3P1/RNBQKB1R w KQkq - 0 6 | -0.26 |
| Benoni | rnbqk2r/1p3pbp/p2p1np1/2pP4/4P3/2N3N1/PP3PPP/R1BQKB1R w KQkq - 0 9 | 0.70 |
| French | rnbqk2r/pp2nppp/4p3/2ppP3/3P2Q1/P1P5/2P2PPP/R1B1KBNR b KQkq - 2 7 | 0.46 |
| Robatsch defence | r1bqk1nr/2pnppbp/p2p2p1/1p6/3PPP2/2NB1N2/PPP3PP/R1BQK2R w KQkq - 2 7 | 0.87 |
| King's Indian | r1b2rk1/pp1n1pbp/1qpp1np1/8/2PNP3/2N3PP/PP3PB1/R1BQR1K1 b - - 0 11 | 0.51 |
| Sicilian | r1bqkb1r/pp3ppp/3p2n1/1N1Pp3/2P5/8/PP3PPP/R1BQKB1R w KQkq - 1 10 | 0.50 |
| QGD semi-Slav | r1b2rk1/pp1nqppp/2p1p3/3p4/2PPn1P1/4PN2/PPQ2P1P/2KRBBR1 b - - 4 12 | 0.36 |
| Ruy Lopez | r1bqk2r/1pp1bppp/p1np1n1n2/4p3/B3P3/2P2N2/PP1P1PPP/RNBQR1K1 b kq - 0 7 | 0.30 |
| Queen's Indian | r2q1rk1/pb1pbppp/np2pn2/2p5/2PP4/P1N2NP1/1PQBPFBP/R3K2R w KQ - 0 10 | 0.94 |
| Scandinavian | rnb1kb1r/ppp1pppp/3q1n2/8/3P4/2N2N2/PPP2PPP/R1BQKB1R b KQkq - 2 5 | 0.58 |
| QGD | r2q1rk1/pb1nbppp/1p3n2/2pp2B1/3P4/2NBPN2/PPPQ2PP/2KR3R w - - 0 11 | 0.50 |
| Vienna game | N2k1b1r/pb1p4/1pnn1qp1/4pp1p/8/1B1Q4/PPPPNPPP/R1B1K2R w KQ - 0 13 | 0.72 |
| King's Indian | rnbq1rk1/ppp2pbp/3p2p1/3Pp2n/2P1P3/2NB4/PP2NPPP/R1BQK2R w KQ - 3 8 | 0.67 |
| Sicilian | rnbqk1nr/1p1pbppp/p3p3/8/4P3/1N1B4/PPP2PPP/RNBQK2R w KQkq - 4 7 | 0.80 |
| English opening | r1bq1rk1/1ppp1ppp/2n2n2/p3p3/1bP1P3/2NP2P1/PP3PBP/R1BQK1NR w KQ - 1 7 | 0.23 |
| French | r1bq1rk1/1p1n1ppp/p1n1p3/2bpP3/3N1P2/2N1B3/PPPQ2PP/2KR1B1R w - - 0 11 | 0.76 |
| Neo-Gruenfeld defence | r1bqkb1r/ppp1p1bp/1nn3p1/4P3/3P1p2/2N1BP2/PP1Q2PP/2KR1BNR w - - 0 11 | 0.42 |
| Two knights defence | r1bqkb1r/p4ppp/2p5/n3p1N1/5n1P/3B4/PPPP1PP1/RNBQK2R w KQkq - 1 10 | 0.17 |
| English opening | rnbqkbnr/p1pp1ppp/1p2p3/8/2P5/2N5/PP1PPPPP/R1BQKBNR w KQkq - 0 3 | 0.49 |
| Caro-Kann | rnbqkb1r/pp2pppp/2p2n2/3p4/4P3/2N2Q2/PPPP1PPP/R1B1KBNR w KQkq - 2 4 | 0.63 |
| King's Indian | r1bq1rk1/pp1npppbp/3p1np1/2pP4/2P1P3/2N1BP1N/PP4PP/R2QKB1R b KQkq - 0 8 | 0.42 |
| Sicilian | rnb1kb1r/2qp1ppp/pp2pn2/8/2PNP3/P1N5/1P3PPP/R1BQKB1R w KQkq - 0 8 | 0.64 |
| Dutch defence | rnbq1rk1/4pp4/2pbpn2/3p1p2/2PP4/6PN/PPQ1PPBP/RNB2RK1 w - - 4 8 | 0.57 |
| Robatsch (modern) defence | rnbqk1nr/2p1ppbp/p2p2p1/1p6/3PP2/2N1B3/PPPQ1PP1/R3KBNR b KQkq - 0 6 | 0.39 |
| Old Benoni defence | rnb1kb1r/p2pppp/1q3n2/1ppP2B1/8/5N2/PPP1PPPP/RN1QKB1R w KQkq - 2 5 | 0.67 |
| Ruy Lopez | r2q1rk1/1ppbbppp/p1np1n2/4p3/B2PP3/2P2N2/PP3PPP/RNBQR1K1 w - - 5 9 | 0.57 |
| Trompovsky attack (Ruth, Opovcensky opening) | rnb1kb1r/pp1ppppp/1q3n2/2pP4/4PB2/2P2P2/PP4PP/RN1QKBNR b KQkq - 0 7 | 0.00 |
| French | r1b1kb1r/pp1n1ppp/2n1p3/q1ppP3/N2P1P2/2P1BN2/PP4PP/R2QKB1R b KQkq - 0 9 | 0.74 |

| | | |
|---|---|---|
| King's Indian | r1bq1rk1/pppnppbp/3p1np1/8/2PPP3/2N2N2/PP2BPPP/R1BQK2R w KQ - 4 7 | 0.76 |
| Sicilian | r1b1k2r/1p1nbppp/pq1ppn2/6B1/4PP2/1NN2Q2/PPP3PP/2KR1B1R b kq - 6 10 | -0.05 |
| QGD | r1bqkb1r/ppp2ppp/2n1pn2/3p4/2PP4/2N2N2/PP2PPPP/R1BQKB1R w KQkq - 2 5 | 0.54 |
| Giuoco Pianissimo | r1bqk2r/bppn1p2/2np3p/p3p1p1/2B1P3/2PP1NB1/PP1N1PPP/R2Q1RK1 w kq - 2 11 | 0.45 |
| Benko gambit | rn1qkb1r/pb1ppppp/5n2/1ppP4/2P5/5N2/PP2PPPP/RNBQKB1R w KQkq - 2 5 | 0.56 |
| Pirc defence | rnbqkb1r/pp2pbpp/2p1np1p/8/3PP3/2N1B2P/PPP2PP1/R2QKBNR w KQkq - 1 6 | 0.73 |
| Old Indian | rn1qkbnr/ppp1pppp/3p4/8/3P2b1/5N2/PPP1PPPP/RNBQKB1R w KQkq - 2 3 | 0.89 |
| Ruy Lopez | r1bq1rk1/2nbbppp/p1np4/1p2p3/4P3/1BP2N1P/PP1P1PP1/RNBQR1K1 w - - 1 10 | 0.64 |
| King's Indian | r1b1qrk1/1pp2pbp/n2p1np1/p2Pp1B1/2P1P3/2N4P/PP1NBPP1/R2QK2R b KQ - 5 10 | 0.21 |
| Sicilian | rn1qk2r/1b1nbppp/2p3/1p4P1/3NP2P/P1N5/1PP2P2/R1BQKB1R w KQkq - 1 11 | 0.43 |
| Dutch | r1bq1rk1/ppp1p1bp/5np1/3Ppp2/2P5/2N3P1/PP2PPBP/R1BQ1RK1 w - - 0 10 | 0.87 |
| Robatsch defence | rnbqk1nr/pp2ppb1/2p3p1/3pP2p/3P1P2/2N5/PPP3PP/R1BQKBNR w KQkq - 0 6 | 0.69 |
| Queen's pawn game | rnbqkb1r/pp1p1ppp/4pn2/2pP4/2P5/5N2/PP2PPPP/RNBQKB1R b KQkq - 0 4 | 0.46 |
| KGA | rnbqkb1r/pppp1p1p/8/4P1pn/5p2/5N2/PPPPB1PP/RNBQ1RK1 b kq - 1 6 | -0.78 |
| QGD | r1bqr1k1/p2nbppp/2p2n2/1p1p2B1/3P4/2NBP3/PPQ1NPPP/2KR3R w - - 0 11 | -0.32 |
| French | rnb1k2r/pppn1ppp/4p3/3pP1q1/3P4/2N5/PPP2PP1/R2QKBNR w KQkq - 0 8 | 0.32 |
| King's Indian | rnbq1rk1/ppp2pbp/3p1np1/3Pp3/2P1P3/2N1B2P/PP3PP1/R2QKBNR b KQ - 0 7 | 0.61 |
| Sicilian | 2rq1rk1/pp1bppbp/3p1np1/4n3/3NP3/1BN1BP2/PPPQ2PP/1K1R3R b - - 6 12 | 0.89 |
| QGD semi-Slav | rnbqkbnr/pppppppp/8/8/3P4/8/PPP1PPPP/RNBQKBNR b KQkq - 0 1 | 0.11 |
| Sicilian | rnbqkb1r/1p3ppp/p2pn2/8/3NP1P1/2N5/PPP2P1P/R1BQKB1R w KQkq - 0 7 | 1.39 |
| King's Indian | r1bq1rk1/ppp1n1bp/3p1np1/3Pp3/2P1Pp2/2NN1P2/PP1BB1PP/R2Q1RK1 w - - 0 13 | 0.85 |
| Vienna game | r1bqkb1r/pppp3p/2nn2p1/1N2pp2/8/1B3Q2/PPPP1PPP/R1B1K1NR w KQkq - 0 8 | 0.58 |
| Dutch defence | rnbq1rk1/ppp3bp/3ppnp1/5p2/2PP4/2N2NP1/PP2PPBP/R1BQ1RK1 w - - 0 8 | 0.78 |
| French | rnbqk1nr/p1p2ppp/1p2p3/3pP3/1b1P4/2N5/PPP2PPP/R1BQKBNR w KQkq - 0 5 | 1.07 |
| Benoni | rnbqkb1r/pp3p1p/3p1np1/2pP4/8/2N5/PP1NPPPP/R1BQKB1R b KQkq - 1 7 | 0.59 |
| Robatsch (modern) defence | rnbqkbnr/1p2p1p1/p2p2p1/8/3PP3/2N5/PPP2PPP/R1BQKBNR w KQkq - 0 4 | 0.65 |
| Budapest defence | rnbqkb1r/pppp1ppp/8/4P3/2P3n1/8/PP2PPPP/RNBQKBNR w KQkq - 1 4 | 0.78 |
| Sicilian defence | r1b1kb1r/pp1p1ppp/1qn1pn2/8/4P3/1NN5/PPP1QPPP/R1BK1B1R b KQkq - 1 7 | 0.43 |
| King's Indian | 1r1q1rk1/1ppbppbp/p1np1np1/8/2PPP3/2N1BP2/PP1QN1PP/2R1KB1R w K - 4 10 | 0.66 |
| Ruy Lopez | r1bqkbnr/ppp3pp/2n5/1B1Pp3/4N3/5N2/PPPP1PPP/R1BQK2R w KQkq - 0 6 | 0.63 |
| QGD Slav | rn1qkb1r/pp1ppp5/7b/P1NP4/2N2P2/1P2P1PP/R1BQKB1R b KQkq - 0 8 | 1.00 |
| Owen defence | rn1qkbnr/pbpp1ppp/1p2p3/8/3PP3/3B1N2/PPP2PPP/RNBQK2R b KQkq - 1 4 | 0.86 |
| Queen's Indian | rnbqkb1r/p2p1ppp/1p2pn2/2p5/2PP4/P4N2/1P2PPPP/RNBQKB1R w KQkq - 0 5 | 0.91 |
| Sicilian | rnbqkbnr/pppppppp/8/8/4P3/8/PPPP1PPP/RNBQKBNR b KQkq - 0 1 | 0.15 |
| King's Indian | r1b1nrk1/pp1n1pbp/1qp3p1/2p1P3/4P3/2N2NPP/PP3PB1/R1BQ1RK1 w - - 1 12 | 0.87 |
| Sicilian | r4rk1/pp1bppbp/2np1np1/q7/2BNP3/2N1BP2/PPPQ2PP/2KR3R w - - 7 11 | 0.84 |
| English opening | r1bqkbnr/pppppppp/2n5/8/2P5/8/PP1PPPPP/RNBQKBNR w KQkq - 1 2 | 0.56 |
| French | r2qkb1r/1b1n1npp/p1n1p3/1pppP3/3P1P2/4BN2/PPPQ2PP/R2NKB1R w KQkq - 2 10 | 0.54 |
| Semi-Benoni ('blockade variation') | rnbqkbnr/pp3ppp/3p4/2pPp3/4P3/8/PPP2PPP/RNBQKBNR w KQkq - 0 4 | 1.43 |
| Scandinavian | rnb1kb1r/1pp1pppp/p2q1n2/8/3P4/2N2NP1/PPP2P1P/R1BQKB1R b KQkq - 0 6 | 0.97 |
| Benko gambit | r1bqkb1r/p2ppppp/n4n2/1ppP5/8/PPQ1PPPP/RNB1KBNR w KQkq - 2 5 | 0.57 |
| Caro-Kann | rn1qkbnr/pp3ppp/4p1b1/2ppP3/3P2P1/2N5/PPP1NP1P/R1BQKB1R w KQkq - 0 7 | 0.00 |
| King's Indian | rnb2rk1/pp2ppbp/3p1np1/8/2P1P3/1NP1p2P/P4PP1/R2QKB1R w KQ - 0 11 | -0.30 |
| Sicilian | rnb1kb1r/1pqp1ppp/p3pn2/8/3NP3/3B4/PPP2PPP/RNBQ1RK1 w kq - 4 7 | 0.58 |
| Dutch defence, Blackburne variation | rnbqkb1r/ppp3pp/4pn2/3p1p2/3P4/6PN/PPP1PPBP/RNBQK2R w KQkq - 0 5 | 0.53 |
| Pirc defence | r1bqkb1r/pppn1pp1/3p1n1p/4p3/3PP1P1/2N4P/PPP2P2/R1BQKBNR w KQkq - 0 6 | 0.74 |
| Benoni defence | rnbq1rk1/pp1p1ppp/3b1n2/2pP4/4PP2/2N5/PP4PP/R1BQKBNR b KQ - 0 7 | 0.61 |
| two knights defence | r1bq1b1r/ppp3pp/4k3/3np3/1nB5/2N2Q2/PPPP1PPP/R1B1K2R w KQ - 4 9 | 0.64 |
| Trompovsky attack (Ruth, Opovcensky opening) | rnb1kb1r/pp1ppppp/1q3n2/2pP2B1/8/2N5/PPP1PPPP/R2QKBNR b KQkq - 2 4 | 0.32 |
| Reti | r1bq1rk1/p2nbppp/2n1p3/1pppP3/8/3P1NP1/PPP2PBP/R1BQRNK1 b - - 2 10 | -0.23 |
| King's Indian | rnbqk2r/ppppppbp/5np1/8/2PP4/2N5/PP2PPPP/R1BQKBNR w KQkq - 2 4 | 0.62 |
| Sicilian | rqb1k1nr/1p1p1ppp/1Bn1p3/1N6/1b2P3/8/PPP2PPP/R2QKB1R w KQkq - 1 10 | -0.41 |
| QGD Slav | rnbqkb1r/p3pppp/2p2n2/1p6/2pPP3/2N2N2/PP3PPP/R1BQKB1R w KQkq - 0 6 | 0.00 |
| KP | r1bqkbnr/pppppppp/2n5/8/4P3/8/PPPP1PPP/RNBQKBNR w KQkq - 1 2 | 0.48 |
| Benoni | rnbqk2r/pp1n1pbp/3p2p1/2pP4/4PP2/2N5/PP2B1PP/R1BQK1NR b KQkq - 4 9 | 0.70 |
| Pirc defence | rnbqk2r/ppp1ppbp/3p1np1/8/3PP3/2N1B3/PPPQ1PPP/R3KBNR b KQkq - 3 5 | 0.46 |
| QGD | r3kbnr/ppp2ppp/2n5/3qp3/3P4/4PP2/PP3P1P/RNBQKB1R w KQkq - 0 7 | 0.84 |
| Ruy Lopez | r2qrbk1/1bp2pp1/p2p1n1p/8/pn1PP3/5N1P/1P1N1PP1/RBBQR1K1 w - - 0 16 | 0.39 |
| Sicilian defence | rnb2rk1/1p2bppp/pq1ppn2/8/4PB2/1NN5/PPPQ1PPP/2KR1B1R w - - 6 10 | 1.00 |
| Dutch | rnbqkb1r/pp2p1pp/2pp1n2/5p2/2PP4/2N2N2/PP2PPPP/R1BQKB1R w KQkq - 2 5 | 0.69 |
| King's Indian | rnbqkbnr/pp1ppp1p/6p1/2p5/8/3PP3/PPP2PPP/RNBQKBNR w KQkq - 0 3 | 0.84 |
| Nimzo-Indian | rn1q1rk1/pbpbp1pp/1p2pn2/4P3/1bPP4/2NB4/PP3PPP/R1BQK1NR b KQ - 0 7 | 0.49 |
| two knights defence | r1bqk2r/pppp1ppp/2n2n2/2b1p1N1/2B1P3/8/PPPP1PPP/RNBQK2R w KQkq - 6 5 | 1.21 |
| Queen's pawn game, Chigorin variation | 2kr1bnr/pppqpppp/2n5/3p4/3P2b1/2P2NP1/PP2PBPP/RNBQ1RK1 b - - 0 6 | 0.85 |
| French | rnbqkbnr/p1p2ppp/1p2p3/3pP3/3P4/8/PPP2PPP/RNBQKBNR w KQkq - 0 4 | 1.00 |
| King's Indian | r1bq1rk1/pppnn1bp/3p4/3Pp1p1/2Pp1p1/2N2P2/PP2BBPP/R2QNRK1 w - - 0 13 | 0.52 |
| Sicilian | r1bq1rk1/1p2bppp/p1nppn2/8/3NP3/2N1BP2/PPPQ2PP/2KR1B1R w - - 5 10 | 0.78 |
| Sicilian | rnbqkbnr/pp1ppppp/8/2p5/4P3/8/PPPP1PPP/RNBQKBNR w KQkq - 0 2 | 0.38 |
| King's Indian | r1bq1rk1/ppp1npbp/3p1np1/3Pp3/2P1P3/2N1BN2/PP2BPPP/R2QK2R w KQ - 1 9 | 0.89 |
| French | rnb1k1nr/ppq3pp/4p3/2ppPp2/3P2Q1/P1P5/2P2PPP/R1B1KBNR w KQkq f6 0 8 | 1.45 |
| QGD | rnbqkbnr/p5pp/2p1pp2/1p6/2pP4/2N2N2/PP1BPPPP/R2QKB1R w KQkq - 0 7 | 0.69 |

| | | |
|---|---|---|
| KGA | r1bqkbnr/ppp2p1p/2np4/1B6/3PPB2/2N2Q2/PPP3PP/R3K2R b KQkq - 2 8 | -0.73 |
| Dutch defence | r1b1qrk1/ppp1p1bp/n2p1np1/5p2/2PP4/1P3NP1/PB2PBPP/RN1Q1RK1 w - - 1 9 | 0.71 |
| Robatsch (modern) defence | r1bqk2r/ppp1ppbp/2np1np1/4P3/2BP4/2N5/PPP1QPPP/R1B1K1NR b KQkq - 0 6 | -0.07 |
| Budapest | rnbqkb1r/pppp1p1p/8/4P1p1/2P2Bn1/8/PP2PPPP/RN1QKBNR w KQkq - 0 5 | 1.17 |
| Sicilian | r1b1kb1r/2q2p1p/p2ppP2/1pn5/3NP3/2N2Q2/PPP4P/2KR1BR1 b kq - 2 16 | 0.99 |
| King's Indian | r1b1qrk1/ppp2bbp/n2p1np1/3Pp1B1/2P1P1P1/2N2N1P/PP3P2/R2QKB1R b KQ - 0 9 | 0.99 |
| Ruy Lopez | r1bqk2r/pppppppp/2n5/1Bb1p3/4P3/2P2N2/PP1P1PPP/RNBQK2R w KQkq - 1 5 | 0.71 |
| Queen's Indian | rn2kb1r/pbnpqppp/1p6/2p5/4P3/2B1N2/1PQ2PPP/RNB2RK1 w kq - 2 11 | 0.96 |
| Caro-Kann | rnb1kb1r/pp3ppp/1q2p3/2ppP3/3P4/3B1NP1/PPP3PP/R1BQK2R b KQkq - 0 9 | 0.68 |
| Blackmar gambit | rnbqkb1r/pp2pppp/2p2n2/8/3P4/2N2N2/PPP3PP/R1BQKB1R w KQkq - 0 6 | -0.53 |
| Pirc | rnb1kb1r/pp3ppp/2pp1n2/q3p3/3PPP2/2NB1N2/PPP3PP/R1BQK2R b KQkq - 1 6 | 0.93 |
| Benoni defence | rnbq1rk1/pp1p1ppp/3b1n2/2pP4/8/2N3P1/PP2PP1P/R1BQKBNR w KQ - 1 7 | 0.74 |
| Sicilian | r1bq1rk1/pp1nppbp/2np2p1/8/2BNP3/2N1BP2/PPPPQ2PP/R3K2R w KQ - 5 10 | 1.17 |
| King's Indian | r1bq1rk1/ppp2pb1/3p1npp/2nPp3/2P1P3/2N1B3/PP1QBPPP/R3K1NR w KQ - 3 10 | 0.83 |
| French | r1bqk2r/2pnbppp/1pn1p3/p2pP3/P2P4/1NP2N2/1P3PPP/R1BQKB1R w KQkq - 0 9 | 1.16 |
| Trompovsky attack (Ruth, Opovcensky opening) | rnb1kb1r/pp1pp1p/1q3np1/2pP4/8/2P2P2/PP2P1PP/RNBQKBNR w KQkq - 0 8 | 0.87 |
| Scotch | r1b1kbnr/pppp1ppp/2n5/8/3NP2q/8/PPP2PPP/RNBQKB1R w KQkq - 1 5 | 0.97 |
| Nimzo-Indian | rn1q1rk1/pb1p2pp/4pn2/1Pp5/1b6/2NBP3/P3PP1P/R1BQK1NR w KQ - 1 9 | 0.53 |
| Robatsch defence | r1bqk2r/ppp1ppbp/2np1np1/8/3PPP2/2N1BN2/PPP3PP/R2QKB1R b KQkq - 4 6 | 0.79 |
| QGD semi-Slav | r1bq1rk1/pp1nbbp1/2p1pn1p/3p4/2PP1BP1/2N1PN2/PPQ2P1P/R3KB1R b KQ - 0 9 | 0.36 |
| Dunst (Sleipner, Heinrichsen) opening | rn1qkbnr/ppp3pp/4bp2/4p3/3pP3/5NN1/PPPP1PPP/R1BQKB1R w KQkq - 0 6 | -0.60 |
| Sicilian | rnbqk2r/pp1p1ppp/4pn2/8/1b1NP3/2N5/PPP2PPP/R1BQKB1R w KQkq - 3 6 | 0.92 |
| King's Indian | r1bqnrk1/ppp1n1bp/3p2p1/3Pp3/2P1Pp2/2N2N2/PP2BBPP/R2QNRK1 b - - 1 12 | 0.74 |
| Philidor | rnbqk2r/ppp2pbp/3p1np1/8/3NP3/2N1B3/PPP2PPP/R2QKB1R w KQkq - 2 7 | 1.08 |
| Dutch, 2.Bg5 variation | rnbqk1nr/ppp1p1bp/6p1/3p1pB1/3P4/2N2P2/PPPQP1PP/R3KBNR b KQkq - 0 5 | 0.67 |
| KP | r2qkbnr/ppp1pppp/2np4/8/3PP1b1/5N2/PPP2PPP/RNBQKB1R w KQkq - 1 4 | 1.41 |
| Czech Benoni defence | r1bqkb1r/pp1n1p1p/3p1np1/2pPp3/2P1P3/2N2N2/PP3PPP/R1BQKB1R w KQkq - 0 7 | 1.26 |
| French | r1bqkb1r/ppp1nppp/2n1p3/3pP3/3P4/2N5/PPP2PPP/R1BQKBNR w KQkq - 1 5 | 1.05 |
| Bogo-Indian defence, Gruenfeld variation | rnbq1rk1/pp1nbppp/4p3/2ppP2/2PP3P/P2B1N2/1P1N1PP1/R1BQK2R b KQ - 0 9 | 0.34 |
| Sicilian | r1b1kb1r/p1qp1pp1/1pn1p2p2/p1P2P/5B2/2P2N2/PPPQ1PP1/R3KB1R w KQkq - 0 10 | 1.19 |
| King's Indian | r1bq1rk1/pp3pbp/3ppnp1/2pPn3/2P1P3/2N1BPN1/PP4PP/R2QKB1R w KQ - 0 10 | 0.46 |
| Latvian counter-gambit | rnbqkbnr/pppp2pp/8/4pp2/4P3/2N2N2/PPPP1PPP/R1BQKB1R b KQkq - 1 3 | 1.28 |
| QGD Slav | rnbqkbnr/pp4pp/2p1p3/3p1p2/2PP4/2N2N2/PP2PPPP/R1BQKB1R w KQkq - 0 5 | 1.17 |
| Scandinavian defence | rn1qkb1r/ppp1pppp/1n6/8/2PP2b1/5N2/PP3PPP/RNBQKB1R w KQkq - 1 6 | 1.33 |
| English | r1b1kb1r/pp1p1p1p/4p3/1N2n1p1/Q1P2Bn1/2N3P1/PP1KPq1P/R4B1R w kq - 0 11 | -0.13 |
| Robatsch defence | rnbqk1nr/1pp2pbp/p2pp1p1/8/3PPP2/2N2N2/PPP3PP/R1BQKB1R w KQkq - 0 6 | 0.95 |
| King's Indian | rnbq1rk1/ppp3bp/3p2p1/3Ppp1n/2P1P3/2N1BP2/PP1Q2PP/R3KBNR w KQ - 0 9 | 0.77 |
| Sicilian | rnb1kb1r/2q2ppp/p3pn2/1p2P1B1/3N4/2N5/PPP3PP/R2QKB1R w KQkq - 1 10 | 0.68 |
| QGD | r3kbnr/pppq1ppp/2n1p3/8/3P4/2N1PP2/PP3P1P/R1BQKB1R w KQkq - 2 8 | 0.93 |
| KGA | r1b1k1nr/ppppqp1p/2n5/3P4/1b2P1pN/2N2pP1/PPP4P/R1BQKB1R w KQkq - 1 9 | 1.08 |
| Benoni | r1bqr1k1/pp3pbp/n2p1np1/2pP4/4P3/2N5/PP1NBPPP/R1BQ1RK1 w - - 7 11 | 0.89 |
| French | rnb1k1nr/pppq2pp/4p3/3pPp2/1b1P2Q1/2N5/PPP2PPP/R1B1KBNR w KQkq f6 0 6 | 1.33 |
| Dutch | rnb1qrk1/ppp1b1pp/3ppn2/5p2/2PP4/2N2NP1/PP2PPBP/R1BQ1RK1 w - - 2 8 | 0.94 |
| Petrov | rnbq3r/ppp1bkpp/3p1n2/8/3PP3/8/PPP2PPP/RNBQKB1R w KQ - 1 6 | -0.47 |
| Sicilian | r1bqkb1r/1p3pp1/p1nppn2/7p/3NP1PP/2N1B3/PPP2P2/R2QKBR1 w Qkq - 0 10 | 0.86 |
| Sicilian | rnbqkb1r/1p2pppp/p2p1n2/8/3NP3/2N5/PPP2PPP/R1BQKB1R w KQkq - 0 6 | 0.32 |
| King's Indian | r1bq1rk1/ppp1npbp/3p1np1/3Pp3/2P1P3/2N2N2/PP2BPPP/R1BQ1RK1 w - - 1 9 | 0.73 |
| French | rnbqkbnr/ppp2ppp/4p3/3pP3/3P4/2N5/PPP2PPP/R1BQKBNR w KQkq - 1 5 | 1.16 |
| QGD Slav | rn1qkb1r/pb3ppp/2p1p3/1p1nP3/P1pPN3/5N2/1P2BPPP/R1BQK2R b KQkq - 3 9 | -0.29 |
| KGA | rnbqkbnr/pppp1p1p/8/6p1/4Pp2/2N2N2/PPPP2PP/R1BQKB1R b KQkq - 1 4 | -0.68 |
| Dutch, 2.Bg5 variation | rnbqk1nr/ppppp1b1/6p1/5pB1/3P4/2N2N2/PPP1PPPP/R2QKB1R w KQkq - 0 5 | 1.15 |
| Robatsch defence | rnbqk2r/ppp1npbp/3pp1p1/8/3PPP2/2N2N2/PPP3PP/R1BQKB1R w KQkq - 2 6 | 0.87 |
| Gruenfeld defence | r2q1rk1/ppp1ppbp/1nn3p1/8/3PP1b1/1QN1BN2/PP3PPP/3RKB1R w K - 7 11 | 1.02 |
| Ruy Lopez | rnbqkbnr/pppp1ppp/8/4p3/4P3/8/PPPP1PPP/RNBQKBNR w KQkq - 0 2 | 0.25 |
| Queen's Indian | rn1qk2r/p1p2ppp/bp1bpn2/4N3/2pP4/1P4P1/P3PBBP/RNBQ1K1R w kq - 4 9 | 0.97 |
| Scandinavian defence | rnbqkbnr/ppp1pppp/8/3p4/4P3/8/PPPP1PPP/RNBQKBNR w KQkq - 0 2 | 0.66 |
| Blumenfeld counter-gambit accepted | rnbqkb1r/p2p2ppp/4pn2/1Pp5/8/5N2/PP2PPPP/RNBQKB1R b KQkq - 0 6 | 0.58 |
| Pirc defence | rnbq1rk1/ppp1ppbp/3p1np1/8/3PP3/2N1BP2/PPP3PP/R2QKBNR w KQ - 3 6 | 0.62 |
| Benoni | rnbqkb1r/pp3p1p/3p1np1/2pP4/8/2N2N2/PP2PPPP/R1BQKB1R w KQkq - 0 7 | 0.72 |
| Sicilian | 1rb2rk1/2qp1ppp/p2bpn2/1p2n3/3BPP2/1NN3Q1/PPP3PP/2KR1B1R b - - 0 14 | 0.88 |
| King's Indian | r1bq1rk1/pp1n1pbp/2pp1np1/4p3/2PPP3/2N2N2/PP2BPPP/R1B2RK1 w - - 0 9 | 0.85 |
| French | rnbqk1nr/ppp2ppp/4p3/3pP3/1b1P4/2N5/PPP2PPP/R1BQKBNR b KQkq - 0 4 | 0.49 |
| Trompovsky attack (Ruth, Opovcensky opening) | rnb1kb1r/pp1ppppp/1q6/2pP4/4nB2/8/PPPNPPPP/R2QKBNR b KQkq - 4 5 | -0.29 |
| Ruy Lopez | r2qkbnr/2p2pp1/p2p4/1p2p2p/3nP1b1/1B3N1P/PPP2PP1/RNBQ1RK1 w kq - 0 9 | 0.89 |
| Nimzo-Indian | rnbq1bk1/pp1p1ppp/3Prn2/8/8/P1N3P1/1P2NP1P/R1BQKB1R w KQ - 1 12 | 0.49 |
| Robatsch (modern) defence | rnbqkbnr/ppp1ppp1p/6p1/8/4P3/8/PPPP1PPP/RNBQKBNR w KQkq - 0 2 | 0.69 |
| Queen's Indian accelerated | rn1qkb1r/pbppppp1/1p2pn2/8/2PP4/2N2P2/PP2P1PP/R1BQKBNR w KQkq - 0 5 | 0.91 |
| Bird | rnbqkbnr/pppp1ppp/8/4p3/5P2/8/PPPPP1PP/RNBQKBNR w KQkq - 0 2 | 0.66 |
| Sicilian | r1bqkb1r/pp3ppp/2nppn2/8/3NP1P1/2N5/PPP2P1P/R1BQKB1R w KQkq - 1 7 | 1.20 |
| King's Indian | r1bq1rk1/pppn2bp/3p1pp1/4p1B1/2PPP1n1/2N2N2/PP2BPPP/R2Q1RK1 w - - 0 10 | 0.92 |

| | | |
|---|---|---|
| KGD | rnb1kbnr/ppppp1ppp/5q2/4p3/4PP2/8/PPPP2PP/RNBQKBNR w KQkq - 1 3 | 1.08 |
| Dutch defence | r1bqkb1r/ppppp2p/2n2np1/5p2/3P4/6PN/PPP1PPBP/RNBQK2R w KQkq - 2 5 | 0.77 |
| Alekhine's defence | rn1qk2r/ppp1bppp/3pp3/3nP3/3P2b1/5N2/PPP1BPPP/RNBQ1RK1 w kq - 2 7 | 0.91 |
| Czech Benoni defence | rnbqkb1r/pp1p1ppp/5n2/2pPp3/2P5/8/PP2PPPP/RNBQKBNR w KQkq e6 0 4 | 1.27 |
| French | rnbqkb1r/ppp2pp1/4pn1p/3p4/3PP3/2NB4/PPP2PPP/R1BQK1NR w KQkq - 2 5 | 1.27 |
| Benko gambit | rn1qkb1r/4pp1p/3p1np1/2pP4/4P3/2N5/PP3PPP/R1BQ1KNR w kq - 0 9 | 0.92 |
| Sicilian | r1b1kb1r/pp3ppp/1q1ppn2/4n3/4PB2/1NN5/PPP2PPP/R2QKB1R w KQkq - 2 9 | 1.28 |
| King's Indian | r1bq1rk1/ppp1ppbp/2np1np1/8/2PPP3/2N2N1P/PP3PP1/R1BQKB1R w KQ - 3 7 | 0.83 |
| Four knights | r1bqkb1r/pppp1ppp/2n2n2/8/3PP3/2N5/PPP2PPP/R1BQKB1R w KQkq - 1 6 | -0.54 |
| QGD Slav | r1bqkb1r/pp2pppp/n1p2n2/8/P1pP4/2N2N2/1P2PPPP/R1BQKB1R w KQkq - 1 6 | 0.84 |
| Caro-Kann | 2kr1b1r/ppqnnpp1/2p1p2p/3pPb2/3P4/2P2N2/PP1NBPPP/R1BQR1K1 w - - 4 10 | 0.78 |
| English | rnbqkb1r/p1n2ppp/4p3/1p6/1pBPP2P/5N2/P3NPP1/R1BQ1K1R w kq - 0 13 | 1.12 |
| Robatsch defence | rnbqk1nr/pp2ppbp/2pp2p1/8/3PPP2/2N5/PPP3PP/R1BQKBNR w KQkq - 0 5 | 0.95 |
| King's Indian | r1bq1rk1/1p1n1pbp/p2p1np1/3Pp3/4P1P1/2N1BP2/PP1QN2P/R3KB1R b KQ - 0 11 | 0.72 |
| Sicilian | r1b2rk1/1pq1bppp/p1nppn2/8/3NPP2/P1N1BQ2/1PP3PP/2KR1B1R w - - 4 11 | 0.81 |
| QGD | r1bqkbnr/ppp1pppp/2n5/8/2pP4/2N5/PP2PPPP/R1BQKBNR w KQkq - 0 4 | 0.83 |
| Ruy Lopez | r2qr1k1/1bpn1ppp/p1np1b2/1p6/3PP3/1B3N1P/PP1N1PP1/1RBQR1K1 w - - 3 14 | 0.92 |
| Benoni defence | rnbqk2r/pp1p1ppp/3b1n2/2pP4/8/2N2N2/PP2PPPP/R1BQKB1R b KQkq - 2 6 | 1.27 |
| French | rnb1k1nr/pppq1ppp/4p3/3pP3/1b1P4/2N5/PPP2PPP/R1BQKBNR w KQkq - 1 5 | 1.01 |
| Dutch | rnbqkb1r/pppp3p/4pnp1/5p2/3P4/5NP1/PPP1PPBP/RNBQK2R w KQkq - 0 5 | 0.84 |
| Vienna | N1bk1b1r/p2pq2p/1pnn2p1/3Qpp2/8/1B6/PPPP1PPP/R1B1K1NR w KQ - 0 11 | 0.89 |
| King's Indian | r1bq2k1/pppnn1bp/3p1r2/3Pp1p1/2P1Pp2/2N2P2/PP2BBPP/2RQNRK1 w - - 2 14 | 0.82 |
| Sicilian | r1bqkb1r/pp2pp2/2np1np1/7p/3NP3/2N1BP2/PPP3PP/R2QKB1R w KQkq - 0 8 | 0.66 |
| Latvian | rnbqkbnr/pppp2pp/8/4pp2/2B1P3/5N2/PPPP1PPP/RNBQK2R b KQkq - 1 3 | 0.00 |
| Nimzo-Indian | rn1q1rk1/pbp2p1p/1p2pPp1/6B1/1bpP2Q1/2N5/PPB2P1P/2KR2N1 w - - 2 13 | 1.21 |
| Robatsch (modern) defence | rnbqkbnr/pp2pp2p1/8/3PPP2/2N5/PPP3PP/R1BQKBNR b KQkq - 0 4 | 0.35 |
| Robatsch (modern) defence | r1bqk2r/2pnppbbp/p2p1np1/1p6/3PPP/2N1BP2/PPPQ4/R3KBNR b KQkq - 0 8 | 0.56 |
| Ruy Lopez | r2qk2r/ppp2ppp/2p5/2b1p3/4P1b1/3P1N2/PPP2P2/RNBQ1RK1 w kq - 0 10 | 0.00 |
| Dutch, 2.Bg5 variation | rnbqkbnr/ppppp1pp/8/5p2/3P4/6P1/PPP1PPPP/RNBQKBNR w KQkq - 0 2 | 0.59 |
| Budapest | rnbqkb1r/pppp1ppp/8/4P3/2P1P1n1/8/PP3PPP/RNBQKBNR b KQkq - 0 4 | 0.64 |
| Sicilian | rnbqkbnr/pp1ppppp/8/8/1p2P3/P7/2PP1PPP/RNBQKBNR w KQkq - 0 3 | -0.48 |
| King's Indian | r1bq1rk1/ppp1ppbp/3p1n2/3Pp1p1/2P1Pp2/2NN1P2/PP2B1PP/R2QBRK1 w - - 0 14 | 0.59 |
| QGD semi-Slav | r2qk2r/pb1n1ppp/2pbpn2/1p6/3P2P1/2N1PN2/PPQ1BP1P/R1B1K2R w KQkq - 2 10 | 0.48 |
| French | rnbqk1nr/ppp1bppp/4p3/3pP3/3P4/2N5/PPP2PPP/R1BQKBNR b KQkq - 0 4 | 0.44 |
| King's Indian | rnbqk2r/ppp1ppbp/3p1np1/8/2PPP3/2N5/PP3PPP/R1BQKBNR w KQkq - 0 5 | 0.60 |
| Trompovsky attack (Ruth, Opovcensky opening) | rnb1kb1r/pp2pppp/1q1p1n2/2pP4/8/2P2P2/PP2P1PP/RNBQKBNR w KQkq - 0 8 | 0.69 |
| Queen's pawn | rn1qk1nr/pbpp1ppp/1p2p3/8/1bBPPP3/3B4/PP3PPP/RNBQK1NR w KQkq - 3 5 | 0.92 |
| Bird | rnbqk1nr/ppp2p1p/3b4/6p1/8/5NP1/PPPPP2P/RNBQKB1R b KQkq - 0 5 | 0.42 |
| Sicilian | r1b1kbnr/pp1ppppp/1qn5/1Bp5/4P3/5N2/PPPP1PPP/RNBQK2R w KQkq - 4 4 | 0.75 |
| Robatsch (modern) defence | r1bqk1nr/pppnppbp/3p2p1/8/3PP3/2N1B3/PPP2PPP/R2QKBNR w KQkq - 2 5 | 0.67 |
| Dutch defence | rnbqkbnr/ppppp2p/6p1/5p2/3P4/6P1/PPP1PP1P/RNBQKBNR w KQkq - 0 3 | 0.81 |
| Ruy Lopez | r1b1k2r/2q1bppp/p2p1n2/npp1p3/3PP3/2P2N2/PPB2PPP/RNBQR1K1 w kq - 1 11 | 0.90 |
| Caro-Kann | rn1qkbnr/pp1b1pp1/4p2p/2ppP2P/3P1PP1/2P5/PP6/RNBQKBNR b KQkq - 0 8 | 0.71 |
| King's Indian | rnbq1rk1/1pp2pbp/3p1np1/p2Pp3/2P1P3/2N1B3/PP2BPPP/R2QK1NR w KQ - 0 8 | 0.82 |
| QGD semi-Slav | r3kb1Q/pb1n1p1p/2p1p3/q7/3P4/8/2NBPPP/q1BQK2R b Kq - 2 14 | 0.33 |
| Queen's Indian | rn1qkb1r/p2p1ppp/bp2pn2/2pP4/2P5/P4N2/1PQ1PPPP/RNB1KB1R b KQkq - 2 6 | 0.75 |
| QGD | r3k1nr/p1p2ppp/1pnq4/4p3/3P4/2P1PP2/P2B1P1P/1R1QKB1R w Kkq - 0 11 | 0.76 |
| Benoni | rnbqk2r/pp3pbp/3p1np1/2pP4/4PP2/2N5/PP4PP/R1BQKBNR w KQkq - 1 8 | 1.04 |
| Queen's Indian accelerated | r1bqkb1r/p1ppnpppp/1p3n2/3Pp3/2P3P1/2N2P2/PP22P2P/R1BQKBNR b KQkq - 0 6 | 0.79 |
| Benoni | r1bqkb1r/pp1n1p1p/3p1np1/2pP4/8/2N5/PP1NPPPP/R1BQKB1R w KQkq - 2 8 | 0.84 |
| Owen defence | rnbqkbnr/p1ppppppp/1p6/8/4P3/8/PPPP1PPP/RNBQKBNR w KQkq - 0 2 | 0.97 |
| Robatsch (modern) defence | rnbqkbnr/pp2p1p1p/2pp2p1/8/2PPP3/8/PP3PPP/RNBQKBNR w KQkq - 0 4 | 0.85 |
| Philidor | r2q1rk1/1pp1bppp/p2pbn2/8/3QPB2/2N2P2/PPP3PP/2KR1B1R w - - 0 11 | 0.81 |
| French | rnb1kbnr/p1pq1ppp/1p2p3/3pP3/3P3P/8/PPP2PP1/RNBQKBNR w KQkq - 1 5 | 1.02 |
| QGD Slav | rn1qkb1r/pp22pppp/2p2n2/4N2b/P1pP4/2N5/1P2PPPP/R1BQKB1R w KQkq - 3 7 | 0.75 |
| Old Indian | r1bq1rk1/1p1nbppp/2pp1n2/4p3/2PPP3/2N2N2/PP2BPPP/R1BQ1RK1 w - - 0 8 | 0.85 |
| Sicilian | r1bq1rk1/1p1nbppp/p1npp3/8/3NP1P1/2N1BP2/PPPQ3P/2KR1B1R w - - 1 11 | 0.77 |
| French | rnbqk1nr/p1p2ppp/1p2p3/3pP3/1b1P3P/2N5/PPP2PP1/R1BQKBNR b KQkq - 0 5 | 1.10 |
| Dutch defence | rnb1qrk1/ppp1p1b1/3p1npp/3P1p2/2P5/2N2NP1/PP2PBPP/R1BQ1RK1 w - - 0 9 | 0.83 |
| Sicilian | r1r3k1/pp1bppbp/2np1np1/q7/3NP3/1BN1BP2/PPPQ2PP/2KR3R w - - 9 12 | 0.76 |
| Scandinavian | rnb1kb1r/1pp1pppp/p2q1n2/8/3P4/2N2N2/PPP2PPP/R1BQKB1R b KQkq - 0 6 | 1.11 |
| King's Indian | r1b2rk1/pp1n1pbp/1qpp1np1/4p3/2PPP3/2N2NPP/PP3PB1/R1BQ1RK1 w - - 1 10 | 1.05 |
| Sicilian | rnbqk2r/pp2bppp/3ppn2/8/3NP1P1/2N5/PPP2P1P/R1BQKB1R w KQkq - 1 7 | 1.19 |
| Sicilian | 2rq1rk1/pp1bppbp/3p1np1/4n3/3NP3/1BN1BP2/PPPQ2PP/1K1R3R b - - 10 12 | 0.72 |
| Ruy Lopez | r2qrbk1/1b3p2/p2p1npp/1ppP4/Pn2P3/5N1P/1P1N1PP1/RBBQR1K1 w - - 0 17 | 1.08 |
| Sicilian | rn1q1rk1/1p2bppp/3pbn2/p3p3/4P3/1NN1BP2/PPPQ2PP/2KR1B1R w - - 0 11 | 1.13 |
| Old Benoni defence | rnbqkbnr/pp1p1ppp/8/2pPp3/8/8/PPP1PPPP/RNBQKBNR w KQkq e6 0 3 | 1.39 |
| Sicilian | r2qkb1r/3b1pp1/p2ppn1p/1p2n3/3NP1P1/2N1BP2/PPPQ3P/2KR1B1R w kq - 1 12 | 1.04 |
| Benoni defence | r1bq1rk1/pp2pbbp/n2p1np1/2pP4/2P1P3/2N2N1P/PP3PP1/R1BQKB1R w KQ - 1 8 | 1.09 |
| QGD semi-Slav | r3kb1r/1bqn1ppp/p4n2/1p1pP3/2pNP3/2N5/PPB2PPP/R1BQ1RK1 w kq - 0 14 | 0.69 |
| Czech Benoni defence | rnbqkb1r/pp3ppp/3p1n2/2pPp3/2P1P3/2N5/PP3PPP/R1BQKBNR b KQkq - 0 5 | 1.16 |
| Sicilian | r2q1rk1/1p1bppbp/p1np1np1/8/3NP1P1/2N1B2P/PPPQ3P/2KR1B1R w - - 1 11 | 0.85 |
| French | rnbqk2r/pp1nbppp/4p3/2ppP1B1/3P3P/2N5/PPP2PP1/R2QKBNR w KQkq - 0 7 | 0.88 |
| English | rnbqkb1r/ppp1pp2/6pp/3n4/7P/2N2N2/PP1PPPP1/R1BQKB1R w KQkq - 0 6 | 0.86 |
| Sicilian | rnbqkb1r/pp3ppp/3ppn2/8/3NP1P1/2N5/PPP2P1P/R1BQKB1R b KQkq - 0 6 | 0.48 |

| | | |
|---|---|---|
| St. George defence | rnbqkbnr/2pp1ppp/p3p3/1P6/3PP3/8/PP3PPP/RNBQKBNR b KQkq - 0 4 | 0.74 |
| Dutch, 2.Bg5 variation | rnbqkb1r/ppppp2p/6pn/5pB1/3P4/2N5/PPP1PPPP/R2QKBNR w KQkq - 2 4 | 0.81 |
| Robatsch (modern) defence | r1bqk1nr/2pnppb1/p2p2p1/1p5p/3PP2/2N1BP1N/PPPQ2P1/R3KB1R b KQkq - 1 8 | 0.60 |
| English opening | r1bqkbnr/ppp1np1p/3p4/3Pp3/2P3p1/2N2NP1/PP2PP1P/R1BQKB1R w KQkq - 1 7 | 1.09 |
| French | rnbqkb1r/p2n1ppp/4p3/1pppP3/3P4/2PB4/PP1N1PPP/R1BQK1NR w KQkq - 0 7 | 0.95 |
| Sicilian | rn1q1rk1/1p2bppp/p2pbn2/4p3/4P1P1/1NN1B3/PPP1BP1P/R2QK2R w KQ - 1 10 | 0.88 |
| French | rnbqkb1r/ppp1nppp/4p3/3p4/3PP3/8/PPPN1PPP/R1BQKBNR w KQkq - 2 4 | 0.78 |
| Scandinavian (centre counter) defence | rnb1kbnr/ppp1pp1p/6p1/3q4/8/5N2/PPPP1PPP/RNBQKB1R w KQkq - 0 4 | 0.84 |
| Robatsch defence | rnbqk1nr/2p1pbbp/pp1p2p1/8/P2PP3/2N2N2/1PP2PPP/R1BQKB1R w KQkq - 0 6 | 0.82 |
| French | r1bqk1nr/pp3ppp/2n1p3/2ppP3/3P4/P1P5/2P2PPP/R1BQKBNR w KQkq - 1 7 | 0.71 |
| QGA | r1bqkb1r/ppp1nppp/5n2/3Pp3/2p1P3/2N1B3/PP3PPP/R2QKBNR w KQkq - 1 7 | 0.75 |
| Old Indian | rn1qkbnr/ppp1pppp/3p4/8/2PP4/5P2/PP3PPP/RNBQKB1R b KQkq - 0 4 | 0.91 |
| Sicilian | r1b1kb1r/1pq2pp1/p1nppn1p/8/3NPPP1/2N1B2P/PPP5/R2QKB1R w KQkq - 1 10 | 0.85 |
| QGD Slav | rn1qkb1r/1p2pppp/p1p2n2/2Pp4/3P2b1/2N2N2/PP2PPPP/R1BQKB1R w KQkq - 1 6 | 0.89 |
| English | 1qr1k2r/pb1nbppp/1p1ppn2/8/2PNPP2/2N1B1P1/PP4BP/R2QR1K1 w k - 1 13 | 0.79 |
| King's Indian defence | r1bqkb1r/pp1n1p1p/3p1np1/2pPp3/2P1P3/5P2/PP2N1PP/RNBQKB1R w KQkq - 1 7 | 1.10 |
| Queen's pawn, Mason variation, Steinitz counter-gambit | r1b1kbnr/pp3ppp/1qn5/3pp3/2pP1B2/2P1P3/PPQ2PPP/RN2KBNR w KQkq - 0 7 | 0.05 |
| Sicilian | r1bqkb1r/1p1npp2/p2p1n1p/6p1/3NP3/2N3B1/PPP1QPPP/R3KB1R b KQkq - 1 9 | 0.86 |
| Robatsch (modern) defence | rnbqk1nr/ppp2pbp/3pp1p1/8/3PP3/2N1B3/PPP2PPP/R2QKBNR w KQkq - 0 5 | 1.20 |
| Caro-Kann | rn2kbnr/pp2ppp1/1qp5/3pPb1p/3P1B1P/3B4/PPP2PPP/RN1KNR b KQkq - 3 6 | 0.00 |
| French | r1bqkbnr/p1p2ppp/1pn1p3/3pP3/3P4/2N5/PPP2PPP/R1BQKBNR w KQkq - 0 5 | 1.11 |
| KP | r1bqkb1r/pppnppppp/2n5/3pP3/3P4/2N5/PPP2PPP/R1BQKBNR w KQkq - 1 5 | 1.21 |
| Sicilian | rnb1k2r/1pq1bppp/p2ppn2/8/3NP3/2N1B3/PPPQBPPP/R3K2R w KQkq - 4 9 | 1.03 |
| King's Indian | r2q1rk1/1ppbppbp/p1np1np1/6B1/2PP3/2N2P2/PP1QN1PP/R3KB1R w KQ - 2 9 | 0.99 |
| Dutch | rnb1qrk1/ppp1b1pp/3ppn2/5p2/1PPP4/2N2NP1/P3PBPP/R1BQ1RK1 b - - 0 8 | 0.86 |
| Robatsch (modern) defence | rnbqk2r/pp2npbp/3pp1p1/2pP4/2P1P3/2N2N2/PP3PPP/R1BQKB1R w KQkq - 2 7 | 1.07 |
| King's Indian | r2q1rk1/ppp1ppbp/3p1np1/n2P4/2P3b1/2N2NP1/PP2PBPP/R1BQ1RK1 w - - 1 9 | 1.01 |
| Neo-Gruenfeld defence | r1bq1rk1/ppp2pbp/1nn3p1/4p3/3PP3/2N1BP2/PP1Q2PP/2KR1BNR w - - 0 10 | 0.81 |
| Vienna | 1k1r3r/pPp2ppp/5n2/1Nb5/3P1pbq/5N2/PPP1K1PP/R1BQ1B1R w - - 3 11 | 0.94 |
| French | rnbqk2r/pp3p2/4p1pp/3pP3/3pn1Q1/P1PB4/2P2PPP/R1B1K1NR w KQkq - 0 11 | 0.76 |
| Ruy Lopez | r1bn1rk1/2q1bppp/p2p1n2/1ppPp3/4P3/2P2N1P/PPBN1PP1/R1BQR1K1 w - - 1 14 | 1.11 |
| Trompovsky attack (Ruth, Opovcensky opening) | r1bqkb1r/pp3p1p/2n2p2/2ppp3/3P4/2P1PN2/PP2BPPP/RN1QK2R b KQkq - 1 7 | -0.67 |
| KGA | rnbqkb1r/pppp1p1p/5n2/4N3/3PPppP/8/PPP3P1/RNBQKB1R b KQkq - 0 6 | -0.89 |
| English | r1bqk1nr/pp1p1pbp/4p1p1/1N6/1pP5/6P1/3PPPBP/R1BQK1NR b KQkq - 1 8 | -1.27 |
| Sicilian | r1bqk1r1/1p2pbbp/p1np1np1/8/2BNP3/2N1BP2/PPPQ2P/R3K2R w KQ - 0 10 | 1.06 |
| English opening | rn1qk2r/pbppppbp/1p4p1/3P4/2P1n3/5NP1/PP2PPBP/RNBQK2R w KQkq - 1 7 | 0.77 |
| King's Indian | r1bq2k1/pppnn1bp/3p1r2/2PPp1p1/1P2Pp2/2NN1P2/P3B1PP/R1BQ1RK1 w - - 1 14 | 1.10 |
| French | rnb1k1nr/ppp2ppp/4p3/3q4/1b1P4/2N5/PPP2PPP/R1BQKBNR w KQkq - 0 5 | 0.94 |
| King's Indian | r1bq1rk1/ppp2pb1/2np1npp/4p3/2PP3/2N1BN2/PP2BPPP/R2Q1RK1 w - - 2 9 | 0.96 |
| QGD | rnbqkb1r/ppp1pppp/8/3n4/3P4/8/PPP2PPPP/RNBQKBNR w KQkq - 0 4 | 1.07 |
| French | r1bqk2r/ppp1nppp/2n1p3/b2pP3/3P4/P1N5/1PP1NPPP/R1BQKB1R w KQkq - 1 7 | 1.06 |
| Ruy Lopez | r1bq1rk1/pppn1ppp/2np1b2/1B2p3/3PP3/2P2N2/PP3PPP/RNBQR1K1 w - - 1 9 | 0.92 |
| Queen's pawn game | r1bq1rk1/pp3pbp/n2ppnp1/2pP4/2P1P3/2NB1N1P/PP3PP1/R1BQK2R w KQ - 3 9 | 1.21 |
| Petrov | rnbq1b1r/ppp2k1p/3p1np1/8/3PP3/8/PPP2PPP/RNBQKB1R w KQ - 0 6 | -0.73 |
| English opening | rnbqkb1r/ppp1pp1p/5np1/8/7P/2N2N2/PPPP1PP1/R1BQKB1R w KQkq - 1 6 | 1.13 |
| Sicilian | rnb1kb1r/1pq2pp1/p2ppn1p/8/3NP1P1/2N1BQ2/PPP2P1P/R3KB1R w KQkq - 2 9 | 1.16 |
| French | r1bq1rk1/p2nbppp/1pn1p3/2ppP3/3P1P2/P1N1BN2/1PPQ2PP/2KR1B1R b - - 1 10 | -1.07 |
| Robatsch (modern) defence | r1bqk2r/2p1ppbb1/p1np1np1/1p5p/3PP2P/2N1BP2/PPPQ2P1/2KR1BNR w kq - 3 9 | 0.95 |
| Queen's pawn | rn1qk1nr/pbppp1ppp/1p2p3/8/2PP4/2N5/PP1QPPPP/R3KBNR w KQkq - 2 6 | 1.09 |
| Owen defence | rn1qkb1r/pb1p1ppp/1p2pn2/2p5/3PP3/2PB1N2/PP2QPPP/RNB1K2R b KQkq - 2 6 | 0.85 |
| King's Indian defence | rnbqkb1r/pppppp1p/6p1/7n/2PP4/5P2/PP2P1PP/RNBQKBNR w KQkq - 1 4 | 0.84 |
| Pirc | rnbqkb1r/pp2pppp/2pp1n2/8/3PP3/2N5/PPP2PPP/R1BQKBNR w KQkq - 0 4 | 0.96 |
| Scandinavian (centre counter) defence | rn1qkb1r/pp3ppp/2p1pn2/8/3P4/2N1BQ1P/PPP2PP1/R3KB1R w KQkq - 0 9 | 0.87 |
| Dutch, Korchnoi attack | rnbqkb1r/ppppp1pp/8/8/3P2n1/8/PPP1PP2/RNBQKBNR w KQkq - 0 5 | 1.13 |
| QGD | r1b1k2r/pp1n1ppp/2pb1n2/q2p2B1/3P4/2NBPN2/PP3PPP/R2QK2R w KQkq - 2 9 | 0.95 |
| King's Indian | r1bq1rk1/ppp1n1bp/3p1np1/3Pp3/1P1P1Pp2/2N1BP2/1P2B1PP/R2QNRK1 w - - 0 13 | 0.97 |
| Sicilian, Szen, hedgehog variation | rq3rk1/3bbppp/ppnppn2/8/2P1P3/N1N1B3/PP2BPPP/2RQ1RK1 w - - 4 13 | 0.68 |
| King's Indian | r1bq1rk1/2npppbp/p2p1np1/1p6/2PP3/2N1BP2/PP1Q2PP/2KR1BNR w - - 0 9 | 1.12 |
| Benoni | rnbq1rk1/p4pbp/1p1p1np1/2pP4/4P3/2N3N1/PP2BPPP/R1BQK2R w KQ - 0 10 | 0.81 |
| Queen's Indian | r2q1rk1/pb1nbppp/1p1ppn2/8/2PNP3/P1N1B3/1PQ1BPPP/2KR3R w - - 6 12 | 1.00 |
| French | rnbqkb1r/pp3ppp/2n1p3/2ppP3/3P4/2P2N2/PP3PPP/RNBQKB1R w KQkq - 3 6 | 0.81 |
| French | r1b1kb1r/p2n1ppp/1pn1p3/q1ppP3/N2P1P2/2P1BN2/PP4PP/R2QKB1R w KQkq - 0 10 | 0.82 |
| Caro-Kann | r2qk1nr/pp1n1ppp/2p1p3/b2pPb2/3P4/5N2/PPPNBPPP/R1BQ1RK1 w kq - 6 8 | 0.81 |
| King's Indian | rnb2rk1/ppp2pbp/3p2p1/3Pp2n/2P1P2q/2N1BP2/PP1Q2PP/R3KBNR w KQ - 3 9 | 1.15 |
| Ruy Lopez | r1bqkb1r/ppppn1pp/5n2/1B2pP2/8/3P1N2/PPP2PPP/RNBQK2R w KQkq - 1 6 | 0.81 |
| Dutch | r1bq1rk1/ppp1p1b1/5npp/3Ppp2/2P5/1QN3P1/PP2PBPP/R1B2RK1 w - - 0 11 | 0.92 |
| King's Indian | rnbq1rk1/pp1p1pbp/4pnp1/2pP4/2P1P3/2N2P2/PP4PP/R1BQKBNR w KQ - 0 7 | 0.73 |
| Polish defence | rnbqkbnr/p1pppppp/8/1p6/3P4/8/PPP1PPPP/RNBQKBNR w KQkq - 0 2 | 1.19 |
| Ruy Lopez | r1bq1rk1/2p1bppp/p1n2n2/P2pp3/1p2P3/1B3N2/1PPP1PPP/RNBQR1K1 w - - 0 10 | 0.62 |
| King's Indian | r1bq1rk1/pp1n1pbp/2pp1np1/4p3/2PP3/2N1BN2/PP2BPPP/R2Q1RK1 w - - 0 9 | 1.09 |

| | | |
|---|---|---|
| Modern defence | rnbqk1nr/ppp3bp/3p4/3Ppp2/2P5/2N5/PP3PPP/R1BQKBNR w KQkq - 0 7 | 0.76 |
| Nimzo-Indian | rnbq1rk1/ppp2ppp/3ppn2/8/2PPP3/2P5/P1Q2PPP/R1B1KBNR w KQ - 0 7 | 0.92 |
| Old Benoni defence | rnbqkb1r/pp2pp1p/3p1np1/2pP4/4PP2/2N5/PPP3PP/R1BQKBNR b KQkq - 0 5 | 0.71 |
| Old Indian | rnq1kb1r/pp2pppp/2pp1n2/8/2PP2b1/1QN2N2/PP2PPPP/R1B1KB1R w KQkq - 0 6 | 1.07 |
| Ruy Lopez | r2qr1k1/1b1n1ppp/p2p1b2/1ppP4/2n1P3/5N1P/PPB2PP1/1RBQRNK1 w - - 1 17 | 0.85 |
| Sicilian | rnbqk2r/pp2bpp1/3ppn1p/8/3NP1P1/2N1B3/PPP2P1P/R2QKB1R w KQkq - 0 8 | 1.15 |
| Nimzo-Indian | rn1qk2r/p1pp1pp1/bp2pn2/7p/1bPP4/2N1P1N1/PP3PPP/R1BQKB1R w KQkq - 0 7 | 0.89 |
| French | r3k1nr/pbpq2pp/1pn1p3/3pPp2/P2P4/2P3Q1/2P2PPP/R1B1KBNR w KQkq - 1 10 | 1.00 |
| King's Indian | r1bqr1k1/1pp1ppbp/p1np1np1/6B1/2PPP3/2N2P2/PP1QN1PP/R3KB1R w KQ - 2 9 | 0.81 |
| Sicilian | r1bqk2r/1p1nppbp/p2p1np1/8/3NP3/2N1BP2/PPPQ2PP/R3KB1R w KQkq - 2 9 | 1.14 |
| Nimzo-Indian | r1bqk2r/pp1n1ppp/4p3/b1Pp4/3P4/P1N5/1P1QNPPP/R3KB1R w KQkq - 1 11 | 0.71 |
| King's Indian | r1b1qrk1/ppp2pbp/3p1np1/2nPp1B1/2P1P1P1/2N2N1P/PP3P2/R2QKB1R w KQ - 1 10 | 1.13 |
| French | rnbq1k1r/pp2nppp/4p3/2ppP3/3P2Q1/P1P5/2P2PPP/R1B1KBNR w KQ - 3 8 | 0.79 |
| Budapest | rnbqkb1r/ppp2ppp/3p4/4P3/2P1P1n1/8/PP3PPP/RNBQKBNR w KQkq - 0 5 | 1.06 |
| Benoni | rnbqnrk1/pp3pbp/3p2p1/2pP4/4P3/2N5/PP1NBPPP/R1BQK2R w KQ - 3 10 | 1.02 |
| Caro-Kann | rn1qkb1r/pp2pp1p/2p2p2/5b2/3P4/2P5/PP3PPP/R1BQKBNR w KQkq - 1 7 | 0.91 |
| Alekhine's defence | rnbqkb1r/ppp1pppp/1n1p4/4P3/3P4/5N2/PPP2PPP/RNBQKB1R w KQkq - 2 5 | 0.92 |
| Vienna | rnbqk2r/pppp1ppp/8/2b1p3/2B1P1n1/P1N2N2/1PPP1PPP/R1BQK2R b KQkq - 4 5 | -0.89 |
| Sicilian | r2q1rk1/1p2bppp/p1bppn2/8/3QP1P1/2N1BP2/PPP4P/2KR1B1R w - - 3 13 | 0.78 |
| Sicilian | r1bq1rk1/pp2bppp/2nppn2/8/3NPP2/2N1BQ2/PPP3PP/R3KB1R w KQ - 3 9 | 1.06 |
| Benoni defence | r1bqr1k1/pp1n1bpp/3p1np1/2pP4/4P3/2NB1N1P/PP3PP1/R1BQ1RK1 w - - 5 11 | 0.93 |
| QGD Slav | r2qkb1r/pp3ppp/2p1pn2/7b/PnBPP3/2N1BN2/1P3PPP/2RQK2R w Kkq - 4 10 | 1.02 |
| Nimzo-Indian | rnbqk2r/p2p1ppp/4pn2/1pp5/1bPP4/2N2NP1/PP2PP1P/R1BQKB1R w KQkq - 0 6 | 0.91 |
| Sicilian | rnbqk2r/3nbppp/p2pp3/1p6/3NP1P1/2N1BP2/PPPQ3P/R3KB1R w KQkq - 1 10 | 0.92 |
| QGD | rnbqkbnr/pp4pp/2p1p3/3p4/2PP2Q1/2N1P3/PP3P1P/R1B1KB1R b KQkq - 0 6 | 0.89 |
| King's Indian | r1bq1rk1/1pp2pbp/p1np1np1/4p3/2PPP3/2N1BP2/PP1QN1PP/R3KB1R w KQ - 0 9 | 0.93 |
| Sicilian | r1b1k1nr/pp1pppbp/1qn3p1/1Bp5/4P3/2P2N2/PP1P1PPP/RNBQ1RK1 w kq - 1 6 | 1.07 |
| Pirc | rnbq1rk1/ppp1ppbp/3p1np1/7P/3PP3/2N5/PPP1BPP1/R1BQK1NR b KQ - 0 6 | 0.77 |
| King's Indian | rnbq1rk1/pp4bp/3ppnp1/2p5/2P1PPP1/2N5/PP2B2P/R1BQK1NR b KQ - 0 9 | -1.05 |
| KGA | rnbqkbnr/pppp2pp/8/4Pp2/2B2p2/8/PPPP2PP/RNBQK1NR b KQkq - 0 4 | -0.95 |
| Trompovsky attack (Ruth, Opovcensky opening) | rnbqkb1r/pppppp1p/8/6p1/3PnB2/8/PPP1PPPP/RN1QKBNR w KQkq - 0 4 | 0.82 |
| Sicilian | rnbqk2r/1p2bppp/p2ppn2/8/3NP3/2N1BQ2/PPP2PPP/2KR1B1R b kq - 1 8 | 0.94 |
| King's Indian | 1rbqr1k1/ppp1ppbp/2np1np1/6B1/2PPP3/2N2P2/PP1QN1PP/R3KB1R w KQ - 7 9 | 0.82 |
| St. George (Baker) defence | rn1qkbnr/1bpppppp/p7/1p6/3PP3/3B1N2/PPP2PPP/RNBQK2R b KQkq - 3 4 | 0.85 |
| KP | r1bqkbnr/pppppppp/2n5/8/1P2P3/8/P1PP1PPP/RNBQKBNR b KQkq - 0 2 | -0.62 |
| Dutch | rnbqk2r/ppp1p1b1/3p1npp/5p2/3P1B1P/2N1PQ2/PPP2PP1/R3KBNR w KQkq - 2 7 | 1.07 |
| Trompovsky attack (Ruth, Opovcensky opening) | rnbqkb1r/pp1p1pp1/4pn1p/8/3pP2B/8/PPPN1PPP/R2QKBNR w KQkq - 0 6 | 0.74 |
| French | rnb1kbnr/pppq1ppp/4p3/3pP3/3P2Q1/2N5/PPP2PPP/R1B1KBNR w KQkq - 3 6 | 0.95 |
| King's Indian | rnbq1rk1/ppp1ppb1/3p1npp/6B1/2PPP3/2N4P/PP3PP1/R2QKBNR w KQ - 0 7 | 0.77 |
| Modern defence | rnbqk2r/ppp1npbp/3p2p1/3Pp3/2P1P3/2N5/PP3PPP/R1BQKBNR w KQkq - 1 6 | 1.07 |
| Sicilian | rnbqkbnr/p2ppppp/1p6/2p5/4P3/2N5/PPPP1PPP/R1BQKBNR w KQkq - 0 3 | 0.76 |
| Sicilian | 3q1rk1/pp1bppbp/3p1np1/8/2rNP2P/2N1BP2/PPPQ2P1/2KR3R w - - 0 14 | 0.92 |
| Queen's pawn game | r2qkb1r/pppnpppp/3p1n2/5b2/3P4/2N2NP1/PPP1PP1P/R1BQKB1R w KQkq - 1 5 | 0.83 |
| Queen's Indian accelerated | rn1qkb1r/pbpppp1p/1p3np1/8/2PP4/2N5/PPQ1PPPP/R1B1KBNR w KQkq - 0 5 | 0.99 |
| Sicilian | r1bqk1nr/pp1pp1bp/6p1/5p2/2BpP3/2N2Q2/PPPP1PPP/R1B1K2R w KQkq - 0 8 | 0.71 |
| King's Indian | rnbqk2r/ppp3bp/3p2p1/3Pp2n/2P1Pp2/2N2P2/PP1Q1BPP/R3KBNR b KQkq - 1 9 | 0.92 |
| Caro-Kann | rnbqkbnr/pp2pppp/2p5/7Q/4p3/1B6/PPPP1PPP/RNB1K1NR b KQkq - 1 4 | -0.90 |
| QGA | r2qkb1r/ppp1pppp/1nn5/3bP3/3P4/2NBB3/PP3PPP/R2QK1NR w KQkq - 7 9 | 0.99 |
| Philidor's defence | rn1qkbnr/ppp2ppp/3p4/4p3/2B1P1b1/5N2/PPPP1PPP/RNBQK2R w KQkq - 2 4 | 0.90 |
| Caro-Kann | rnbqkb1r/pp2pp2/2p2p2/7p/2BP4/8/PPP2PPP/R1BQK1NR w KQkq - 0 7 | 1.03 |
| Pirc defence | r1bqkb1r/pppnpp1p/3p1np1/8/3P1P2/2N5/PPP2P1P/R1BQKBNR w KQkq - 0 5 | 0.88 |
| French | r1bqkb1r/ppp1nppp/2n1p3/3pP3/3P3P/2N5/PPP2PP1/R1BQKBNR b KQkq - 0 5 | 0.84 |
| Trompovsky attack (Ruth, Opovcensky opening) | rnb1kb1r/pp1ppppp/1q6/2pP4/7P/8/PPPBPPP1/R2QKBNR b KQkq - 0 6 | 0.99 |
| French | rnbqk1nr/pp3ppp/4p3/b2pP3/1p1P4/P1N5/2P2PPP/R1BQKBNR w KQkq - 0 7 | 1.04 |
| Robatsch (modern) defence | r1bqk2r/1ppnnpbp/p2pp1p1/8/3PP3/2PB1N2/PP3PPP/RNBQR1K1 w kq - 2 8 | 0.96 |
| King's Indian | rnbq1rk1/pp3pbp/3ppnp1/2pP4/2P1P1PP/2N5/PP3PP1/R1BQK1NR b KQ - 0 8 | -0.82 |
| Ruy Lopez | r1bq1rk1/2p2ppp/pn1p1b2/np2p3/3PP3/1BP2N1P/PP3PP1/R1BQRNK1 w - - 5 13 | 1.00 |
| Vienna game, Max Lange defence | r1bqkbnr/pppp2pp/2n5/4pp2/2B1P3/2N5/PPPP1PPP/R1BQK1NR w KQkq - 0 4 | 0.95 |
| English | r1bqkbnr/ppp3pp/3p4/3Pnp2/2P1p3/2N2NP1/PP2PP1P/R1BQKB1R w KQkq - 1 7 | 1.05 |
| Alekhine's defence | rnbqkb1r/ppp1p1p1/1n1p2p1/4P3/2PP1P2/8/PP4PP/RNBQKBNR w KQkq - 0 6 | 0.88 |
| King's Indian | r1bq1rk1/ppp1pn1b1/3p1npp/P2Ppp2/1PP1P3/2N2N2/4BPPP/R1BQR1K1 w - - 0 13 | 1.01 |
| Caro-Kann | r2qkb1r/pp1nnppp/2p1p1b1/3pP3/P2P4/5N2/1PP1BPPP/RNBQ1RK1 w kq - 1 8 | 1.05 |
| French | rnbqk2r/ppp1bppp/4pn2/8/8/1PN5/PBPPQPPP/R3KBNR w KQkq - 3 7 | 1.05 |
| Sicilian | rn1qkb1r/pp1b1ppp/3p1n2/4p3/3QP3/1PN5/PBP2PPP/R3KBNR w KQkq - 0 7 | 1.06 |
| Sicilian | r1bqkb1r/pp1ppppp/2n5/2p1P3/8/2P2N2/PPP2PPP/R1BQKB1R w KQkq - 1 6 | 1.12 |
| Blumenfeld counter-gambit, Dus-Chotimursky variation | rnb1kb1r/p2p1ppp/4pn2/qppP2B1/2P5/5N2/PP2PPPP/RN1QKB1R w KQkq - 2 6 | 0.91 |
| Two knights defence | r1bqkb1r/p4pp1/2p4p/n2p3p/4N3/3B4/PPPP1PPP/RNBQK2R w KQkq - 2 10 | 0.89 |
| Czech Benoni defence | r1bq1rk1/pp1nbppp/3p1n2/2pPp3/2P1P3/2N2NP1/PP3PBP/R1BQK2R b KQ - 2 8 | 0.97 |
| King's Indian defence, 3.Nc3 | rnbq1rk1/pp3pb1/3p1npp/2pPp3/2P4B/2N5/PP1NPPPP/R2QKB1R w KQ e6 0 9 | 1.22 |
| Centre game | r1bqk2r/ppppnpbp/2n3p1/8/4P3/2N1Q3/PPPB1PPP/R3KBNR w KQkq - 4 7 | 0.93 |

| Opening | FEN | Value |
|---|---|---|
| Sicilian | r2qkb1r/1b1n1pp1/p2ppn1p/1p6/3NP1P1/2N1BP2/PPPQ3P/2KR1B1R w kq - 2 11 | 0.98 |
| KGA | rnbqkbnr/pppp1ppp/8/8/4Pp2/5Q2/PPPP2PP/RNB1KBNR b KQkq - 1 3 | -1.06 |
| King's Indian | r1bq1rk1/ppn1ppbp/3p1np1/2pP2B1/2P1P3/2N5/PP1QBPPP/R3K1NR w KQ - 1 9 | 1.05 |
| Pirc | rnbq1rk1/p1p1ppbp/1p1p1np1/8/3PPP2/2N2N2/PPPB2PP/R2QKB1R w KQ - 0 7 | 1.17 |
| English opening | rnbqkb1r/ppp2pp1/3p1n2/7p/2P1p2N/2N3P1/PP1PPP1P/R1BQKB1R w KQkq - 2 6 | 1.00 |
| Dutch defence | rnbqk2r/ppppp1bp/6pn/5p2/2PP4/2N3P1/PP2PP1P/R1BQKBNR w KQkq - 1 5 | 0.92 |
| French | rnbqkb1r/pp1n1p1p/4p1p1/2ppP3/3P1P2/2P5/PP1N2PP/R1BQKBNR w KQkq - 0 7 | 1.00 |
| Sicilian | r1bqkb1r/pp1npppp/3p1n2/8/3NP3/2N5/PPP2PPP/R1BQKB1R w KQkq - 3 6 | 0.88 |
| King's Indian | r1bq1rk1/ppp2pbp/n2p1np1/3Pp3/2P1P2/2N5/PP2BPP1/R1BQK1NR w KQ - 1 8 | 1.13 |
| Budapest | r1b1k2r/pppp1ppp/2n5/3QP3/2P2Bn1/q1P2N2/P3PPPP/R1BK2R w - 3 9 | 0.91 |
| Blackmar gambit | rn1qkb1r/pp3ppp/2p1pn2/4Nb2/2BP4/2N5/PPP3PP/R1BQK2R w KQkq - 0 8 | -0.78 |
| KP | r1bqkb1r/pppppppp/2n2n2/8/4P3/2N5/PPPP1PPP/R1BQKBNR w KQkq - 3 3 | 0.75 |
| King's Indian | rn1q1rk1/1ppn1pbp/3p2p1/p2Pp3/2P1P1b1/2N1BN2/PP1QBPPP/R3K2R w KQ - 0 10 | 1.22 |
| Ruy Lopez | r1b2rk1/2q1bppp/p2p1n2/np1Pp3/4P3/5N1P/PPBN1PP1/R1BQR1K1 w - - 1 15 | 1.00 |
| Scandinavian (centre counter) defence | rn2kb1r/pp2pppp/p2n2n2/q7/2BP2b1/2N2N1P/PPP2PP1/R1BQK2R b KQkq - 0 7 | 0.98 |
| King's Indian defence | rnbqk2r/p2pppbp/5np1/1PpP4/7P/8/PP2PPP1/RNBQKBNR w KQkq - 1 6 | 1.00 |
| French | rnbqk2r/pp1n1ppp/4p3/2ppP3/1b1P4/2NQ4/PPP1NPPP/R1B1KB1R w KQkq - 0 7 | 1.15 |
| Giuoco Piano | r1bqk1nr/pppp1pp1/2n4p/2b1p3/2BPP3/5N2/PPP2PPP/RNBQ1RK1 b kq - 0 5 | 0.95 |
| Queen's Indian | r2qkb1r/p1pp1ppp/bpn1pn2/8/2PP4/PQ3N2/1P2PPPP/RNB1KB1R w KQkq - 3 6 | 0.97 |
| King's Indian | r1bqnrk1/pp1n1pbp/2pp2p1/4p3/2PPP3/2N2NPP/PP3PB1/R1BQ1RK1 w - - 1 10 | 0.96 |
| French | rnbqkb1r/p1p1nppp/1p2p3/3pP3/3P4/5N2/PPP2PPP/RNBQKB1R w KQkq - 2 5 | 1.06 |
| Sicilian | r1bqkb1r/pp1p1ppp/2n1pn2/1Bp5/4P3/2N2N2/PPPP1PPP/R1BQK2R w KQkq - 0 5 | 1.05 |
| Sicilian | rnb1kb1r/1pq2ppp/p2ppn2/8/2PNP3/2N5/PP2BPPP/R1BQK2R w KQkq - 2 8 | 1.03 |
| Czech Benoni defence | r1bq1rk1/1p1nbppp/p2p1n2/2pPp3/2P1P3/2N2N2/PP2BPPP/R1BQ1RK1 w - - 0 9 | 1.29 |
| Ruy Lopez | r3r1k1/2qbbppp/p2p1n2/nppPp3/4P3/2P2N1P/PPBN1PP1/R1BQR1K1 w - - 1 14 | 1.01 |
| Dutch | rnbq1rk1/1pp1b1pp/3ppn2/p4p2/1PPP4/5NP1/P3PBPPP/RNBQ1RK1 w - - 0 8 | 0.93 |
| QGA, Smyslov variation | r1bq1rk1/ppp1ppbp/2n2np1/8/2BP4/2N1PN2/PP3PPP/R1BQ1RK1 w - - 5 8 | 0.90 |
| English | r2q1rk1/2pn1ppp/1p1ppn2/p5B1/2PP4/P1QBP3/1P1N1PbP/R3K2R w KQ - 0 12 | 0.95 |
| Pirc defence | r1bqk2r/ppp1ppbp/2n1np1/8/3PP3/2N1B3/PPPQ1PPP/R3KBNR w KQkq - 4 6 | 0.99 |
| King's Indian | r1bq1rk1/pppp1nbp/3p2p1/3Pp3/2P1Pn2/2N2N2/PP1BBPPP/R2QR1K1 w - - 5 11 | 0.90 |
| French | rnbqkbnr/p1p2ppp/1p2p3/3pP3/3P4/P1N5/1PP2PPP/R1BQKBNR w KQkq - 1 6 | 0.96 |
| Sicilian | r1bqkbnr/4nppp/p2p4/1p1Np3/4P3/N7/PPP2PPP/R1BQKB1R w KQkq - 2 9 | 0.89 |
| Sicilian | r1bq1rk1/1p2bppp/p2ppn2/n7/3NP3/1BN1B3/PPP1QPPP/2KR3R w - - 2 11 | 0.88 |
| Queen's pawn | r1bqkb1r/pp2pppp/2n2n2/2pp4/3P1P2/2PBP3/PP4PP/RNBQK1NR b KQkq - 0 5 | -0.88 |
| KGA | rnbqkbnr/ppp2ppp/8/3N4/4Pp2/8/PPPP2PP/R1BQKBNR b KQkq - 0 4 | -1.15 |
| Benoni defence | rnb1kb1r/1p2qp1p/p2p1np1/2pP4/P7/2N2N1P/1P2PPP1/R1BQKB1R w KQkq - 1 9 | 1.06 |
| Gruenfeld defence | rnbqk2r/ppp1ppbp/5np1/3P4/3P4/2N5/PP2PPP1/R1BQKBNR w KQkq - 1 7 | 1.10 |
| Sicilian | r1b1kb1r/pp3ppp/1qnppn2/8/4P3/1NN3P1/PPP2P1P/R1BQKB1R w KQkq - 0 8 | 0.97 |
| Caro-Kann | rn1qkb1r/pp2pppp/2p5/3pPb2/4n3/5N2/PPPPNPPP/R1BQKB1R w KQkq - 3 6 | 1.15 |
| King's Indian | r1bq1rk1/1pp1ppbp/2np1np1/p7/2PPP3/2N1BP2/PP2N1PP/R2QKB1R w KQ - 0 8 | 0.78 |
| Dutch, 2.Bg5 variation | rnbqkbnr/pp1pp2p/5pB1/3P4/2N5/PPPP1PPPP/R1BQKBNR w KQkq - 0 4 | 1.02 |
| Sicilian | rnb1kb1r/ppq1pppp/3p1n2/8/3NP3/2N5/PPP2PPP/R1BQKB1R w KQkq - 3 6 | 1.03 |
| Benko's opening | rnbqk1nr/pp1pp1bp/6p1/2pP1p2/8/6P1/PPP1PBPP/RNBQK1NR w KQkq - 0 5 | 1.14 |
| Ruy Lopez | r1bq1rk1/4bppp/p1pp1n2/np2p3/4P3/2P2N1P/PPBR1PP1/RNBQR1K1 w - - 0 11 | 0.92 |
| King's Indian | r1bqn1k1/ppp1n1bp/3p1r2/2PPp1p1/4Pp2/2N2P2/PP2BBPP/R2QNRK1 w - - 1 14 | 0.99 |
| Gruenfeld | rnbqkb1r/ppp1ppp/1n4p1/8/3PP3/2N5/PP3PPP/R1BQKBNR w KQkq - 1 6 | 1.22 |
| Giuoco Piano | r1b1k1nr/ppppqppp/2n5/2b1p3/2B1P3/2P2N2/PP1P1PPP/RNBQK2R w KQkq - 1 5 | 1.03 |
| QGD semi-Slav | rnbqk2r/p4ppp/2p1pn2/1p4B1/PbpPP3/2N2N2/1P3PPP/R2QKB1R w KQkq - 1 8 | 0.79 |
| Bogo-Indian defence | r1bq1rk1/1pp1nppp/4pn2/p2p4/2PP4/P2BPN2/1PQN1PPP/R3K2R w KQ - 1 10 | 1.27 |
| Queen's Indian | rn1q1rk1/p1ppb1pp/1p2p3/5p2/2PP1B1P/3QPN2/PP3PP1/R4K1R w - - 0 13 | 1.03 |
| King's Indian | rnb2rk1/pp3ppp/2pp1np1/q3p1B1/2PPP3/2N2P2/PP1QN1PP/R3KB1R w KQ - 2 9 | 1.16 |
| Vienna | b1kr1bnr/p1ppqp1p/Ppn5/1N4p1/3Ppp2/5N2/1PP2KPP/R1BQ1B1R w - - 1 12 | -0.57 |
| Ruy Lopez | 1rbn1rk1/2q1bppp/3p1n2/1p1Pp3/1Pp1P3/2P2N1P/2BN1PP1/R1BQR1K1 w - - 0 17 | 1.05 |
| QGA | r1b2rk1/pp1nppbp/2p2np1/7q/2QPP3/2N2NP1/PP3BP/R1B2RK1 w - - 1 11 | 0.88 |
| English opening | rnbqk1nr/ppp3bp/3p4/4pp2/2P5/2N3P1/PP1P1PBP/R1BQK1NR w KQkq - 0 7 | 1.06 |
| French | rnbqkb1r/1ppn1ppp/p3p3/3pP3/3P4/5N2/PPPPN1PP/R1B1KBNR w KQkq - 1 6 | 1.00 |
| Sicilian | r1b2rk1/pp2bppp/2nppn2/q7/3NP3/1BN1B3/PPP1QPPP/R3K2R w KQ - 6 10 | 1.08 |
| French | rnbqk2r/pp3ppp/4p3/2ppPn2/3P2Q1/P1P5/2P2PPP/R1B1KBNR w KQkq - 3 8 | 1.02 |
| Philidor's defence | rn1qkbnr/ppp2ppp/3p4/4p3/3PP1b1/5N2/PPP2PPP/RNBQKB1R w KQkq - 1 4 | 1.23 |
| Scandinavian (centre counter) defence | r1bqkbnr/pp2pppp/2n5/8/8/8/PPPP1PPP/RNBQKBNR w KQkq - 0 4 | 0.89 |
| Sicilian | r1bqk2r/1p1nbppp/p1npp3/6P1/3NP2P/2N1B3/PPP1QP2/R3KB1R b KQkq - 1 10 | 0.89 |
| Sicilian | r1b1kb1r/pp3ppp/2nP4/q1p1p3/1nP1N3/PQ3N2/1P1P1PPP/R1B1KB1R w KQkq - 0 10 | 1.09 |
| French | r1b1k2r/pp1nbppp/1qn1p3/3pP3/3P4/3B1N2/PP2NPPP/R1BQ1K1R w kq - 3 11 | 0.85 |
| QGD semi-Slav | r1bqkb1r/1p1n1ppp/2p1pn2/p1pPp4/1P1P4/2N1PN2/P4PPP/R1BQKB1R w KQkq - 0 8 | 0.96 |
| Ruy Lopez | r2q1rk1/2p1bppp/p1n5/1pnpP3/6b1/2P2N2/PPBN1PPP/R1BQR1K1 w - - 7 13 | 0.97 |
| Ponziani | r1b1kb1r/ppp2ppp/2n2n2/8/3P4/2P1PB1q/PP1N3P/RNBQK2R w KQkq - 2 10 | -0.75 |
| Old Indian defence | r2qkb1r/pppnpppp/3p1n2/5b2/2PP4/1Q3N2/PP2PPPP/RNB1KB1R w KQkq - 4 5 | 0.95 |
| Sicilian | 5rk1/pp1bpbbp/3p1np1/q7/2r1P1P1/2N1BP2/PPPQN3/2KR3R w - - 3 17 | 0.86 |
| QGD semi-Slav | r1b1kb1r/5ppp/pqp1pn2/3p4/Q2P4/4PN2/PP1B1PPP/R3KB1R w KQkq - 1 11 | 1.10 |
| Scandinavian (centre counter) defence | r1b1kb1r/ppp1pppp/2n2n2/q7/3P4/2N2N2/PPP2PPP/R1BQKB1R w KQkq - 3 6 | 1.18 |
| QGD | r2q1rk1/pb1nbppp/1p3n2/3p2B1/3p3P/2NBPN2/PPQ2PP1/2KR3R w - - 0 12 | 1.03 |
| KGA | rn2kbnr/ppp2ppp/3pb3/8/2B1Pp1q/2N5/PPPPPQ1PP/R1B2KNR b kq - 3 6 | -0.76 |
| QGD | r1bqrnk1/p4ppp/2pb1n2/1p1p2pN/3P3P/2NBP3/PPQB1PP1/2KR3R b - - 1 12 | -0.99 |

| | | |
|---|---|---|
| Robatsch (modern) defence | rn1qk1nr/1bp2pb1/3pp1pp/p7/Pp1PP2P/4BP2/1PPQN1P1/2KR1BNR w kq - 0 11 | 0.99 |
| King's Indian | r1b2rk1/pp1nqpbp/2pp2p1/4p2n/2PPP3/2N1BNPP/PP3PB1/R2Q1RK1 w - - 3 11 | 1.20 |
| King's Indian | r1bq1rk1/ppp2pbp/2np1np1/4p3/2PPP3/2N1BP2/PP2N1PP/R2QKB1R w KQ - 0 8 | 1.12 |
| Nimzo-Indian | rnbqk2r/pp3ppp/4p3/2pp4/1bPPn3/2N1PN2/PPQ2PPP/R1B1KB1R w KQkq - 0 7 | 1.17 |
| Ruy Lopez | r1b1k2r/2q1bppp/p1np1n2/1pp1p3/3P3/2P2N2/PPBN1PPP/R1BQR1RK1 w kq - 3 12 | 1.02 |
| Ruy Lopez | r1bqkb1r/pp3ppp/2p3n1/8/3pP3/1B6/PPP2PPP/RNBQ1RK1 w kq - 1 10 | 0.84 |
| QGD | r1b1k1nr/pp1n2pp/2pbp2q/3p1p2/2PP4/2N1P3/PP2BPPP/R1BQNRK1 w kq - 6 9 | 1.19 |
| QGD | r1bq1rk1/pp2bppp/2n3n1/4P3/2Pp4/3B1N2/PP1N1PPP/R1BQR1K1 w - - 8 11 | 1.10 |
| Sicilian, Szen ('anti-Taimanov') variation | r2qkb1r/1p3p1p/p2pbp2/4p3/3nP3/N1N5/PPP2PPP/R2QKB1R w KQkq - 2 12 | -0.90 |
| King's Indian | r1bq1rk1/pppnppbp/3p1np1/8/2PPPP2/2N2N2/PP4PP/R1BQKB1R w KQ - 3 7 | 0.98 |
| Pirc | rnbq1rk1/pp2ppbp/2pp1np1/8/3PPP2/2N2N2/PPP1B1PP/R1BQK2R w KQ - 0 7 | 1.06 |
| Dutch-Indian, Alekhine variation | rn1q1rk1/pbppb1pp/1p2p3/5p2/2PPn3/2N2NP1/PPQBPPBP/R4RK1 w - - 2 10 | 1.14 |
| Old Indian | r1b2rk1/ppqnbppp/2pp1n2/4p3/2PPP3/2NB1N1P/PP3PP1/R1BQ1RK1 w - - 1 9 | 1.09 |
| Sicilian | r1bq1rk1/ppn3bp/2np2p1/4ppP1/4PP1P/2N1B3/PPP1N3/R2QKB1R w KQ f6 0 13 | -1.06 |
| Ruy Lopez | r3k2r/1bpq1pp1/pbnp1n1p/1p2p3/3PP2B/1BP2N2/PP3PPP/RN1QR1K1 w kq - 2 12 | 1.17 |
| Queen's Indian | r2qkb1r/2pp1ppp/ppn1p1b1/8/2PP4/P3P1P1/1P1N1P1P/R1BQKB1R w KQkq - 0 10 | 1.16 |
| King's Indian | r1b1qrk1/1ppn1pbn/p2p2pp/3Pp3/2P1P2B/2N5/PP1NBPPP/R2Q1RK1 w - - 4 12 | 0.98 |
| King's Indian | rnbq1rk1/pp1p1pbp/5np1/2pPp3/2P5/2N3P1/PP2PPBP/R1BQK1NR w KQ e6 0 7 | 1.11 |
| Czech Benoni defence | r1bqnrk1/pp1nbppp/3p4/2pPp3/2P1P3/P1N2N2/1P2BPPP/R1BQ1RK1 b - - 0 9 | 0.98 |
| English | rnbqk2r/ppn1ppbp/6p1/2p5/3PP3/5NP1/PP3PBP/RNBQK2R w KQkq - 1 8 | 1.13 |
| Benoni | r1bqr1k1/pp1n1pbp/3p4/2pP3p/4P3/2N5/PPQN1PPP/R1B2RK1 w - - 0 13 | 1.11 |
| QGA | 1n1qkbnr/rpp2ppp/p3p3/7b/2BP4/1Q2PN1P/PP3PP1/RNB1K2R w KQk - 3 8 | 1.04 |
| Sicilian | r1b1k2r/1p1pnppp/p1n3q1/4p3/4P3/2N5/PPP2PPP/R1BQKB1R w KQkq - 5 10 | 1.05 |
| French | r1b1k1r1/pp2np1Q/4p3/2qpn3/8/P1P5/2PBBPPP/R3K1NR w KQq - 0 13 | 1.05 |
| Queen's pawn game | r2qkb1r/pppn1ppp/3p1p2/8/3PP1b1/5N2/PPPN1PPP/R2QKB1R w KQkq - 1 7 | 0.91 |
| Caro-Kann | r1b2rk1/pp2bppp/2nqpn2/8/3P4/P1NB1N2/1P3PPP/R1BQR1K1 w - - 1 12 | 1.10 |

