# OpenReview forum: "Mastering Board Games by External and Internal Planning with Language Models"
_ICML.cc/2025/Conference — ICML 2025 spotlightposter_

### Official Review · Reviewer_2VtU · 2025-03-12

**Overall Recommendation:** 5

**Summary:**

- This paper focuses on game playing for board games with LLMs
	- it compares external search, where model acts as a proposal function for a symbolic search algorithm, with internal search, where the model is trained on search trajectories to perform search itself
- The paper claims three contributions
	- Multi action-value model, which can be used to score action-value pairs
	- External search, where the MAV is used with MCTS to guide search
	- Internal search, where the search procedure is used to generate trajectories that are then used to train a model to directly do search itself.
- Models are evaluated across different board games, including Chess, variants of chess, Connect Four, and Hex)
- MAV is trained on text representations of the game whih include the game being played and the input state/legal output space
	- MAV models are trained from scratch on a custom text representation
- MAV is integrated into an MCTS solver for external search
	- MAV is used to predict state transitions and actions during the rollout phase

- MAV is also trained with additional search information in the prompt on search traces from symbolics solvers to internalize search
- MAV models are compared agaisnt numerous chess-playing baselines, achieving competitive ELO.
	- MAV-MCTS consistently performs better than MAV alone, and larger models perform better.
	- External MCTS improves with more simulations
	- The internal search MAV improves with increased search budget, scaling w.r.t. number of tokens.

## update after rebuttal
The rebuttal has addressed my remaining questions, and I will maintain my score of 5.

**Claims And Evidence:**

- The claims are generally supported by results:
	- MAV models combined with symbolic search perform better than a variety of baselines across multiple games
	- MAV can be trained to internalize search

**Essential References Not Discussed:**

Paper covers essential references

**Experimental Designs Or Analyses:**

The experimental designs are sensible and the token scaling analysis supports the results well.

**Methods And Evaluation Criteria:**

- The method is described clearly and Fig 1 is helpful in showing the input/output format.
- The evaluation criteria seem robust and the method is evaluated on multiple games.No theoretical claims made.

**Other Comments Or Suggestions:**

N/A

**Other Strengths And Weaknesses:**

- Strengths:
	- By including the state representation and action in the output space, MAV can act not only as a kind of Q value function but also as a world model and policy.

- Weaknesses:
	- It's not clear what the relative gains for Connect Four and Hex are, since there are no baselines there.
	- MAV training requires access to a large amount of training data; it's not clear how it would generalize to real-world domains where there is no simulator/game engine. This is mentioned in the limitations.
	- Values are discretized; it would be nice to see how sensitive the method is to the number of bins here.
	- The MCTS rollout requires a parser that can verify/parse illegal actions, which reduces hallucination. It's also not clear here how one would implement this outside of a game environment with manually-defined rules.

**Questions For Authors:**

The current method outputs a state, top-5 actions etc. as a kind of CoT output. One of the big benefits of CoT is the ability to average/marginalize across reasoning paths, e.g. with self-consistency. Have you explored something like self-consistency on the best action prediction -- is this the same as just running more simulations?

**Relation To Broader Scientific Literature:**

The paper is positioned well w.r.t. prior work on learning to search/game playing. The related work section is in the appendix but I think this decision makes sense as it is quite extensive.

**Theoretical Claims:**

No theoretical claims made.

---

> ### Author Rebuttal · Authors · 2025-03-31
>
> Thank you for taking the time to carefully review our paper and for the positive feedback!
>
> We have not tried self-consistency since we currently use greedy decoding, which we found to improve the strength of the MAV models compared to sampling. However, sampling multiple outputs from the internal search model and using self-consistency is a very promising idea for further improving the model strength! One piece of evidence that this approach would work is the performance of MAV with mean scoring, which outperforms MAV with greedy decoding, and can be viewed as a form of weighted voting.

---

### Official Review · Reviewer_a6GX · 2025-03-14

**Overall Recommendation:** 4

**Summary:**

This paper enhances LLM planning capabilities in board games through two approaches: external search (model-guided MCTS without game engines) and internal search (in-context linearized search trees). Using a pre-trained Multi-Action-Value (MAV) model for Chess, Chess960, Connect Four, and Hex, both approaches significantly improve playing strength, with external search achieving Grandmaster-level chess performance using human-comparable search budgets.

**Claims And Evidence:**

The performance claims are well-supported by comprehensive empirical evidence showing increasing Elo ratings with larger search budgets. Tournament results against Stockfish at various strengths provide credible evidence for the Grandmaster-level performance claim, with appropriate calibration between internal and external Elo ratings.

**Essential References Not Discussed:**

N/A

**Experimental Designs Or Analyses:**

The tournament methodology using random pairings is sound with sufficient sample sizes. Analysis of model capabilities on both in-distribution and out-of-distribution positions demonstrates robustness. The experiments show a clear relationship between computational resources and performance.

**Methods And Evaluation Criteria:**

The methodology is appropriate and comprehensive.

**Other Comments Or Suggestions:**

N/A

**Other Strengths And Weaknesses:**

Strengths:

The paper presents a unified approach that integrates world modeling, policy, and value prediction in a single model, streamlining what previously required separate components. Its implementation of state tracking without external game engines enables self-contained operation, representing a significant advancement in model independence. The methodology demonstrates successful generalization across multiple diverse board games, suggesting broad applicability of the core techniques. The achievement of Grandmaster-level chess performance with human-comparable search budgets marks an important milestone in LLM reasoning capabilities. The innovative linearization technique for search trees enables effective in-context planning, creating a pathway for self-contained reasoning in language models.

Weakness:

The internal search performance remains inferior to external search, indicating room for further refinement. What's the bottleneck here?

Meanwhile, the paper provides insufficient analysis of computational efficiency between the approaches, making it difficult to assess practical trade-offs for real-world applications.

**Questions For Authors:**

1. How does the computational efficiency of internal vs. external search compare? At what point (if any) does internal search become more efficient?

2. Have you explored hybrid approaches combining both search methods?

3. How dependent is the approach on high-quality annotated game data? Do you have some intuitive explanations?

4. What are the primary failure modes of each approach?

5. A side question is that what modifications would be needed to extend this to imperfect information games?

**Relation To Broader Scientific Literature:**

See above.

**Theoretical Claims:**

The paper doesn't make substantial theoretical claims requiring formal proofs.

---

> ### Author Rebuttal · Authors · 2025-03-31
>
> Thank you for taking the time to carefully review our paper and for the positive feedback!
>
> > The internal search performance remains inferior to external search
>
> We evaluated internal search only up to breadth=4, depth=2, resulting in a search budget of 21 nodes, whereas external search was evaluated using a search budget of 100 to 2000 nodes. Moreover, the underlying search algorithms are different (minimax and MCTS, respectively). Therefore, the performance gains are not directly comparable. The main bottleneck for scaling up internal search to larger search budgets is scaling up the context length and the inference latency increase that this would entail. Nevertheless, even while being more challenging to scale up to large search budgets, we believe that our results provide interesting insights into the internal search capabilities of LLMs.
>
> 1. The current advantages of external search are that (i) it is (depending on the exact search method and its implementation) potentially more easily parallelizable, whereas the internal search constructs the tree sequentially, and requires some extra tokens to capture the tree structure and the decisions pertaining to its expansion, (ii) the number of FLOPS required for internal search grows quadratically with the number of positions evaluated due to the quadratic cost of attention in transformer models, whereas it grows linearly for external search. Yet, this strength may also (contextually) be a weakness given that running the external search requires numerous LLM calls, whereas the invocation of internal search requires a single call, which may be advantageous when/if bottlenecked by the query volume, and potential delays in query completion.
> 2. We haven’t explored hybrid approaches yet, but that is a really interesting suggestion! A hybrid approach may be able to benefit from a degree of parallelism that is possible to achieve through the external controller, and yet expand sub-trees (rather than single moves) in different branches via internal search when that is potentially useful. It is an open question how best to combine the two, though hybrid approaches (of a slightly different nature) have been successfully utilized in chess engines previously, when it comes to different ways in which neural networks can be incorporated, given that certain positions may require either deep tactical calculations (big trees) or intricate strategic play (shallow trees, but harder/higher quality policies and evaluation).
> 3. Since the performance of the MAV model is capped by the performance of the game engine used to train it, the current approach depends on the ability to obtain or generate high-quality game-play data for the model to train on. There are many games where this is possible/easy to achieve. For other games, one can use techniques such as self-play as was done in AlphaZero.
> 4. For chess and chess960, the models are somewhat susceptible to drawing a winning game via the 50-move rule or 3-fold repetition. Enforcing these rules requires storing history (or additional variables), which MAV doesn’t do. This happens in particular with the internal search model, which wasn’t trained to predict the quickest win, while in the external search, it can be enforced. Speaking more broadly, these rules are imposed by the external governing bodies (such as international or national chess federations), and under some regulations, 75-move and 5-fold repetition are used. Given this ambiguity, it might be difficult for models trained in a supervised fashion to recognize the pattern even if history was enabled. Additionally, the MAV model is more likely to misevaluate a move, since it does not perform search.
> 5. There is a straightforward (but nontrivial) way to extend these methods to imperfect information games using Information-State Monte Carlo Tree Search ([IS-MCTS, Cowling et al. 2012](https://eprints.whiterose.ac.uk/75048/1/CowlingPowleyWhitehouse2012.pdf)). The value functions would be with respect to ground states (where all the information is revealed), and MAV would be trained in the same way as chess. The model would need to be augmented with an additional generative output that could sample a ground state given the current observation (excluding the hidden information), encoding a distribution that is learnable from data. The search would then start by first sampling possible world states from the generator, and run simulations as is currently done, starting from each one, but then aggregate statistics over information states, similar to the application of IS-MCTS with generative networks in [Li et al. 2023](https://arxiv.org/abs/2302.00797).

---

### Official Review · Reviewer_wDBQ · 2025-03-20

**Overall Recommendation:** 4

**Summary:**

The paper introduces a specialized Multi-Action Value (MAV) 2B model trained exclusively on game data from Chess, Chess960, Connect Four, and Hex. MAV is designed to predict legal moves, track game states, identify the top-k actions, and determine the resulting board state after executing the optimal action. The authors convincingly demonstrate that the MAV model generalizes effectively to previously unseen board configurations.

Additionally, the paper explores two distinct planning methodologies applied to MAV:

1) External Search: The authors implement an external loop utilizing Monte Carlo Tree Search (MCTS), significantly enhancing MAV's performance. This integration achieves results competitive with current state-of-the-art chess engines.

2) Internal Search: The internal search approach leverages distillation from search trees. This technique also shows a consistent and smooth performance improvement as the number of test-time tokens increases.

**Claims And Evidence:**

The claims are largely supported by clear quantitative results.

**Essential References Not Discussed:**

The authors have clearly listed related works and the corresponding contributions.

**Experimental Designs Or Analyses:**

The experimental design is clear, effectively demonstrating the improvements contributed by each component and providing thorough comparisons with the existing engine. Additionally, it includes a detailed error analysis covering precision, recall, legality, and gameplay aspects

**Methods And Evaluation Criteria:**

Methods:
The paper proposed three approaches to enhance LLM planning capabilities and

Evaluation:
For Table 1: It would be beneficial for the authors to clarify the distinction between Internal Elo and External Elo. The current description suggests that External Elo is aligned with human performance, while a brief note that Internal Elo reflects performance among agents would enhance clarity.
It would also be interesting to examine the external Elo ratings for games beyond chess to assess how training primarily on one game generalizes to others.

**Other Comments Or Suggestions:**

line 363: and terminal states,. -> remove the extra comma

**Other Strengths And Weaknesses:**

Strengths:
- Innovative Framework: The paper offers a novel exploration by integrating external and internal search strategies within a single model framework.
- Advanced planning: It convincingly demonstrates how LLMs can be enhanced to plan and reason through complex, sequential decision-making tasks.
- Comprehensive Evaluation: The experimental evaluation is thorough, clearly showcasing the impact of each contribution.


Minor Weakness:
- the MAV model appears to lack essential language capabilities. It would be beneficial to see if the LLM could provide textual reasoning for its solutions or incorporate additional communication functionalities typical of large language models.
- Providing detailed information on the training costs for pretraining the MAV model and executing the internal search distillation process would offer valuable insights into the resource requirements of this approach, particularly when compared to other methods that enhance reasoning and planning.

**Questions For Authors:**

1) Could you elaborate on the computational trade-offs between external and internal search methods, specifically regarding training cost and inference latency?

2) How do you calibrate Internal Elo ratings relative to External Elo ratings, given that the latter are anchored to the CCLR Blitz ratings?

3) How would the MAV model’s performance be affected if it were limited to predicting only the best move? Would it still function effectively as a chess world model? If not, which aspects of its chess world modeling capabilities would be compromised, and to what extent?

4) Do you believe that fine-tuning a language model (already pre-trained on other tasks) on chess or other board game data could enhance its overall reasoning capabilities?

**Relation To Broader Scientific Literature:**

The paper builds upon and extends recent work on:

1) MCTS and AlphaZero/MuZero paradigms for game playing.
2) Internal planning methods such as Chain-of-Thought and Tree-of-Thought approaches.

**Theoretical Claims:**

The discussion is mostly empirical, and while there is a theoretical underpinning (such as mapping centipawn evaluations to win probabilities), no rigorous proofs were provided or checked.

---

> ### Author Rebuttal · Authors · 2025-03-31
>
> Thank you for taking the time to carefully review our paper and for the positive feedback!
>
> 1. Regarding training cost, we used more than an order of magnitude more tokens to train the model on MAV data compared to the tokens used for fine-tuning the MAV model on internal search traces. We will clarify this in the text. Regarding inference, the main computational trade-off is that external search can be easily parallelized, whereas internal search is sequential. However, internal search could be optimized via techniques such as speculative decoding.
> 2. To estimate the external Elo of our agents, we use a 1D linear interpolation (specifically scypi.interpolate.interp1d) to map from internal Elo to external Elo given the known [CCLR Blitz Elos](https://github.com/official-stockfish/Stockfish/commit/a08b8d4) of the common instances of Stockfish in both pools.
> 3. Our preliminary results suggest that limiting MAV to predict only the best move at inference time doesn’t significantly affect its strength. However, training the model to evaluate multiple moves seems to be critical for strength in light of the results from [Ruoss et al. 2024](https://proceedings.neurips.cc/paper_files/paper/2024/file/78f0db30c39c850de728c769f42fc903-Paper-Conference.pdf) and for being able to use the model for external and internal search.
> 4. Regarding fine-tuning an LLM (already pre-trained on other tasks) on board game data to enhance its overall reasoning capabilities: This is a very interesting question and sounds like a plausible hypothesis. However, the volume of data required to make the MAV models play at a really high level likely exceeds the volumes traditionally used in fine-tuning stages by a fair margin. So, if the goal is to retain strong playing strength in such models, this would impose some additional restrictions. Nevertheless, recent evidence from other studies, e.g., [Muennighoff et al. 2025, s1: Simple test-time scaling](https://arxiv.org/abs/2501.19393), indicates that it may be possible to unlock the reasoning capabilities from pre-training by a fairly small amount of data, opening up new possibilities. Whether it is possible to do this in this case (by e.g. pre-training on some subset of games and then generalizing to a broader set through a smaller number of examples, or broader capabilities beyond board games) is an interesting question that would hopefully be explored in more depth in future work.

---

### Decision · Program_Chairs · 2025-05-01

**Decision:**

Accept (spotlight poster)

**Comment:**

This paper introduces a new method, called MAV, which uses LLMs to play board games. The authors propose two methods, an external search, which uses LLM as a policy and value function within MCTS, and an internal search, where the LLM search itself. The results show that the proposed method, especially the external search, achieves significant improvements.

All reviewers unanimously recommend acceptance. The method is novel, and empirical results are strong. Compared to previous game-specific algorithms, the proposed method achieves high performance within a lower computational budget, bringing a significant impact in combining game-specific approaches with LLM. One weakness is that the experiments are limited to board games, so it remains unclear whether this method can be generalized to other games (like video games) or other applications. Nevertheless, the overall strengths of the paper outweigh its weaknesses, I therefore recommend acceptance.